# Elucidating the Design Space of Diffusion-Based Generative Models

**Tero Karras**
NVIDIA
tkarras@nvidia.com

**Miika Aittala**
NVIDIA
maittala@nvidia.com

**Timo Aila**
NVIDIA
taila@nvidia.com

**Samuli Laine**
NVIDIA
slaine@nvidia.com

## Abstract

We argue that the theory and practice of diffusion-based generative models are currently unnecessarily convoluted and seek to remedy the situation by presenting a design space that clearly separates the concrete design choices. This lets us identify several changes to both the sampling and training processes, as well as preconditioning of the score networks. Together, our improvements yield new state-of-the-art FID of 1.79 for CIFAR-10 in a class-conditional setting and 1.97 in an unconditional setting, with much faster sampling (35 network evaluations per image) than prior designs. To further demonstrate their modular nature, we show that our design changes dramatically improve both the efficiency and quality obtainable with pre-trained score networks from previous work, including improving the FID of a previously trained ImageNet-64 model from 2.07 to near-SOTA 1.55, and after re-training with our proposed improvements to a new SOTA of 1.36.

## 1 Introduction

Diffusion-based generative models [45] have emerged as a powerful new framework for neural image synthesis, in both unconditional [16, 36, 48] and conditional [17, 35, 36, 38, 39, 41, 42, 48] settings, even surpassing the quality of GANs [13] in certain situations [9]. They are also rapidly finding use in other domains such as audio [27, 37] and video [19] generation, image segmentation [4, 54] and language translation [34]. As such, there is great interest in applying these models and improving them further in terms of image/distribution quality, training cost, and generation speed.

The literature on these models is dense on theory, and derivations of sampling schedule, training dynamics, noise level parameterization, etc., tend to be based as directly as possible on theoretical frameworks, which ensures that the models are on a solid theoretical footing. However, this approach has a danger of obscuring the available design space — a proposed model may appear as a tightly coupled package where no individual component can be modified without breaking the entire system.

As our first contribution, we take a look at the theory behind these models from a practical standpoint, focusing more on the "tangible" objects and algorithms that appear in the training and sampling phases, and less on the statistical processes from which they might be derived. The goal is to obtain better insights into how these components are linked together and what degrees of freedom are available in the design of the overall system. We focus on the broad class of models where a neural network is used to model the score [22] of a noise level dependent marginal distribution of the training data corrupted by Gaussian noise. Thus, our work is in the context of *denoising score matching* [51].

36th Conference on Neural Information Processing Systems (NeurIPS 2022).

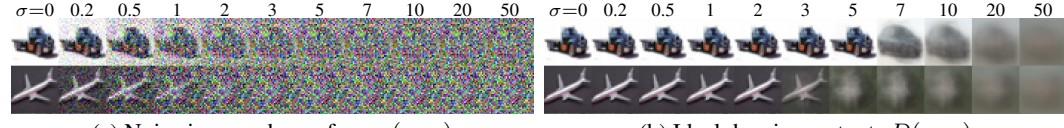

(a) Noisy images drawn from $p(\boldsymbol{x}; \sigma)$        (b) Ideal denoiser outputs $D(\boldsymbol{x}; \sigma)$

Figure 1: Denoising score matching on CIFAR-10. **(a)** Images from the training set corrupted with varying levels of additive Gaussian noise. High levels of noise lead to oversaturated colors; we normalize the images for cleaner visualization. **(b)** Optimal denoising result from minimizing Eq. 2 analytically (see Appendix B.3). With increasing noise level, the result approaches dataset mean.

Our second set of contributions concerns the sampling processes used to synthesize images using diffusion models. We identify the best-performing time discretization for sampling, apply a higher-order Runge–Kutta method for the sampling process, evaluate different sampler schedules, and analyze the usefulness of stochasticity in the sampling process. The result of these improvements is a significant drop in the number of sampling steps required during synthesis, and the improved sampler can be used as a drop-in replacement with several widely used diffusions models [36, 48].

The third set of contributions focuses on the training of the score-modeling neural network. While we continue to rely on the commonly used network architectures (DDPM [16], NCSN [47]), we provide the first principled analysis of the preconditioning of the networks' inputs, outputs, and loss functions in a diffusion model setting and derive best practices for improving the training dynamics. We also suggest an improved distribution of noise levels during training, and note that non-leaking augmentation [25] — typically used with GANs — is beneficial for diffusion models as well.

Taken together, our contributions enable significant improvements in result quality, e.g., leading to record FIDs of 1.79 for CIFAR-10 [28] and 1.36 for ImageNet [8] in 64×64 resolution. With all key ingredients of the design space explicitly tabulated, we believe that our approach will allow easier innovation on the individual components, and thus enable more extensive and targeted exploration of the design space of diffusion models. Our implementation and pre-trained models are available at
`https://github.com/NVlabs/edm`

## 2   Expressing diffusion models in a common framework

Let us denote the data distribution by $p_{\text{data}}(\boldsymbol{x})$, with standard deviation $\sigma_{\text{data}}$, and consider the family of mollified distributions $p(\boldsymbol{x}; \sigma)$ obtained by adding i.i.d. Gaussian noise of standard deviation $\sigma$ to the data. For $\sigma_{\max} \gg \sigma_{\text{data}}$, $p(\boldsymbol{x}; \sigma_{\max})$ is practically indistinguishable from pure Gaussian noise. The idea of diffusion models is to randomly sample a noise image $\boldsymbol{x}_0 \sim \mathcal{N}(\boldsymbol{0}, \sigma_{\max}^2 \mathbf{I})$, and sequentially denoise it into images $\boldsymbol{x}_i$ with noise levels $\sigma_0 = \sigma_{\max} > \sigma_1 > \cdots > \sigma_N = 0$ so that at each noise level $\boldsymbol{x}_i \sim p(\boldsymbol{x}_i; \sigma_i)$. The endpoint $\boldsymbol{x}_N$ of this process is thus distributed according to the data.

Song et al. [48] present a stochastic differential equation (SDE) that maintains the desired distribution $p$ as sample $\boldsymbol{x}$ evolves over time. This allows the above process to be implemented using a stochastic solver that both removes and adds noise at each iteration. They also give a corresponding "probability flow" ordinary differential equation (ODE) where the only source of randomness is the initial noise image $\boldsymbol{x}_0$. Contrary to the usual order of treatment, we begin by examining the ODE, as it offers a fruitful setting for analyzing sampling trajectories and their discretizations. The insights carry over to stochastic sampling, which we reintroduce as a generalization in Section 4.

**ODE formulation.**   A probability flow ODE [48] continuously increases or reduces noise level of the image when moving forward or backward in time, respectively. To specify the ODE, we must first choose a schedule $\sigma(t)$ that defines the desired noise level at time $t$. For example, setting $\sigma(t) \propto \sqrt{t}$ is mathematically natural, as it corresponds to constant-speed heat diffusion [12]. However, we will show in Section 3 that the choice of schedule has major practical implications and should not be made on the basis of theoretical convenience.

The defining characteristic of the probability flow ODE is that evolving a sample $\boldsymbol{x}_a \sim p(\boldsymbol{x}_a; \sigma(t_a))$ from time $t_a$ to $t_b$ (either forward or backward in time) yields a sample $\boldsymbol{x}_b \sim p(\boldsymbol{x}_b; \sigma(t_b))$. Following previous work [48], this requirement is satisfied (see Appendix B.1 and B.2) by

$$\mathrm{d}\boldsymbol{x} = -\dot{\sigma}(t)\, \sigma(t)\, \nabla_{\boldsymbol{x}} \log p\big(\boldsymbol{x}; \sigma(t)\big)\, \mathrm{d}t, \tag{1}$$

Table 1: Specific design choices employed by different model families. $N$ is the number of ODE solver iterations that we wish to execute during sampling. The corresponding sequence of time steps is $\{t_0, t_1, \ldots, t_N\}$, where $t_N = 0$. If the model was originally trained for specific choices of $N$ and $\{t_i\}$, the originals are denoted by $M$ and $\{u_j\}$, respectively. The denoiser is defined as $D_\theta(\boldsymbol{x}; \sigma) = c_{\text{skip}}(\sigma)\boldsymbol{x} + c_{\text{out}}(\sigma)F_\theta\big(c_{\text{in}}(\sigma)\boldsymbol{x}; c_{\text{noise}}(\sigma)\big)$; $F_\theta$ represents the raw neural network layers.

| | | VP [48] | VE [48] | iDDPM [36] + DDIM [46] | Ours ("EDM") |
|---|---|---|---|---|---|
| **Sampling (Section 3)** | | | | | |
| ODE solver | | Euler | Euler | Euler | 2$^{\text{nd}}$ order Heun |
| Time steps | $t_{i<N}$ | $1 + \frac{i}{N-1}(\epsilon_{\text{s}} - 1)$ | $\sigma_{\max}^2 \left(\sigma_{\min}^2/\sigma_{\max}^2\right)^{\frac{i}{N-1}}$ | $u_{\lfloor j_0 + \frac{M-1-j_0}{N-1} i + \frac{1}{2} \rfloor}$, where $u_M = 0$ $u_{j-1} = \sqrt{\frac{u_j^2+1}{\max(\bar\alpha_{j-1}/\bar\alpha_j, C_1)} - 1}$ | $\big(\sigma_{\max}^{\frac{1}{\rho}} + \frac{i}{N-1}(\sigma_{\min}^{\frac{1}{\rho}} - \sigma_{\max}^{\frac{1}{\rho}})\big)^\rho$ |
| Schedule | $\sigma(t)$ | $\sqrt{e^{\frac{1}{2}\beta_{\text{d}}t^2 + \beta_{\min}t} - 1}$ | $\sqrt{t}$ | $t$ | $t$ |
| Scaling | $s(t)$ | $1/\sqrt{e^{\frac{1}{2}\beta_{\text{d}}t^2 + \beta_{\min}t}}$ | 1 | 1 | 1 |
| **Network and preconditioning (Section 5)** | | | | | |
| Architecture of $F_\theta$ | | DDPM++ | NCSN++ | DDPM | (any) |
| Skip scaling $c_{\text{skip}}(\sigma)$ | | 1 | 1 | 1 | $\sigma_{\text{data}}^2 / \left(\sigma^2 + \sigma_{\text{data}}^2\right)$ |
| Output scaling $c_{\text{out}}(\sigma)$ | | $-\sigma$ | $\sigma$ | $-\sigma$ | $\sigma \cdot \sigma_{\text{data}} / \sqrt{\sigma_{\text{data}}^2 + \sigma^2}$ |
| Input scaling $c_{\text{in}}(\sigma)$ | | $1/\sqrt{\sigma^2 + 1}$ | 1 | $1/\sqrt{\sigma^2 + 1}$ | $1/\sqrt{\sigma^2 + \sigma_{\text{data}}^2}$ |
| Noise cond. $c_{\text{noise}}(\sigma)$ | | $(M-1)\,\sigma^{-1}(\sigma)$ | $\ln(\frac{1}{2}\sigma)$ | $M - 1 - \arg\min_j |u_j - \sigma|$ | $\frac{1}{4}\ln(\sigma)$ |
| **Training (Section 5)** | | | | | |
| Noise distribution | | $\sigma^{-1}(\sigma) \sim \mathcal{U}(\epsilon_{\text{t}}, 1)$ | $\ln(\sigma) \sim \mathcal{U}(\ln(\sigma_{\min}), \ln(\sigma_{\max}))$ | $\sigma = u_j, \; j \sim \mathcal{U}\{0, M-1\}$ | $\ln(\sigma) \sim \mathcal{N}(P_{\text{mean}}, P_{\text{std}}^2)$ |
| Loss weighting $\lambda(\sigma)$ | | $1/\sigma^2$ | $1/\sigma^2$ | $1/\sigma^2$ (note: *) | $\left(\sigma^2 + \sigma_{\text{data}}^2\right) / (\sigma \cdot \sigma_{\text{data}})^2$ |
| **Parameters** | | $\beta_{\text{d}} = 19.9, \beta_{\min} = 0.1$ $\epsilon_{\text{s}} = 10^{-3}, \epsilon_{\text{t}} = 10^{-5}$ $M = 1000$ | $\sigma_{\min} = 0.02$ $\sigma_{\max} = 100$ | $\bar\alpha_j = \sin^2(\frac{\pi}{2} \frac{j}{M(C_2+1)})$ $C_1 = 0.001, C_2 = 0.008$ $M = 1000, j_0 = 8^\dagger$ | $\sigma_{\min} = 0.002, \sigma_{\max} = 80$ $\sigma_{\text{data}} = 0.5, \rho = 7$ $P_{\text{mean}} = -1.2, P_{\text{std}} = 1.2$ |

\* iDDPM also employs a second loss term $L_{\text{vlb}}$     $\dagger$ In our tests, $j_0 = 8$ yielded better FID than $j_0 = 0$ used by iDDPM

where the dot denotes a time derivative. $\nabla_{\boldsymbol{x}} \log p(\boldsymbol{x}; \sigma)$ is the *score function* [22], a vector field that points towards higher density of data at a given noise level. Intuitively, an infinitesimal forward step of this ODE nudges the sample away from the data, at a rate that depends on the change in noise level. Equivalently, a backward step nudges the sample towards the data distribution.

**Denoising score matching.** The score function has the remarkable property that it does not depend on the generally intractable normalization constant of the underlying density function $p(\boldsymbol{x}; \sigma)$ [22], and thus can be much easier to evaluate. Specifically, if $D(\boldsymbol{x}; \sigma)$ is a denoiser function that minimizes the expected $L_2$ denoising error for samples drawn from $p_{\text{data}}$ separately for every $\sigma$, i.e.,

$$\mathbb{E}_{\boldsymbol{y} \sim p_{\text{data}}} \mathbb{E}_{\boldsymbol{n} \sim \mathcal{N}(\boldsymbol{0}, \sigma^2 \mathbf{I})} \| D(\boldsymbol{y} + \boldsymbol{n}; \sigma) - \boldsymbol{y} \|_2^2, \; \text{then} \; \nabla_{\boldsymbol{x}} \log p(\boldsymbol{x}; \sigma) = \big(D(\boldsymbol{x}; \sigma) - \boldsymbol{x}\big)/\sigma^2, \quad (2, 3)$$

where $\boldsymbol{y}$ is a training image and $\boldsymbol{n}$ is noise. In this light, the score function isolates the noise component from the signal in $\boldsymbol{x}$, and Eq. 1 amplifies (or diminishes) it over time. Figure 1 illustrates the behavior of ideal $D$ in practice. The key observation in diffusion models is that $D(\boldsymbol{x}; \sigma)$ can be implemented as a neural network $D_\theta(\boldsymbol{x}; \sigma)$ trained according to Eq. 2. Note that $D_\theta$ may include additional pre- and post-processing steps, such as scaling $\boldsymbol{x}$ to an appropriate dynamic range; we will return to such *preconditioning* in Section 5.

**Time-dependent signal scaling.** Some methods (see Appendix C.1) introduce an additional scale schedule $s(t)$ and consider $\boldsymbol{x} = s(t)\hat{\boldsymbol{x}}$ to be a scaled version of the original, non-scaled variable $\hat{\boldsymbol{x}}$. This changes the time-dependent probability density, and consequently also the ODE solution trajectories. The resulting ODE is a generalization of Eq. 1:

$$\mathrm{d}\boldsymbol{x} = \left[ \frac{\dot{s}(t)}{s(t)} \boldsymbol{x} - s(t)^2 \, \dot{\sigma}(t) \, \sigma(t) \, \nabla_{\boldsymbol{x}} \log p\left( \frac{\boldsymbol{x}}{s(t)}; \sigma(t) \right) \right] \mathrm{d}t. \quad (4)$$

Note that we explicitly undo the scaling of $\boldsymbol{x}$ when evaluating the score function to keep the definition of $p(\boldsymbol{x}; \sigma)$ independent of $s(t)$.

**Solution by discretization.** The ODE to be solved is obtained by substituting Eq. 3 into Eq. 4 to define the point-wise gradient, and the solution can be found by numerical integration, i.e., taking

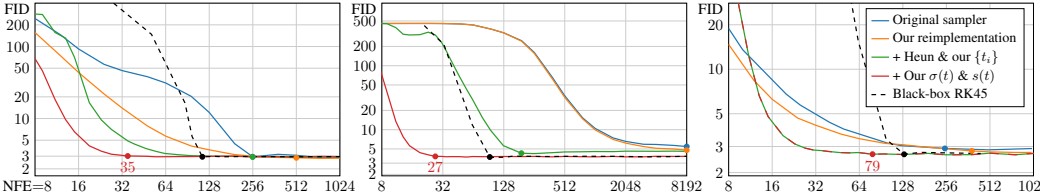

(a) Uncond. CIFAR-10, VP ODE     (b) Uncond. CIFAR-10, VE ODE     (c) Class-cond. ImageNet-64, DDIM

Figure 2: Comparison of deterministic sampling methods using three pre-trained models. For each curve, the dot indicates the lowest NFE whose FID is within 3% of the lowest observed FID.

finite steps over discrete time intervals. This requires choosing both the integration scheme (e.g., Euler or a variant of Runge–Kutta), as well as the discrete sampling times $\{t_0, t_1, \ldots, t_N\}$. Many prior works rely on Euler's method, but we show in Section 3 that a 2nd order solver offers a better computational tradeoff. For brevity, we do not provide a separate pseudocode for Euler's method applied to our ODE here, but it can be extracted from Algorithm 1 by omitting lines 6–8.

**Putting it together.**  Table 1 presents formulas for reproducing deterministic variants of three earlier methods in our framework. These methods were chosen because they are widely used and achieve state-of-the-art performance, but also because they were derived from different theoretical foundations. Some of our formulas appear quite different from the original papers as indirection and recursion have been removed; see Appendix C for details. The main purpose of this reframing is to bring into light all the independent components that often appear tangled together in previous work. In our framework, there are no implicit dependencies between the components — any choices (within reason) for the individual formulas will, in principle, lead to a functioning model. In other words, changing one component does not necessitate changes elsewhere in order to, e.g., maintain the property that the model converges to the data in the limit. In practice, some choices and combinations will of course work better than others.

## 3  Improvements to deterministic sampling

Improving the output quality and/or decreasing the computational cost of sampling are common topics in diffusion model research (e.g., [10, 24, 30, 31, 32, 36, 43, 50, 52, 53, 56]). Our hypothesis is that the choices related to the sampling process are largely independent of the other components, such as network architecture and training details. In other words, the training procedure of $D_\theta$ should not dictate $\sigma(t)$, $s(t)$, and $\{t_i\}$, nor vice versa; from the viewpoint of the sampler, $D_\theta$ is simply a black box [52, 53]. We test this by evaluating different samplers on three *pre-trained* models, each representing a different theoretical framework and model family. We first measure baseline results for these models using their original sampler implementations, and then bring these samplers into our unified framework using the formulas in Table 1, followed by our improvements. This allows us to evaluate different practical choices and propose general improvements to the sampling process that are applicable to all models.

We evaluate the "DDPM++ cont. (VP)" and "NCSN++ cont. (VE)" models by Song et al. [48] trained on unconditional CIFAR-10 [28] at 32×32, corresponding to the variance preserving (VP) and variance exploding (VE) formulations [48], originally inspired by DDPM [16] and SMLD [47]. We also evaluate the "ADM (dropout)" model by Dhariwal and Nichol [9] trained on class-conditional ImageNet [8] at 64×64, corresponding to the improved DDPM (iDDPM) formulation [36]. This model was trained using a discrete set of $M = 1000$ noise levels. Further details are given in Appendix C.

We evaluate the result quality in terms of Fréchet inception distance (FID) [15] computed between 50,000 generated images and all available real images. Figure 2 shows FID as a function of neural function evaluations (NFE), i.e., how many times $D_\theta$ is evaluated to produce a single image. Given that the sampling process is dominated entirely by the cost of $D_\theta$, improvements in NFE translate directly to sampling speed. The original deterministic samplers are shown in blue, and the reimplementations of these methods in our unified framework (orange) yield similar but consistently better results. The differences are explained by certain oversights in the original implementations as well as our more careful treatment of discrete noise levels in the case of DDIM; see Appendix C. Note that our reimplementations are fully specified by Algorithm 1 and Table 1, even though the original codebases are structured very differently from each other.

**Algorithm 1** Deterministic sampling using Heun's 2nd order method with arbitrary $\sigma(t)$ and $s(t)$.

---

1: **procedure** HEUNSAMPLER($D_\theta(\boldsymbol{x}; \sigma)$, $\sigma(t)$, $s(t)$, $t_{i \in \{0,\ldots,N\}}$)
2:      **sample** $\boldsymbol{x}_0 \sim \mathcal{N}\big(\boldsymbol{0},\ \sigma^2(t_0)\, s^2(t_0)\, \mathbf{I}\big)$      ▷ Generate initial sample at $t_0$
3:      **for** $i \in \{0,\ldots,N-1\}$ **do**      ▷ Solve Eq. 4 over $N$ time steps
4:          $\boldsymbol{d}_i \leftarrow \left( \dfrac{\dot{\sigma}(t_i)}{\sigma(t_i)} + \dfrac{\dot{s}(t_i)}{s(t_i)} \right) \boldsymbol{x}_i - \dfrac{\dot{\sigma}(t_i) s(t_i)}{\sigma(t_i)} D_\theta\left( \dfrac{\boldsymbol{x}_i}{s(t_i)}; \sigma(t_i) \right)$      ▷ Evaluate $\mathrm{d}\boldsymbol{x}/\mathrm{d}t$ at $t_i$
5:          $\boldsymbol{x}_{i+1} \leftarrow \boldsymbol{x}_i + (t_{i+1} - t_i) \boldsymbol{d}_i$      ▷ Take Euler step from $t_i$ to $t_{i+1}$
6:          **if** $\sigma(t_{i+1}) \neq 0$ **then**      ▷ Apply 2nd order correction unless $\sigma$ goes to zero
7:              $\boldsymbol{d}_i' \leftarrow \left( \dfrac{\dot{\sigma}(t_{i+1})}{\sigma(t_{i+1})} + \dfrac{\dot{s}(t_{i+1})}{s(t_{i+1})} \right) \boldsymbol{x}_{i+1} - \dfrac{\dot{\sigma}(t_{i+1}) s(t_{i+1})}{\sigma(t_{i+1})} D_\theta\left( \dfrac{\boldsymbol{x}_{i+1}}{s(t_{i+1})}; \sigma(t_{i+1}) \right)$    ▷ Eval. $\mathrm{d}\boldsymbol{x}/\mathrm{d}t$ at $t_{i+1}$
8:              $\boldsymbol{x}_{i+1} \leftarrow \boldsymbol{x}_i + (t_{i+1} - t_i)\big( \tfrac{1}{2}\boldsymbol{d}_i + \tfrac{1}{2}\boldsymbol{d}_i' \big)$      ▷ Explicit trapezoidal rule at $t_{i+1}$
9:      **return** $\boldsymbol{x}_N$      ▷ Return noise-free sample at $t_N$

---

**Discretization and higher-order integrators.** Solving an ODE numerically is necessarily an approximation of following the true solution trajectory. At each step, the solver introduces *truncation error* that accumulates over the course of $N$ steps. The local error generally scales superlinearly with respect to step size, and thus increasing $N$ improves the accuracy of the solution.

The commonly used Euler's method is a first order ODE solver with $\mathcal{O}(h^2)$ local error with respect to step size $h$. Higher-order Runge–Kutta methods [49] scale more favorably but require multiple evaluations of $D_\theta$ per step. Linear multistep methods have also been recently proposed for sampling diffusion models [30, 56]. Through extensive tests, we have found Heun's 2nd order method [2] (a.k.a. improved Euler, trapezoidal rule) — previously explored in the context of diffusion models by Jolicoeur-Martineau et al. [24] — to provide an excellent tradeoff between truncation error and NFE. As illustrated in Algorithm 1, it introduces an additional correction step for $\boldsymbol{x}_{i+1}$ to account for change in $\mathrm{d}\boldsymbol{x}/\mathrm{d}t$ between $t_i$ and $t_{i+1}$. This correction leads to $\mathcal{O}(h^3)$ local error at the cost of one additional evaluation of $D_\theta$ per step. Note that stepping to $\sigma = 0$ would result in a division by zero, so we revert to Euler's method in this case. We discuss the general family of 2nd order solvers in Appendix D.2.

The time steps $\{t_i\}$ determine how the step sizes and thus truncation errors are distributed between different noise levels. We provide a detailed analysis in Appendix D.1, concluding that the step size should decrease monotonically with decreasing $\sigma$ and it does not need to vary on a per-sample basis. We adopt a parameterized scheme where the time steps are defined according to a sequence of noise levels $\{\sigma_i\}$, i.e., $t_i = \sigma^{-1}(\sigma_i)$. We set $\sigma_{i<N} = (Ai + B)^\rho$ and select the constants $A$ and $B$ so that $\sigma_0 = \sigma_{\max}$ and $\sigma_{N-1} = \sigma_{\min}$, which gives

$$\sigma_{i<N} = \big( \sigma_{\max}^{\frac{1}{\rho}} + \tfrac{i}{N-1}(\sigma_{\min}^{\frac{1}{\rho}} - \sigma_{\max}^{\frac{1}{\rho}}) \big)^\rho \quad \text{and} \quad \sigma_N = 0. \tag{5}$$

Here $\rho$ controls how much the steps near $\sigma_{\min}$ are shortened at the expense of longer steps near $\sigma_{\max}$. Our analysis in Appendix D.1 shows that setting $\rho = 3$ nearly equalizes the truncation error at each step, but that $\rho$ in range of 5 to 10 performs much better for sampling images. This suggests that errors near $\sigma_{\min}$ have a large impact. We set $\rho = 7$ for the remainder of this paper.

Results for Heun's method and Eq. 5 are shown as the green curves in Figure 2. We observe consistent improvement in all cases: Heun's method reaches the same FID as Euler's method with considerably lower NFE.

**Trajectory curvature and noise schedule.** The shape of the ODE solution trajectories is defined by functions $\sigma(t)$ and $s(t)$. The choice of these functions offers a way to reduce the truncation errors discussed above, as their magnitude can be expected to scale proportional to the curvature of $\mathrm{d}\boldsymbol{x}/\mathrm{d}t$. We argue that the best choice for these functions is $\sigma(t) = t$ and $s(t) = 1$, which is also the choice made in DDIM [46]. With this choice, the ODE of Eq. 4 simplifies to $\mathrm{d}\boldsymbol{x}/\mathrm{d}t = \big( \boldsymbol{x} - D(\boldsymbol{x}; t) \big)/t$ and $\sigma$ and $t$ become interchangeable.

An immediate consequence is that at any $\boldsymbol{x}$ and $t$, a single Euler step to $t = 0$ yields the denoised image $D_\theta(\boldsymbol{x}; t)$. The tangent of the solution trajectory therefore always points towards the denoiser output. This can be expected to change only slowly with the noise level, which corresponds to largely linear solution trajectories. The 1D ODE sketch of Figure 3c supports this intuition; the solution trajectories approach linear at both large and small noise levels, and have substantial curvature in only a small region in between. The same effect can be seen with real data in Figure 1b, where the

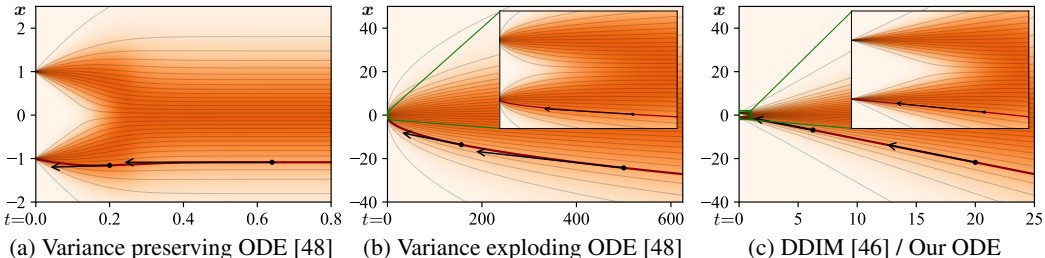

Figure 3: A sketch of ODE curvature in 1D where $p_{\text{data}}$ is two Dirac peaks at $\boldsymbol{x} = \pm 1$. Horizontal $t$ axis is chosen to show $\sigma \in [0, 25]$ in each plot, with insets showing $\sigma \in [0, 1]$ near the data. Example local gradients are shown with black arrows. **(a)** Variance preserving ODE of Song et al. [48] has solution trajectories that flatten out to horizontal lines at large $\sigma$. Local gradients start pointing towards data only at small $\sigma$. **(b)** Variance exploding variant has extreme curvature near data and the solution trajectories are curved everywhere. **(c)** With the schedule used by DDIM [46] and us, as $\sigma$ increases the solution trajectories approach straight lines that point towards the mean of data. As $\sigma \to 0$, the trajectories become linear and point towards the data manifold.

change between different denoiser targets occurs in a relatively narrow $\sigma$ range. With the advocated schedule, this corresponds to high ODE curvature being limited to this same range.

The effect of setting $\sigma(t) = t$ and $s(t) = 1$ is shown as the red curves in Figure 2. As DDIM already employs these same choices, the red curve is identical to the green one for ImageNet-64. However, VP and VE benefit considerably from switching away from their original schedules.

**Discussion.** The choices that we made in this section to improve deterministic sampling are summarized in the *Sampling* part of Table 1. Together, they reduce the NFE needed to reach high-quality results by a large factor: $7.3\times$ for VP, $300\times$ for VE, and $3.2\times$ for DDIM, corresponding to the highlighted NFE values in Figure 2. In practice, we can generate 26.3 high-quality CIFAR-10 images per second on a single NVIDIA V100. The consistency of improvements corroborates our hypothesis that the sampling process is orthogonal to how each model was originally trained. As further validation, we show results for the adaptive RK45 method [11] using our schedule as the dashed black curves in Figure 2; the cost of this sophisticated ODE solver outweighs its benefits.

## 4   Stochastic sampling

Deterministic sampling offers many benefits, e.g., the ability to turn real images into their corresponding latent representations by inverting the ODE. However, it tends to lead to worse output quality [46, 48] than stochastic sampling that injects fresh noise into the image in each step. Given that ODEs and SDEs recover the same distributions in theory, what exactly is the role of stochasticity?

**Background.** The SDEs of Song et al. [48] can be generalized [20, 55] as a sum of the probability flow ODE of Eq. 1 and a time-varying *Langevin diffusion* SDE [14] (see Appendix B.5):

$$\mathrm{d}\boldsymbol{x}_{\pm} = \underbrace{-\dot{\sigma}(t)\sigma(t)\nabla_{\boldsymbol{x}} \log p\big(\boldsymbol{x}; \sigma(t)\big)\,\mathrm{d}t}_{\text{probability flow ODE (Eq. 1)}} \pm \underbrace{\underbrace{\beta(t)\sigma(t)^2 \nabla_{\boldsymbol{x}} \log p\big(\boldsymbol{x}; \sigma(t)\big)\,\mathrm{d}t}_{\text{deterministic noise decay}} + \underbrace{\sqrt{2\beta(t)}\sigma(t)\,\mathrm{d}\omega_t}_{\text{noise injection}}}_{\text{Langevin diffusion SDE}}, \quad (6)$$

where $\omega_t$ is the standard Wiener process. $\mathrm{d}\boldsymbol{x}_+$ and $\mathrm{d}\boldsymbol{x}_-$ are now separate SDEs for moving forward and backward in time, related by the time reversal formula of Anderson [1]. The Langevin term can further be seen as a combination of a deterministic score-based denoising term and a stochastic noise injection term, whose net noise level contributions cancel out. As such, $\beta(t)$ effectively expresses the relative rate at which existing noise is replaced with new noise. The SDEs of Song et al. [48] are recovered with the choice $\beta(t) = \dot{\sigma}(t)/\sigma(t)$, whereby the score vanishes from the forward SDE.

This perspective reveals why stochasticity is helpful in practice: The implicit Langevin diffusion drives the sample towards the desired marginal distribution at a given time, actively correcting for any errors made in earlier sampling steps. On the other hand, approximating the Langevin term

---

**Algorithm 2** Our stochastic sampler with $\sigma(t) = t$ and $s(t) = 1$.

---

1: **procedure** STOCHASTICSAMPLER($D_\theta(\boldsymbol{x};\sigma)$, $t_{i\in\{0,\dots,N\}}$, $\gamma_{i\in\{0,\dots,N-1\}}$, $S_{\text{noise}}$)
2:      **sample** $\boldsymbol{x}_0 \sim \mathcal{N}\big(\boldsymbol{0},\ t_0^2\,\mathbf{I}\big)$
3:      **for** $i \in \{0,\dots,N-1\}$ **do**        $\triangleright\ \gamma_i = \begin{cases} \min\big(\frac{S_{\text{churn}}}{N}, \sqrt{2}-1\big) & \text{if } t_i \in [S_{\text{tmin}}, S_{\text{tmax}}] \\ 0 & \text{otherwise} \end{cases}$
4:          **sample** $\boldsymbol{\epsilon}_i \sim \mathcal{N}\big(\boldsymbol{0},\ S_{\text{noise}}^2\,\mathbf{I}\big)$
5:          $\hat{t}_i \leftarrow t_i + \gamma_i t_i$        $\triangleright$ Select temporarily increased noise level $\hat{t}_i$
6:          $\hat{\boldsymbol{x}}_i \leftarrow \boldsymbol{x}_i + \sqrt{\hat{t}_i^2 - t_i^2}\,\boldsymbol{\epsilon}_i$        $\triangleright$ Add new noise to move from $t_i$ to $\hat{t}_i$
7:          $\boldsymbol{d}_i \leftarrow \big(\hat{\boldsymbol{x}}_i - D_\theta(\hat{\boldsymbol{x}}_i; \hat{t}_i)\big)/\hat{t}_i$        $\triangleright$ Evaluate $\mathrm{d}\boldsymbol{x}/\mathrm{d}t$ at $\hat{t}_i$
8:          $\boldsymbol{x}_{i+1} \leftarrow \hat{\boldsymbol{x}}_i + (t_{i+1} - \hat{t}_i)\boldsymbol{d}_i$        $\triangleright$ Take Euler step from $\hat{t}_i$ to $t_{i+1}$
9:          **if** $t_{i+1} \neq 0$ **then**
10:             $\boldsymbol{d}_i' \leftarrow \big(\boldsymbol{x}_{i+1} - D_\theta(\boldsymbol{x}_{i+1}; t_{i+1})\big)/t_{i+1}$        $\triangleright$ Apply 2$^{\text{nd}}$ order correction
11:             $\boldsymbol{x}_{i+1} \leftarrow \hat{\boldsymbol{x}}_i + (t_{i+1} - \hat{t}_i)\big(\frac{1}{2}\boldsymbol{d}_i + \frac{1}{2}\boldsymbol{d}_i'\big)$
12:      **return** $\boldsymbol{x}_N$

---

with discrete SDE solver steps introduces error in itself. Previous results [3, 24, 46, 48] suggest that non-zero $\beta(t)$ is helpful, but as far as we can tell, the implicit choice for $\beta(t)$ in Song et al. [48] enjoys no special properties. Hence, the optimal amount of stochasticity should be determined empirically.

**Our stochastic sampler.** We propose a stochastic sampler that combines our 2$^{\text{nd}}$ order deterministic ODE integrator with explicit Langevin-like "churn" of adding and removing noise. A pseudocode is given in Algorithm 2. At each step $i$, given the sample $\boldsymbol{x}_i$ at noise level $t_i (= \sigma(t_i))$, we perform two sub-steps. First, we add noise to the sample according to a factor $\gamma_i \geq 0$ to reach a higher noise level $\hat{t}_i = t_i + \gamma_i t_i$. Second, from the increased-noise sample $\hat{\boldsymbol{x}}_i$, we solve the ODE backward from $\hat{t}_i$ to $t_{i+1}$ with a single step. This yields a sample $\boldsymbol{x}_{i+1}$ with noise level $t_{i+1}$, and the iteration continues. We stress that this is not a general-purpose SDE solver, but a sampling procedure tailored for the specific problem. Its correctness stems from the alternation of two sub-steps that each maintain the correct distribution (up to truncation error in the ODE step). The predictor-corrector sampler of Song et al. [48] has a conceptually similar structure to ours.

To analyze the main difference between our method and Euler–Maruyama, we first note a subtle discrepancy in the latter when discretizing Eq. 6. One can interpret Euler–Maruyama as first adding noise and then performing an ODE step, not from the intermediate state after noise injection, but assuming that $\boldsymbol{x}$ and $\sigma$ remained at the initial state at the beginning of the iteration step. In our method, the parameters used to evaluate $D_\theta$ on line 7 of Algorithm 2 correspond to the state after noise injection, whereas an Euler–Maruyama -like method would use $\boldsymbol{x}_i; t_i$ instead of $\hat{\boldsymbol{x}}_i; \hat{t}_i$. In the limit of $\Delta_t$ approaching zero there may be no difference between these choices, but the distinction appears to become significant when pursuing low NFE with large steps.

**Practical considerations.** Increasing the amount of stochasticity is effective in correcting errors made by earlier sampling steps, but it has its own drawbacks. We have observed (see Appendix E.1) that excessive Langevin-like addition and removal of noise results in gradual loss of detail in the generated images with all datasets and denoiser networks. There is also a drift toward oversaturated colors at very low and high noise levels. We suspect that practical denoisers induce a slightly non-conservative vector field in Eq. 3, violating the premises of Langevin diffusion and causing these detrimental effects. Notably, our experiments with analytical denoisers (such as the one in Figure 1b) have not shown such degradation.

If the degradation is caused by flaws in $D_\theta(\boldsymbol{x};\sigma)$, they can only be remedied using heuristic means during sampling. We address the drift toward oversaturated colors by only enabling stochasticity within a specific range of noise levels $t_i \in [S_{\text{tmin}}, S_{\text{tmax}}]$. For these noise levels, we define $\gamma_i = S_{\text{churn}}/N$, where $S_{\text{churn}}$ controls the overall amount of stochasticity. We further clamp $\gamma_i$ to never introduce more new noise than what is already present in the image. Finally, we have found that the loss of detail can be partially counteracted by setting $S_{\text{noise}}$ slightly above 1 to inflate the standard deviation for the newly added noise. This suggests that a major component of the hypothesized non-conservativity of $D_\theta(\boldsymbol{x};\sigma)$ is a tendency to remove slightly too much noise — most likely due to regression toward the mean that can be expected to happen with any $L_2$-trained denoiser [29].

**Evaluation.** Figure 4 shows that our stochastic sampler outperforms previous samplers [24, 36, 48] by a significant margin, especially at low step counts. Jolicoeur-Martineau et al. [24] use a standard

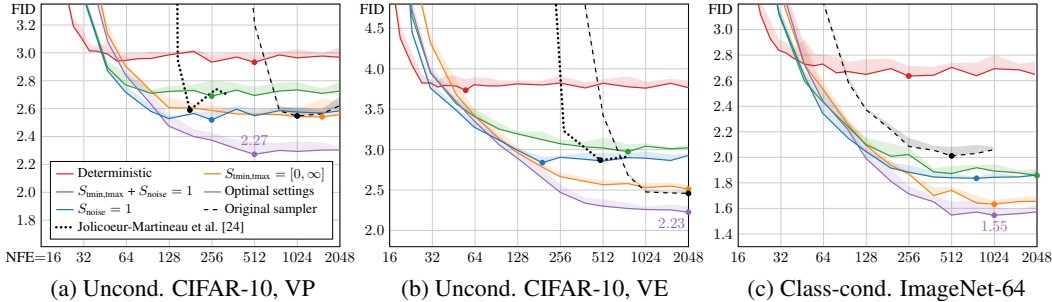

(a) Uncond. CIFAR-10, VP     (b) Uncond. CIFAR-10, VE     (c) Class-cond. ImageNet-64

Figure 4: Evaluation of our stochastic sampler (Algorithm 2). The purple curve corresponds to optimal choices for $\{S_{\text{churn}}, S_{\text{tmin}}, S_{\text{tmax}}, S_{\text{noise}}\}$; orange, blue, and green correspond to disabling the effects of $S_{\text{tmin,tmax}}$ and/or $S_{\text{noise}}$. The red curves show reference results for our deterministic sampler (Algorithm 1), equivalent to setting $S_{\text{churn}} = 0$. The dashed black curves correspond to the original stochastic samplers from previous work: Euler–Maruyama [48] for VP, predictor-corrector [48] for VE, and iDDPM [36] for ImageNet-64. The dots indicate lowest observed FID.

higher-order adaptive SDE solver [40] and its performance is a good baseline for such solvers in general. Our sampler has been tailored to the use case by, e.g., performing noise injection and ODE step sequentially, and it is not adaptive. It is an open question if adaptive solvers can be a net win over a well-tuned fixed schedule in sampling diffusion models.

Through sampler improvements alone, we are able to bring the ImageNet-64 model that originally achieved FID 2.07 [9] to 1.55 that is very close to the state-of-the-art; previously, FID 1.48 has been reported for cascaded diffusion [17], 1.55 for classifier-free guidance [18], and 1.52 for StyleGAN-XL [44]. While our results showcase the potential gains achievable through sampler improvements, they also highlight the main shortcoming of stochasticity: For best results, one must make several heuristic choices — either implicit or explicit — that depend on the specific model. Indeed, we had to find the optimal values of $\{S_{\text{churn}}, S_{\text{tmin}}, S_{\text{tmax}}, S_{\text{noise}}\}$ on a case-by-case basis using grid search (Appendix E.2). This raises a general concern that using stochastic sampling as the primary means of evaluating model improvements may inadvertently end up influencing the design choices related to model architecture and training.

## 5  Preconditioning and training

There are various known good practices for training neural networks in a supervised fashion. For example, it is advisable to keep input and output signal magnitudes fixed to, e.g., unit variance, and to avoid large variation in gradient magnitudes on a per-sample basis [5, 21]. Training a neural network to model $D$ directly would be far from ideal — for example, as the input $\boldsymbol{x} = \boldsymbol{y} + \boldsymbol{n}$ is a combination of clean signal $\boldsymbol{y}$ and noise $\boldsymbol{n} \sim \mathcal{N}(\boldsymbol{0}, \sigma^2\mathbf{I})$, its magnitude varies immensely depending on noise level $\sigma$. For this reason, the common practice is to not represent $D_\theta$ as a neural network directly, but instead train a different network $F_\theta$ from which $D_\theta$ is derived.

Previous methods [36, 46, 48] address the input scaling via a $\sigma$-dependent normalization factor and attempt to precondition the output by training $F_\theta$ to predict $\boldsymbol{n}$ scaled to unit variance, from which the signal is then reconstructed via $D_\theta(\boldsymbol{x}; \sigma) = \boldsymbol{x} - \sigma F_\theta(\cdot)$. This has the drawback that at large $\sigma$, the network needs to fine-tune its output carefully to cancel out the existing noise $\boldsymbol{n}$ exactly and give the output at the correct scale; note that any errors made by the network are amplified by a factor of $\sigma$. In this situation, it would seem much easier to predict the expected output $D(\boldsymbol{x}; \sigma)$ directly. In the same spirit as previous parameterizations that adaptively mix signal and noise (e.g., [10, 43, 50]), we propose to precondition the neural network with a $\sigma$-dependent skip connection that allows it to estimate either $\boldsymbol{y}$ or $\boldsymbol{n}$, or something in between. We thus write $D_\theta$ in the following form:

$$D_\theta(\boldsymbol{x}; \sigma) = c_{\text{skip}}(\sigma)\,\boldsymbol{x} + c_{\text{out}}(\sigma)\,F_\theta\big(c_{\text{in}}(\sigma)\,\boldsymbol{x};\ c_{\text{noise}}(\sigma)\big), \tag{7}$$

where $F_\theta$ is the neural network to be trained, $c_{\text{skip}}(\sigma)$ modulates the skip connection, $c_{\text{in}}(\sigma)$ and $c_{\text{out}}(\sigma)$ scale the input and output magnitudes, and $c_{\text{noise}}(\sigma)$ maps noise level $\sigma$ into a conditioning input for $F_\theta$. Taking a weighted expectation of Eq. 2 over the noise levels gives the overall training loss $\mathbb{E}_{\sigma, \boldsymbol{y}, \boldsymbol{n}}\big[\lambda(\sigma)\,\|D(\boldsymbol{y} + \boldsymbol{n}; \sigma) - \boldsymbol{y}\|_2^2\big]$, where $\sigma \sim p_{\text{train}}$, $\boldsymbol{y} \sim p_{\text{data}}$, and $\boldsymbol{n} \sim \mathcal{N}(\boldsymbol{0}, \sigma^2\mathbf{I})$. The probability of sampling a given noise level $\sigma$ is given by $p_{\text{train}}(\sigma)$ and the corresponding weight is given by

Table 2: Evaluation of our training improvements. The starting point (config A) is VP & VE using our **deterministic** sampler. At the end (configs E,F), VP & VE only differ in the architecture of $F_\theta$.

| | CIFAR-10 [28] at 32×32 | | | | FFHQ [26] 64×64 | | AFHQv2 [7] 64×64 | |
| | Conditional | | Unconditional | | Unconditional | | Unconditional | |
| **Training configuration** | VP | VE | VP | VE | VP | VE | VP | VE |
|---|---|---|---|---|---|---|---|---|
| A  Baseline [48]   (*pre-trained) | 2.48 | 3.11 | 3.01* | 3.77* | 3.39 | 25.95 | 2.58 | 18.52 |
| B  + Adjust hyperparameters | 2.18 | 2.48 | 2.51 | 2.94 | 3.13 | 22.53 | 2.43 | 23.12 |
| C  + Redistribute capacity | 2.08 | 2.52 | 2.31 | 2.83 | 2.78 | 41.62 | 2.54 | 15.04 |
| D  + Our preconditioning | 2.09 | 2.64 | 2.29 | 3.10 | 2.94 | 3.39 | 2.79 | 3.81 |
| E  + Our loss function | 1.88 | 1.86 | 2.05 | 1.99 | 2.60 | 2.81 | 2.29 | 2.28 |
| F  + Non-leaky augmentation | **1.79** | **1.79** | **1.97** | **1.98** | **2.39** | **2.53** | **1.96** | **2.16** |
| NFE | 35 | 35 | 35 | 35 | 79 | 79 | 79 | 79 |

$\lambda(\sigma)$. We can equivalently express this loss with respect to the raw network output $F_\theta$ in Eq. 7:

$$\mathbb{E}_{\sigma, \boldsymbol{y}, \boldsymbol{n}}\Big[ \underbrace{\lambda(\sigma)\, c_{\text{out}}(\sigma)^2}_{\text{effective weight}} \big\| \underbrace{F_\theta\big(c_{\text{in}}(\sigma) \cdot (\boldsymbol{y}+\boldsymbol{n}); c_{\text{noise}}(\sigma)\big)}_{\text{network output}} - \underbrace{\tfrac{1}{c_{\text{out}}(\sigma)}\big(\boldsymbol{y} - c_{\text{skip}}(\sigma) \cdot (\boldsymbol{y}+\boldsymbol{n})\big)}_{\text{effective training target}} \big\|_2^2 \Big]. \quad (8)$$

This form reveals the effective training target of $F_\theta$, allowing us to determine suitable choices for the preconditioning functions from first principles. As detailed in Appendix B.6, we derive our choices shown in Table 1 by requiring network inputs and training targets to have unit variance ($c_{\text{in}}$, $c_{\text{out}}$), and amplifying errors in $F_\theta$ as little as possible ($c_{\text{skip}}$). The formula for $c_{\text{noise}}$ is chosen empirically.

Table 2 shows FID for a series of training setups, evaluated using our deterministic sampler from Section 3. We start with the baseline training setup of Song et al. [48], which differs considerably between the VP and VE cases; we provide separate results for each (config A). To obtain a more meaningful point of comparison, we re-adjust the basic hyperparameters (config B) and improve the expressive power of the model (config C) by removing the lowest-resolution layers and doubling the capacity of the highest-resolution layers instead; see Appendix F.3 for further details. We then replace the original choices of $\{c_{\text{in}}, c_{\text{out}}, c_{\text{noise}}, c_{\text{skip}}\}$ with our preconditioning (config D), which keeps the results largely unchanged — except for VE that improves considerably at 64×64 resolution. Instead of improving FID per se, the main benefit of our preconditioning is that it makes the training more robust, enabling us to turn our focus on redesigning the loss function without adverse effects.

**Loss weighting and sampling.**   Eq. 8 shows that training $F_\theta$ as preconditioned in Eq. 7 incurs an effective per-sample loss weight of $\lambda(\sigma)c_{\text{out}}(\sigma)^2$. To balance the effective loss weights, we set $\lambda(\sigma) = 1/c_{\text{out}}(\sigma)^2$, which also equalizes the initial training loss over the entire $\sigma$ range as shown in Figure 5a (green curve). Finally, we need to select $p_{\text{train}}(\sigma)$, i.e., how to choose noise levels during training. Inspecting the per-$\sigma$ loss after training (blue and orange curves) reveals that a significant reduction is possible only at intermediate noise levels; at very low levels, it is both difficult and irrelevant to discern the vanishingly small noise component, whereas at high levels the training targets are always dissimilar from the correct answer that approaches dataset average. Therefore, we target the training efforts to the relevant range using a simple log-normal distribution for $p_{\text{train}}(\sigma)$ as detailed in Table 1 and illustrated in Figure 5a (red curve).

Table 2 shows that our proposed $p_{\text{train}}$ and $\lambda$ (config E) lead to a dramatic improvement in FID in all cases when used in conjunction with our preconditioning (config D). In concurrent work, Choi et al. [6] propose a similar scheme to prioritize noise levels that are most relevant w.r.t. forming the perceptually recognizable content of the image. However, they only consider the choice of $\lambda$ in isolation, which results in a smaller overall improvement.

**Augmentation regularization.**   To prevent potential overfitting that often plagues diffusion models with smaller datasets, we borrow an augmentation pipeline from the GAN literature [25]. The pipeline consists of various geometric transformations (see Appendix F.2) that we apply to a training image prior to adding noise. To prevent the augmentations from leaking to the generated images, we provide the augmentation parameters as a conditioning input to $F_\theta$; during inference we set the them to zero to guarantee that only non-augmented images are generated. Table 2 shows that data augmentation provides a consistent improvement (config F) that yields new state-of-the-art FIDs of 1.79 and 1.97 for conditional and unconditional CIFAR-10, beating the previous records of 1.85 [44] and 2.10 [50].

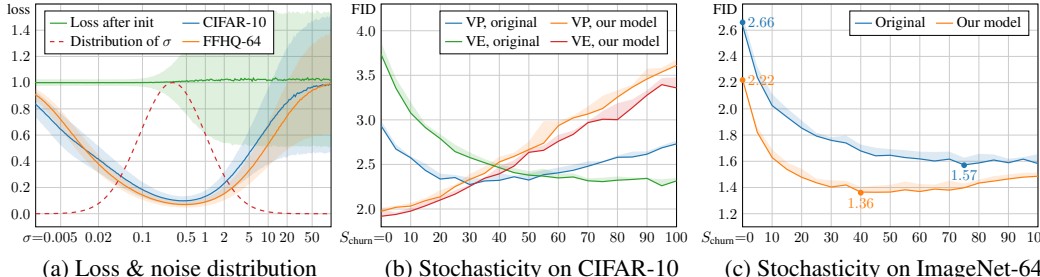

| (a) Loss & noise distribution | (b) Stochasticity on CIFAR-10 | (c) Stochasticity on ImageNet-64 |

Figure 5: **(a)** Observed initial (green) and final loss per noise level, representative of the the $32{\times}32$ (blue) and $64{\times}64$ (orange) models considered in this paper. The shaded regions represent the standard deviation over 10k random samples. Our proposed training sample density is shown by the dashed red curve. **(b)** Effect of $S_{\text{churn}}$ on unconditional CIFAR-10 with 256 steps (NFE = 511). For the original training setup of Song et al. [48], stochastic sampling is highly beneficial (blue, green), while deterministic sampling ($S_{\text{churn}} = 0$) leads to relatively poor FID. For our training setup, the situation is reversed (orange, red); stochastic sampling is not only unnecessary but harmful. **(c)** Effect of $S_{\text{churn}}$ on class-conditional ImageNet-64 with 256 steps (NFE = 511). In this more challenging scenario, stochastic sampling turns out to be useful again. Our training setup improves the results for both deterministic and stochastic sampling.

**Stochastic sampling revisited.** Interestingly, the relevance of stochastic sampling appears to diminish as the model itself improves, as shown in Figure 5b,c. When using our training setup in CIFAR-10 (Figure 5b), the best results were obtained with deterministic sampling, and any amount of stochastic sampling was detrimental.

**ImageNet-64.** As a final experiment, we trained a class-conditional ImageNet-64 model from scratch using our proposed training improvements. This model achieved a new state-of-the-art FID of 1.36 compared to the previous record of 1.48 [17]. We used the ADM architecture [9] with no changes, and trained it using our config E with minimal tuning; see Appendix F.3 for details. We did not find overfitting to be a concern, and thus chose to not employ augmentation regularization. As shown in Figure 5c, the optimal amount of stochastic sampling was much lower than with the pre-trained model, but unlike with CIFAR-10, stochastic sampling was clearly better than deterministic sampling. This suggests that more diverse datasets continue to benefit from stochastic sampling.

## 6  Conclusions

Our approach of putting diffusion models to a common framework exposes a modular design. This allows a targeted investigation of individual components, potentially helping to better cover the viable design space. In our tests this let us simply replace the samplers in various earlier models, drastically improving the results. For example, in ImageNet-64 our sampler turned an average model (FID 2.07) to a challenger (1.55) for the previous SOTA model (1.48) [17], and with training improvements achieved SOTA FID of 1.36. We also obtained new state-of-the-art results on CIFAR-10 while using only 35 model evaluations, deterministic sampling, and a small network. The current high-resolution diffusion models rely either on separate super-resolution steps [17, 35, 39], subspace projection [23], very large networks [9, 48], or hybrid approaches [38, 41, 50] — we believe that our contributions are orthogonal to these extensions. That said, many of our parameter values may need to be re-adjusted for higher resolution datasets. Furthermore, we feel that the precise interaction between stochastic sampling and the training objective remains an interesting question for future work.

**Societal impact.** Our advances in sample quality can potentially amplify negative societal effects when used in a large-scale system like DALL·E 2, including types of disinformation or emphasizing sterotypes and harmful biases [33]. The training and sampling of diffusion models needs a lot of electricity; our project consumed ~250MWh on an in-house cluster of NVIDIA V100s.

**Acknowledgments.** We thank Jaakko Lehtinen, Ming-Yu Liu, Tuomas Kynkäänniemi, Axel Sauer, Arash Vahdat, and Janne Hellsten for discussions and comments, and Tero Kuosmanen, Samuel Klenberg, and Janne Hellsten for maintaining our compute infrastructure.

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
