# Appendices:
# Elucidating the Design Space of Diffusion-Based Generative Models

## A  Additional results

Figure 6 presents generated images for class-conditional ImageNet-64 [3] using the pre-trained ADM model by Dhariwal and Nichol [4]. The original DDIM [14] and iDDPM [13] samplers are compared to ours in both deterministic and stochastic settings (Sections 3 and 4). Figure 7 shows the corresponding results that we obtain by training the model from scratch using our improved training configuration (Section 5).

The original samplers and training configurations by Song et al. [15] are compared to ours in Figures 8 and 9 (unconditional CIFAR-10 [12]), Figure 10 (class-conditional CIFAR-10), and Figure 11 (FFHQ [11] and AFHQv2 [2]). For ease of comparison, the same latent codes $x_0$ are used for each dataset/scenario across different training configurations and ODE choices. Figure 12 shows generated image quality with various NFE when using deterministic sampling.

Tables 3 and 4 summarize the numerical results on deterministic and stochastic sampling methods in various datasets, previously shown as functions of NFE in Figures 2 and 4.

## B  Derivation of formulas

### B.1  Original ODE / SDE formulation from previous work

Song et al. [15] define their forward SDE (Eq. 5 in [15]) as

$$\mathrm{d}\boldsymbol{x} = \boldsymbol{f}(\boldsymbol{x}, t)\,\mathrm{d}t + g(t)\,\mathrm{d}\omega_t, \tag{9}$$

where $\omega_t$ is the standard Wiener process and $\boldsymbol{f}(\cdot, t) : \mathbb{R}^d \to \mathbb{R}^d$ and $g(\cdot) : \mathbb{R} \to \mathbb{R}$ are the drift and diffusion coefficients, respectively, where $d$ is the dimensionality of the dataset. These coefficients are selected differently for the variance preserving (VP) and variance exploding (VE) formulations, and $\boldsymbol{f}(\cdot)$ is always of the form $\boldsymbol{f}(\boldsymbol{x}, t) = f(t)\,\boldsymbol{x}$, where $f(\cdot) : \mathbb{R} \to \mathbb{R}$. Thus, the SDE can be equivalently written as

$$\mathrm{d}\boldsymbol{x} = f(t)\,\boldsymbol{x}\,\mathrm{d}t + g(t)\,\mathrm{d}\omega_t. \tag{10}$$

The perturbation kernels of this SDE (Eq. 29 in [15]) have the general form

$$p_{0t}\big(\boldsymbol{x}(t) \mid \boldsymbol{x}(0)\big) = \mathcal{N}\big(\boldsymbol{x}(t);\ s(t)\,\boldsymbol{x}(0),\ s(t)^2\,\sigma(t)^2\,\mathbf{I}\big), \tag{11}$$

where $\mathcal{N}(\boldsymbol{x}; \boldsymbol{\mu}, \boldsymbol{\Sigma})$ denotes the probability density function of $\mathcal{N}(\boldsymbol{\mu}, \boldsymbol{\Sigma})$ evaluated at $\boldsymbol{x}$,

$$s(t) = \exp\left(\int_0^t f(\xi)\,\mathrm{d}\xi\right), \quad \text{and} \quad \sigma(t) = \sqrt{\int_0^t \frac{g(\xi)^2}{s(\xi)^2}\,\mathrm{d}\xi}. \tag{12}$$

The marginal distribution $p_t(\boldsymbol{x})$ is obtained by integrating the perturbation kernels over $\boldsymbol{x}(0)$:

$$p_t(\boldsymbol{x}) = \int_{\mathbb{R}^d} p_{0t}(\boldsymbol{x} \mid \boldsymbol{x}_0)\,p_{\mathrm{data}}(\boldsymbol{x}_0)\,\mathrm{d}\boldsymbol{x}_0. \tag{13}$$

Song et al. [15] define the probability flow ODE (Eq. 13 in [15]) so that it obeys this same $p_t(\boldsymbol{x})$:

$$\mathrm{d}\boldsymbol{x} = \left[f(t)\,\boldsymbol{x} - \tfrac{1}{2}\,g(t)^2\,\nabla_{\boldsymbol{x}} \log p_t(\boldsymbol{x})\right]\,\mathrm{d}t. \tag{14}$$

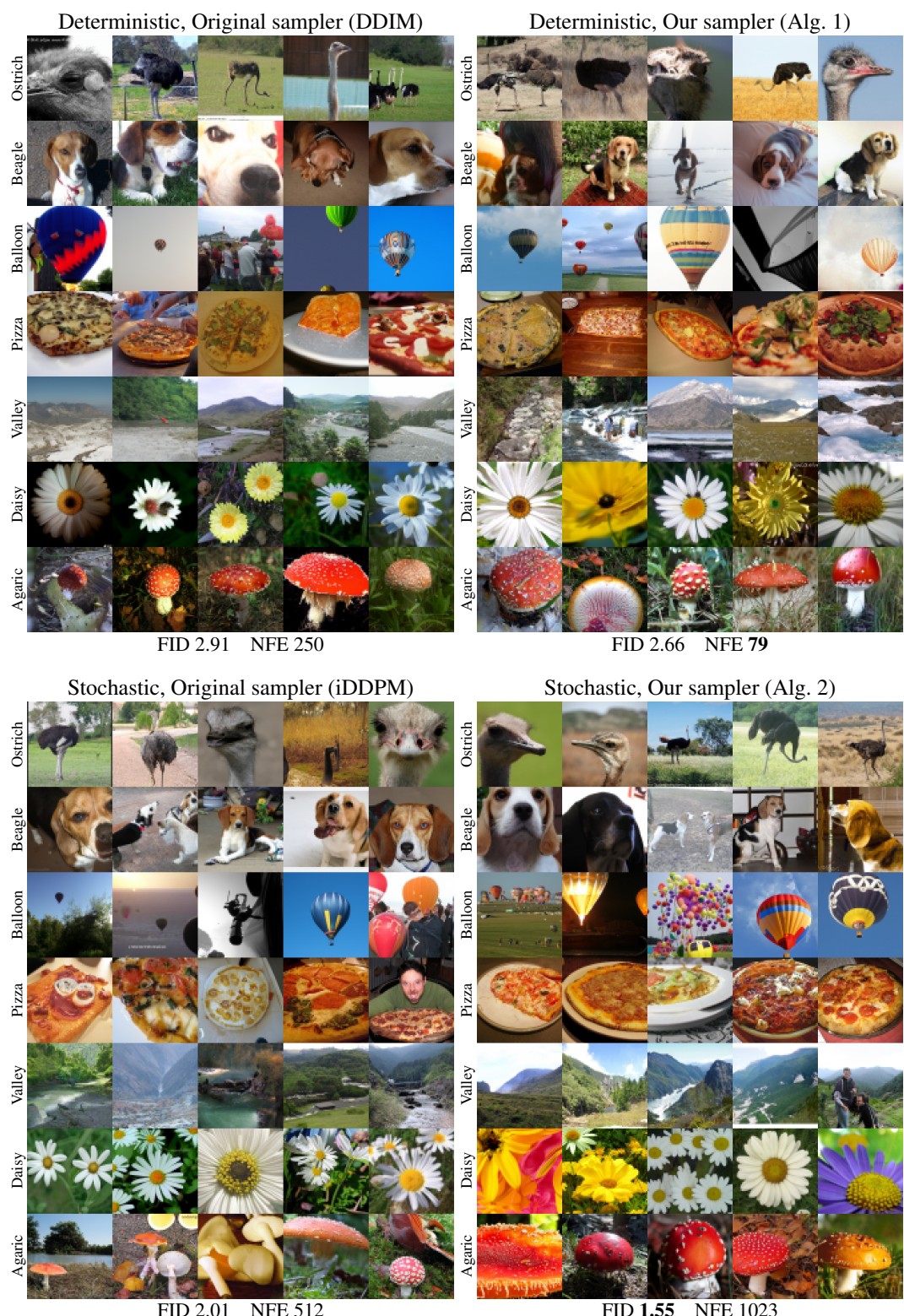

Figure 6: Results for different samplers on class-conditional ImageNet [3] at 64×64 resolution, using the pre-trained model by Dhariwal and Nichol [4]. The cases correspond to dots in Figures 2c and 4c.

Deterministic, Our sampler & training configuration

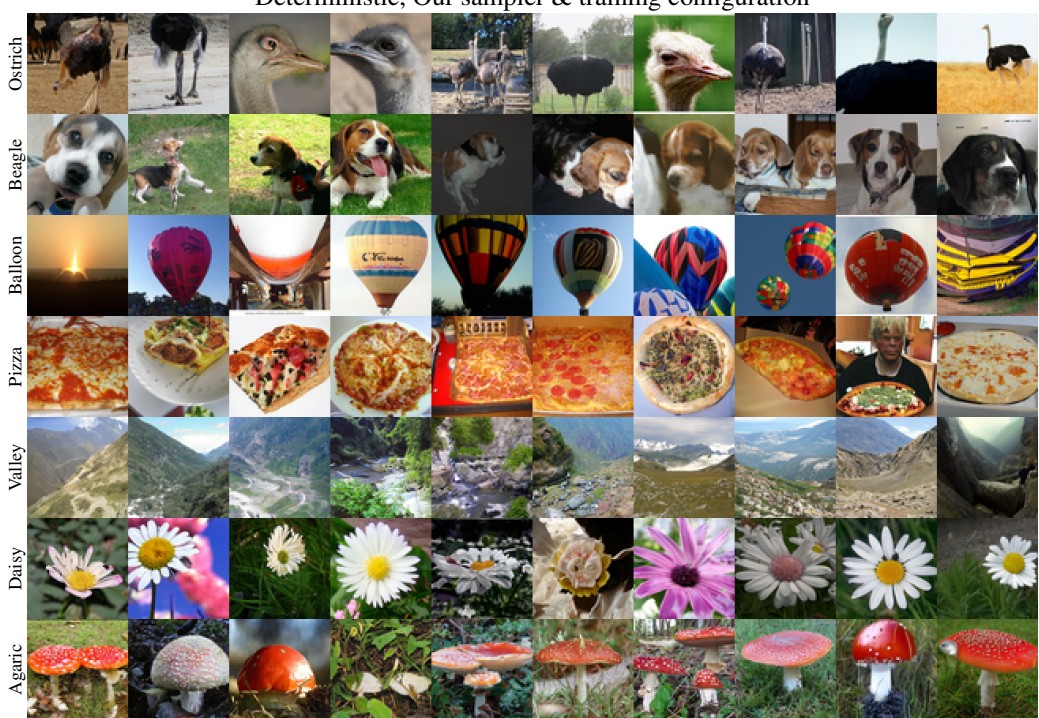

FID 2.23    NFE **79**

Stochastic, Our sampler & training configuration

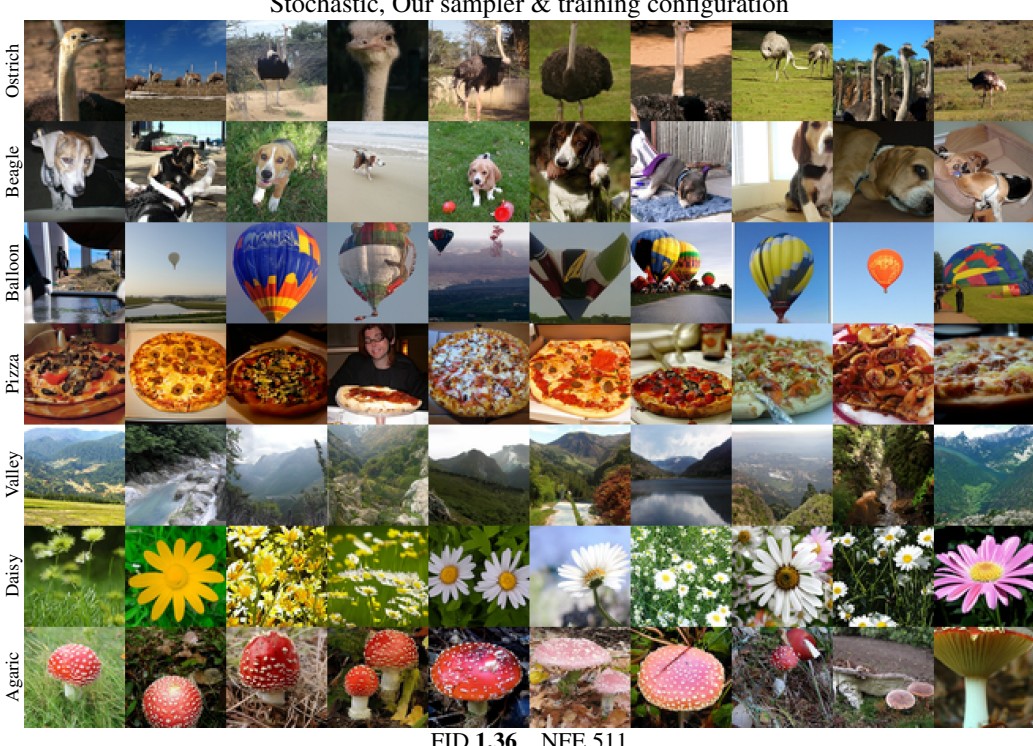

FID **1.36**    NFE 511

Figure 7: Results for our training configuration on class-conditional ImageNet [3] at 64×64 resolution, using our deterministic and stochastic samplers.

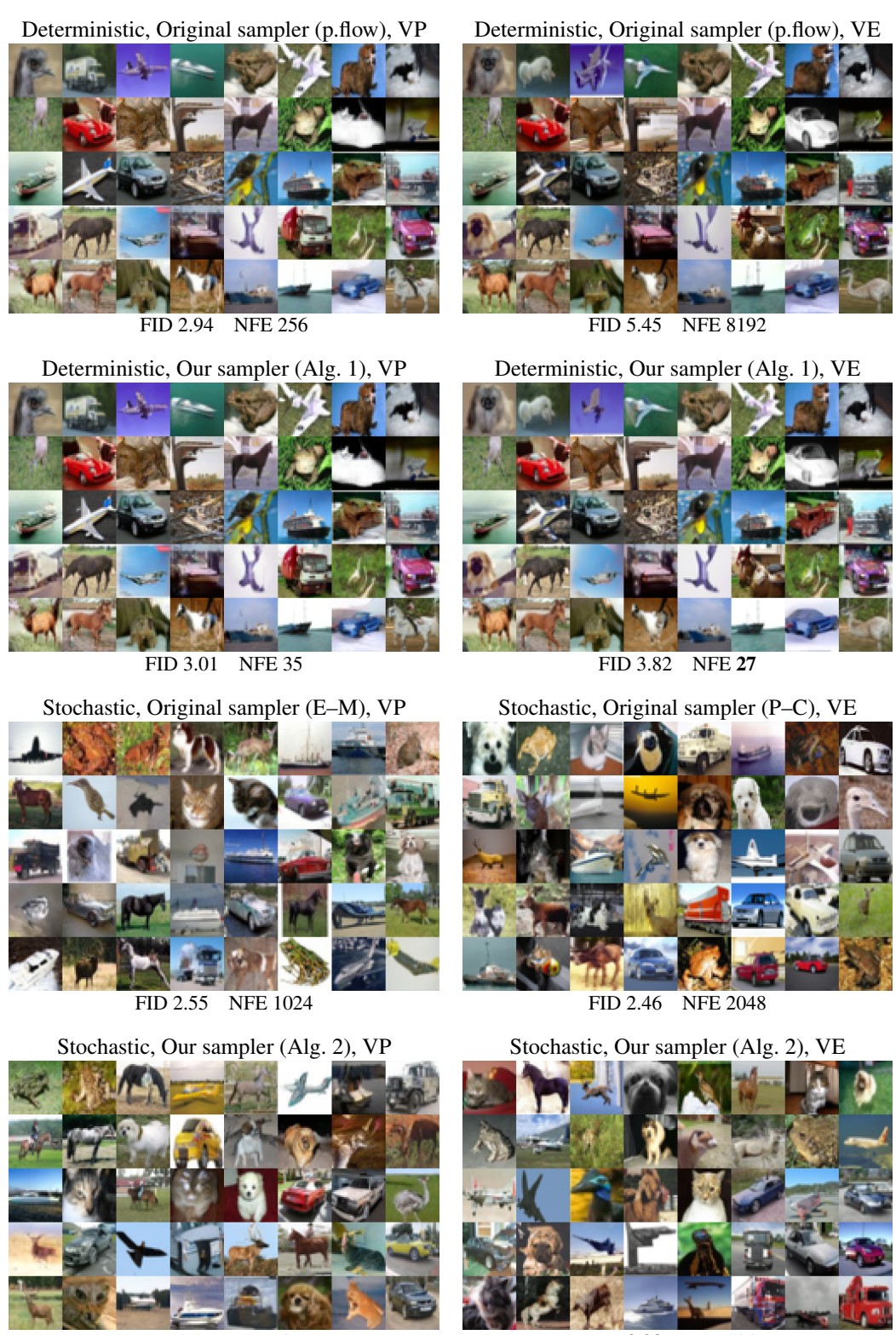

Figure 8: Results for different samplers on unconditional CIFAR-10 [12] at 32×32 resolution, using the pre-trained models by Song et al. [15]. The cases correspond to dots in Figures 2a,b and 4a,b.

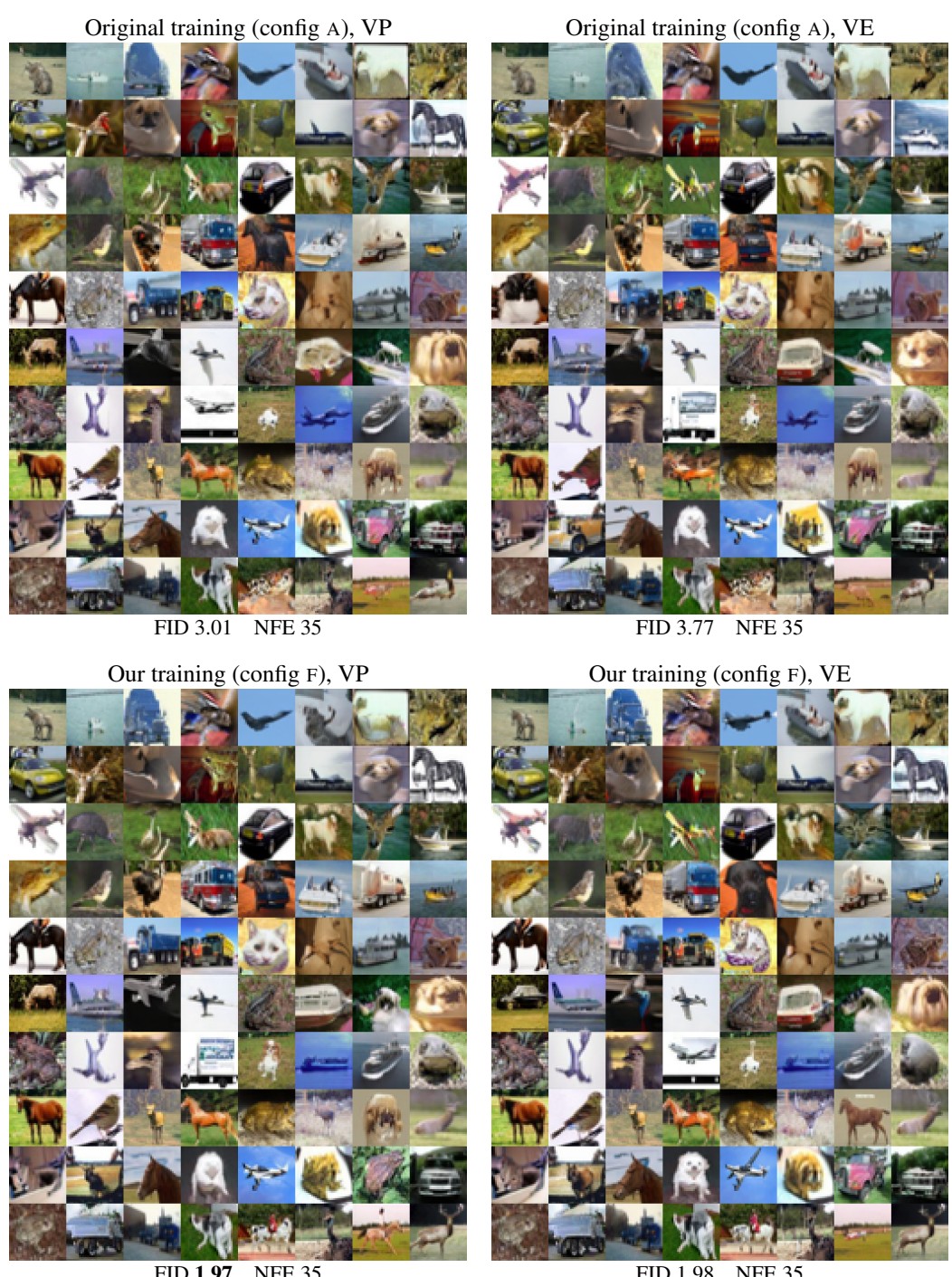

Figure 9: Results for different training configurations on unconditional CIFAR-10 [12] at 32×32 resolution, using our deterministic sampler with the same set of latent codes ($x_0$) in each case.

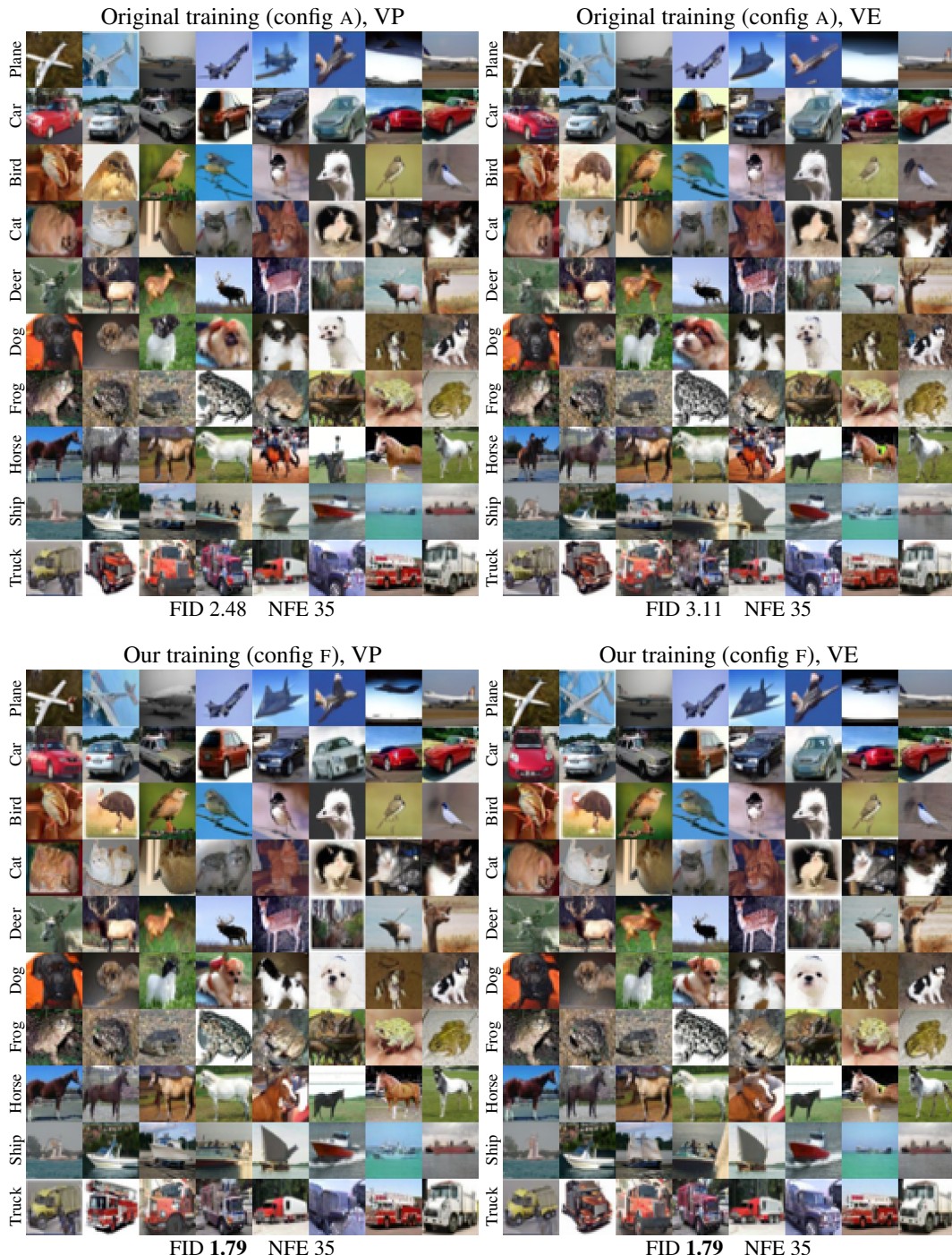

Figure 10: Results for different training configurations on class-conditional CIFAR-10 [12] at 32×32 resolution, using our deterministic sampler with the same set of latent codes ($x_0$) in each case.

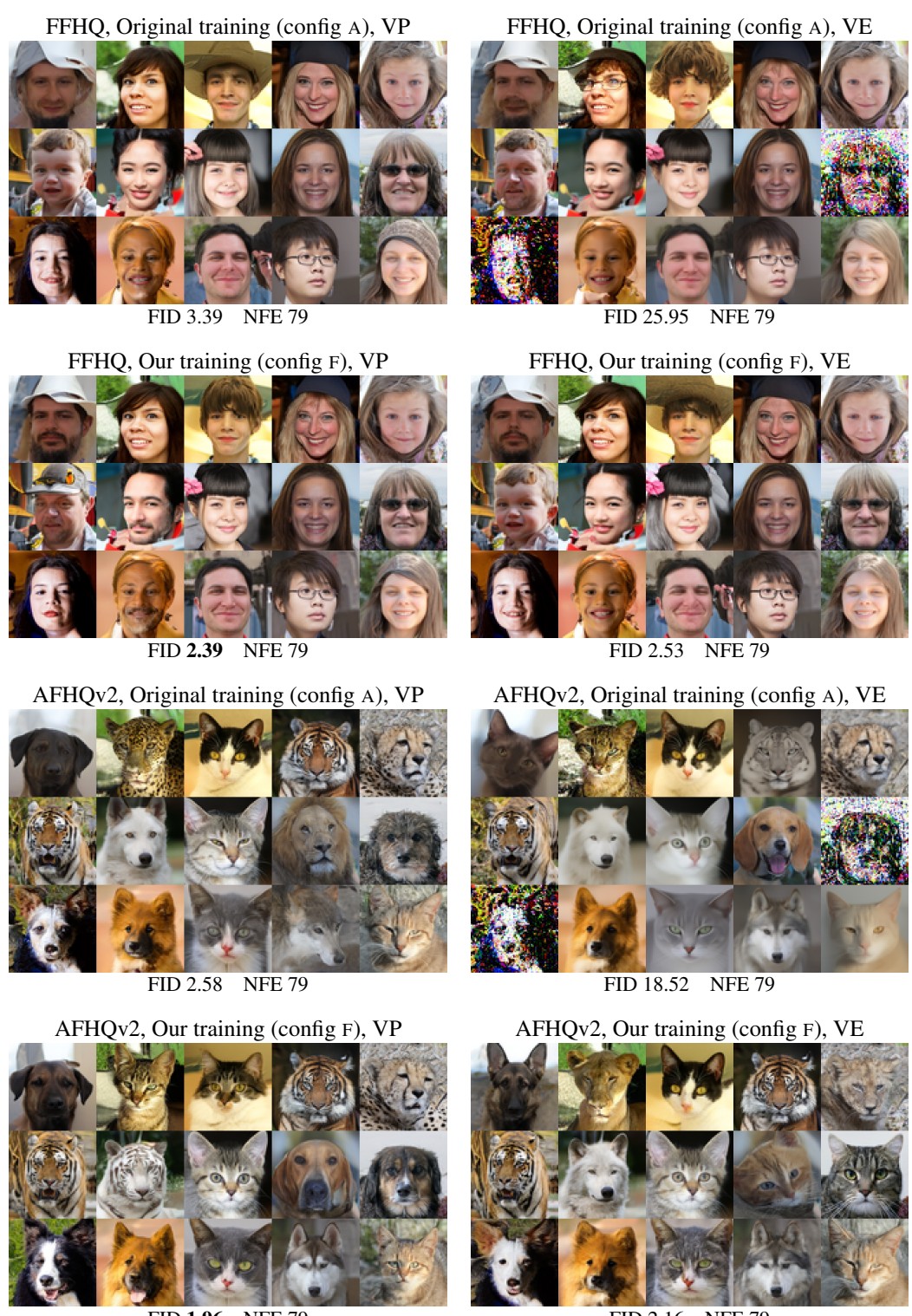

Figure 11: Results for different training configurations on FFHQ [11] and AFHQv2 [2] at 64×64 resolution, using our deterministic sampler with the same set of latent codes ($x_0$) in each case.

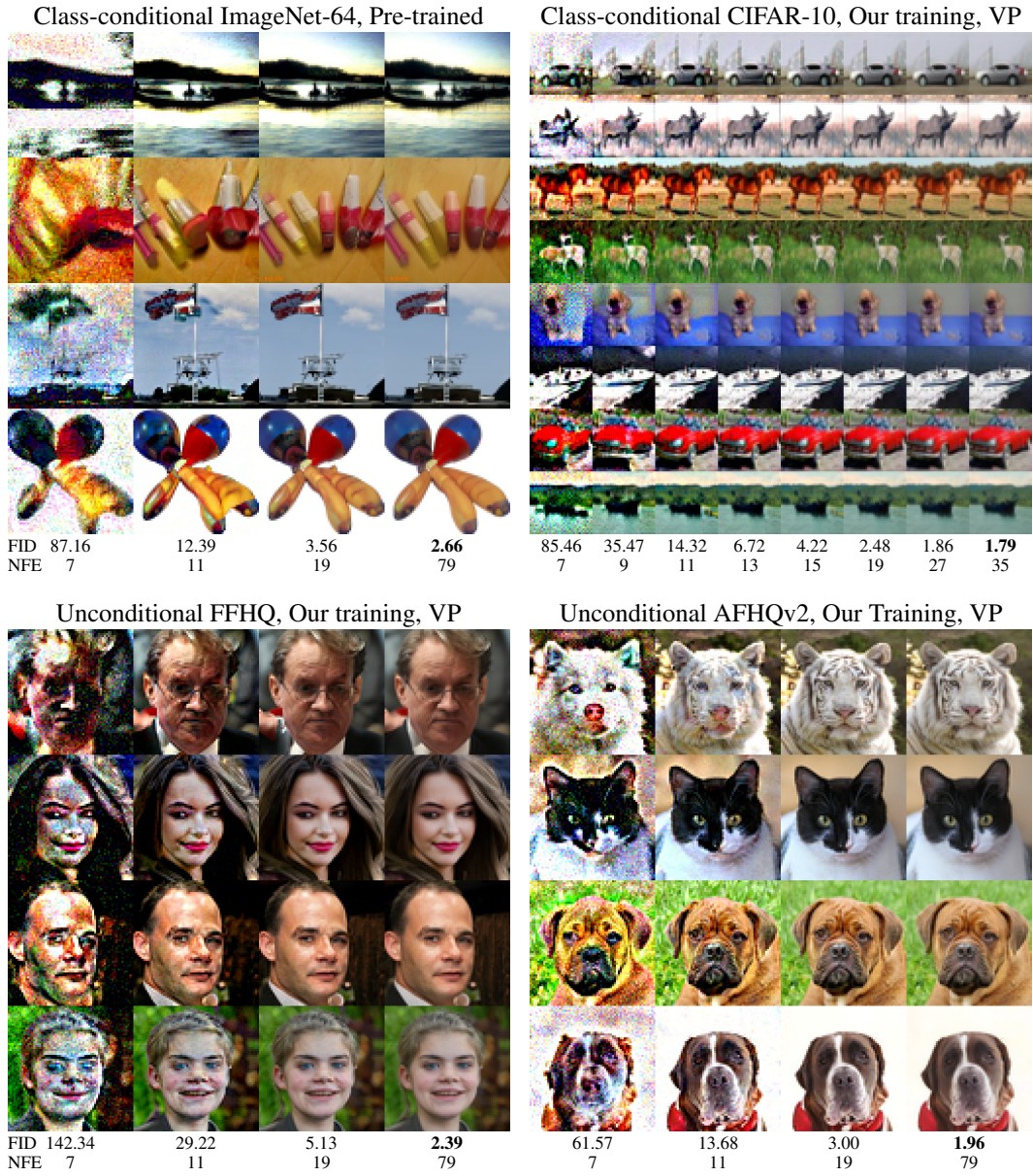

Figure 12: Image quality and FID as a function of NFE using our deterministic sampler. At 32×32 resolution, reasonable image quality is reached around NFE = 13, but FID keeps improving until NFE = 35. At 64×64 resolution, reasonable image quality is reached around NFE = 19, but FID keeps improving until NFE = 79.

Table 3: Evaluation of our improvements to deterministic sampling. The values correspond to the curves shown in Figure 2. We summarize each curve with two key values: the lowest observed FID for any NFE ("FID"), and the lowest NFE whose FID is within 3% of the lowest FID ("NFE"). The values marked with "–" are identical to the ones above them, because our sampler uses the same $\sigma(t)$ and $s(t)$ as DDIM.

| | Unconditional CIFAR-10 at 32×32 | | | | Class-conditional ImageNet-64 | |
| | VP | | VE | | | |
| **Sampling method** | FID ↓ | NFE ↓ | FID ↓ | NFE ↓ | FID ↓ | NFE ↓ |
|---|---|---|---|---|---|---|
| Original sampler [15, 4] | 2.85 | 256 | 5.45 | 8192 | 2.85 | 250 |
| Our Algorithm 1 | **2.79** | 512 | 4.78 | 8192 | 2.73 | 384 |
| + Heun & our $t_i$ | 2.88 | 255 | 4.23 | 191 | **2.64** | **79** |
| + Our $\sigma(t)$ & $s(t)$ | 2.93 | **35** | 3.73 | **27** | – | – |
| Black-box RK45 | 2.94 | 115 | **3.69** | 93 | 2.66 | 131 |

Table 4: Evaluation and ablations of our improvements to stochastic sampling. The values correspond to the curves shown in Figure 4.

| | Unconditional CIFAR-10 at 32×32 | | | | Class-conditional ImageNet-64 | |
| | VP | | VE | | | |
| **Sampling method** | FID ↓ | NFE ↓ | FID ↓ | NFE ↓ | FID ↓ | NFE ↓ |
|---|---|---|---|---|---|---|
| Deterministic baseline (Alg. 1) | 2.93 | **35** | 3.73 | **27** | 2.64 | **79** |
| Alg. 2, $S_{\text{tmin,tmax}} = [0, \infty]$, $S_{\text{noise}} = 1$ | 2.69 | 95 | 2.97 | 383 | 1.86 | 383 |
| Alg. 2, $S_{\text{tmin,tmax}} = [0, \infty]$ | 2.54 | 127 | 2.51 | 511 | 1.63 | 767 |
| Alg. 2, $S_{\text{noise}} = 1$ | 2.52 | 95 | 2.84 | 191 | 1.84 | 255 |
| Alg. 2, Optimal settings | **2.27** | 383 | **2.23** | 767 | **1.55** | 511 |
| Previous work [15, 4] | 2.55 | 768 | 2.46 | 1024 | 2.01 | 384 |

## B.2 Our ODE formulation (Eq. 1 and Eq. 4)

The original ODE formulation (Eq. 14) is built around the functions $f$ and $g$ that correspond directly to specific terms that appear in the formula; the properties of the marginal distribution (Eq. 12) can only be derived indirectly based on these functions. However, $f$ and $g$ are of little practical interest in themselves, whereas the marginal distributions are of utmost importance in terms of training the model in the first place, bootstrapping the sampling process, and understanding how the ODE behaves in practice. Given that the idea of the probability flow ODE is to match a particular set of marginal distributions, it makes sense to treat the marginal distributions as first-class citizens and define the ODE directly based on $\sigma(t)$ and $s(t)$, eliminating the need for $f(t)$ and $g(t)$.

Let us start by expressing the marginal distribution of Eq. 13 in closed form:

$$
p_t(\boldsymbol{x}) = \int_{\mathbb{R}^d} p_{0t}(\boldsymbol{x} \mid \boldsymbol{x}_0) \, p_{\text{data}}(\boldsymbol{x}_0) \, \mathrm{d}\boldsymbol{x}_0 \tag{15}
$$

$$
= \int_{\mathbb{R}^d} p_{\text{data}}(\boldsymbol{x}_0) \left[ \mathcal{N}\big(\boldsymbol{x}; \, s(t) \, \boldsymbol{x}_0, \, s(t)^2 \, \sigma(t)^2 \, \mathbf{I}\big) \right] \mathrm{d}\boldsymbol{x}_0 \tag{16}
$$

$$
= \int_{\mathbb{R}^d} p_{\text{data}}(\boldsymbol{x}_0) \left[ s(t)^{-d} \, \mathcal{N}\big(\boldsymbol{x}/s(t); \, \boldsymbol{x}_0, \, \sigma(t)^2 \, \mathbf{I}\big) \right] \mathrm{d}\boldsymbol{x}_0 \tag{17}
$$

$$
= s(t)^{-d} \int_{\mathbb{R}^d} p_{\text{data}}(\boldsymbol{x}_0) \, \mathcal{N}\big(\boldsymbol{x}/s(t); \, \boldsymbol{x}_0, \, \sigma(t)^2 \, \mathbf{I}\big) \, \mathrm{d}\boldsymbol{x}_0 \tag{18}
$$

$$
= s(t)^{-d} \left[ p_{\text{data}} * \mathcal{N}\big(\mathbf{0}, \, \sigma(t)^2 \, \mathbf{I}\big) \right] \big(\boldsymbol{x}/s(t)\big), \tag{19}
$$

where $p_a * p_b$ denotes the convolution of probability density functions $p_a$ and $p_b$. The expression inside the brackets corresponds to a mollified version of $p_{\text{data}}$ obtained by adding i.i.d. Gaussian noise to the samples. Let us denote this distribution by $p(\boldsymbol{x}; \sigma)$:

$$
p(\boldsymbol{x}; \sigma) = p_{\text{data}} * \mathcal{N}\big(\mathbf{0}, \, \sigma(t)^2 \, \mathbf{I}\big) \quad \text{and} \quad p_t(\boldsymbol{x}) = s(t)^{-d} \, p\big(\boldsymbol{x}/s(t); \sigma(t)\big). \tag{20}
$$

We can now express the probability flow ODE (Eq. 14) using $p(\boldsymbol{x}; \sigma)$ instead of $p_t(\boldsymbol{x})$:

$$\mathrm{d}\boldsymbol{x} = \left[ f(t)\boldsymbol{x} - \tfrac{1}{2} g(t)^2 \, \nabla_{\boldsymbol{x}} \log \left[ p_t(\boldsymbol{x}) \right] \right] \, \mathrm{d}t \tag{21}$$

$$= \left[ f(t)\boldsymbol{x} - \tfrac{1}{2} g(t)^2 \, \nabla_{\boldsymbol{x}} \log \left[ s(t)^{-d} \, p\big(\boldsymbol{x}/s(t); \sigma(t)\big) \right] \right] \, \mathrm{d}t \tag{22}$$

$$= \left[ f(t)\boldsymbol{x} - \tfrac{1}{2} g(t)^2 \left[ \nabla_{\boldsymbol{x}} \log s(t)^{-d} + \nabla_{\boldsymbol{x}} \log p\big(\boldsymbol{x}/s(t); \sigma(t)\big) \right] \right] \, \mathrm{d}t \tag{23}$$

$$= \left[ f(t)\boldsymbol{x} - \tfrac{1}{2} g(t)^2 \, \nabla_{\boldsymbol{x}} \log p\big(\boldsymbol{x}/s(t); \sigma(t)\big) \right] \, \mathrm{d}t. \tag{24}$$

Next, let us rewrite $f(t)$ in terms of $s(t)$ based on Eq. 12:

$$\exp\left( \int_0^t f(\xi) \, \mathrm{d}\xi \right) = s(t) \tag{25}$$

$$\int_0^t f(\xi) \, \mathrm{d}\xi = \log s(t) \tag{26}$$

$$\mathrm{d}\left[ \int_0^t f(\xi) \, \mathrm{d}\xi \right] / \mathrm{d}t = \mathrm{d}\left[ \log s(t) \right] / \mathrm{d}t \tag{27}$$

$$f(t) = \dot{s}(t)/s(t). \tag{28}$$

Similarly, we can also rewrite $g(t)$ in terms of $\sigma(t)$:

$$\sqrt{\int_0^t \frac{g(\xi)^2}{s(\xi)^2} \, \mathrm{d}\xi} = \sigma(t) \tag{29}$$

$$\int_0^t \frac{g(\xi)^2}{s(\xi)^2} \, \mathrm{d}\xi = \sigma(t)^2 \tag{30}$$

$$\mathrm{d}\left[ \int_0^t \frac{g(\xi)^2}{s(\xi)^2} \, \mathrm{d}\xi \right] / \mathrm{d}t = \mathrm{d}\left[ \sigma(t)^2 \right] / \mathrm{d}t \tag{31}$$

$$g(t)^2/s(t)^2 = 2 \, \dot{\sigma}(t) \, \sigma(t) \tag{32}$$

$$g(t)/s(t) = \sqrt{2 \, \dot{\sigma}(t) \, \sigma(t)} \tag{33}$$

$$g(t) = s(t) \sqrt{2 \, \dot{\sigma}(t) \, \sigma(t)}. \tag{34}$$

Finally, substitute $f$ (Eq. 28) and $g$ (Eq. 34) into the ODE of Eq. 24:

$$\mathrm{d}\boldsymbol{x} = \left[ [f(t)] \, \boldsymbol{x} - \tfrac{1}{2} [g(t)]^2 \, \nabla_{\boldsymbol{x}} \log p\big(\boldsymbol{x}/s(t); \sigma(t)\big) \right] \, \mathrm{d}t \tag{35}$$

$$= \left[ \left[ \dot{s}(t)/s(t) \right] \boldsymbol{x} - \tfrac{1}{2} \left[ s(t) \sqrt{2 \, \dot{\sigma}(t) \, \sigma(t)} \right]^2 \nabla_{\boldsymbol{x}} \log p\big(\boldsymbol{x}/s(t); \sigma(t)\big) \right] \, \mathrm{d}t \tag{36}$$

$$= \left[ \left[ \dot{s}(t)/s(t) \right] \boldsymbol{x} - \tfrac{1}{2} \left[ 2 \, s(t)^2 \, \dot{\sigma}(t) \, \sigma(t) \right] \nabla_{\boldsymbol{x}} \log p\big(\boldsymbol{x}/s(t); \sigma(t)\big) \right] \, \mathrm{d}t \tag{37}$$

$$= \left[ \frac{\dot{s}(t)}{s(t)} \, \boldsymbol{x} - s(t)^2 \, \dot{\sigma}(t) \, \sigma(t) \, \nabla_{\boldsymbol{x}} \log p\left( \frac{\boldsymbol{x}}{s(t)}; \sigma(t) \right) \right] \, \mathrm{d}t. \tag{38}$$

Thus we have obtained Eq. 4 in the main paper, and Eq. 1 is recovered by setting $s(t) = 1$:

$$\mathrm{d}\boldsymbol{x} = -\dot{\sigma}(t) \, \sigma(t) \, \nabla_{\boldsymbol{x}} \log p\big(\boldsymbol{x}; \sigma(t)\big) \, \mathrm{d}t. \tag{39}$$

Our formulation (Eq. 4) highlights the fact that every realization of the probability flow ODE is simply a reparameterization of the same canonical ODE; changing $\sigma(t)$ corresponds to reparameterizing $t$, whereas changing $s(t)$ corresponds to reparameterizing $\boldsymbol{x}$.

## B.3  Denoising score matching (Eq. 2 and Eq. 3)

For the sake of completeness, we derive the connection between score matching and denoising for a finite dataset. For a more general treatment and further background on the topic, see Hyvärinen [7] and Vincent [19].

Let us assume that our training set consists of a finite number of samples $\{\boldsymbol{y}_1, \ldots, \boldsymbol{y}_Y\}$. This implies $p_{\text{data}}(\boldsymbol{x})$ is represented by a mixture of Dirac delta distributions:

$$p_{\text{data}}(\boldsymbol{x}) = \frac{1}{Y} \sum_{i=1}^{Y} \delta(\boldsymbol{x} - \boldsymbol{y}_i), \tag{40}$$

which allows us to also express $p(\boldsymbol{x}; \sigma)$ in closed form based on Eq. 20:

$$\begin{align}
p(\boldsymbol{x}; \sigma) &= p_{\text{data}} * \mathcal{N}(\boldsymbol{0}, \sigma(t)^2 \, \mathbf{I}) \tag{41} \\
&= \int_{\mathbb{R}^d} p_{\text{data}}(\boldsymbol{x}_0) \, \mathcal{N}(\boldsymbol{x}; \, \boldsymbol{x}_0, \, \sigma^2 \, \mathbf{I}) \, \mathrm{d}\boldsymbol{x}_0 \tag{42} \\
&= \int_{\mathbb{R}^d} \left[ \frac{1}{Y} \sum_{i=1}^{Y} \delta(\boldsymbol{x}_0 - \boldsymbol{y}_i) \right] \mathcal{N}(\boldsymbol{x}; \, \boldsymbol{x}_0, \, \sigma^2 \, \mathbf{I}) \, \mathrm{d}\boldsymbol{x}_0 \tag{43} \\
&= \frac{1}{Y} \sum_{i=1}^{Y} \int_{\mathbb{R}^d} \mathcal{N}(\boldsymbol{x}; \, \boldsymbol{x}_0, \, \sigma^2 \, \mathbf{I}) \, \delta(\boldsymbol{x}_0 - \boldsymbol{y}_i) \, \mathrm{d}\boldsymbol{x}_0 \tag{44} \\
&= \frac{1}{Y} \sum_{i=1}^{Y} \mathcal{N}(\boldsymbol{x}; \, \boldsymbol{y}_i, \, \sigma^2 \, \mathbf{I}). \tag{45}
\end{align}$$

Let us now consider the denoising score matching loss of Eq. 2. By expanding the expectations, we can rewrite the formula as an integral over the noisy samples $\boldsymbol{x}$:

$$\begin{align}
\mathcal{L}(D; \sigma) &= \mathbb{E}_{\boldsymbol{y} \sim p_{\text{data}}} \mathbb{E}_{\boldsymbol{n} \sim \mathcal{N}(\boldsymbol{0}, \sigma^2 \mathbf{I})} \left\| D(\boldsymbol{y} + \boldsymbol{n}; \sigma) - \boldsymbol{y} \right\|_2^2 \tag{46} \\
&= \mathbb{E}_{\boldsymbol{y} \sim p_{\text{data}}} \mathbb{E}_{\boldsymbol{x} \sim \mathcal{N}(\boldsymbol{y}, \sigma^2 \mathbf{I})} \left\| D(\boldsymbol{x}; \sigma) - \boldsymbol{y} \right\|_2^2 \tag{47} \\
&= \mathbb{E}_{\boldsymbol{y} \sim p_{\text{data}}} \int_{\mathbb{R}^d} \mathcal{N}(\boldsymbol{x}; \, \boldsymbol{y}, \, \sigma^2 \, \mathbf{I}) \left\| D(\boldsymbol{x}; \sigma) - \boldsymbol{y} \right\|_2^2 \, \mathrm{d}\boldsymbol{x} \tag{48} \\
&= \frac{1}{Y} \sum_{i=1}^{Y} \int_{\mathbb{R}^d} \mathcal{N}(\boldsymbol{x}; \, \boldsymbol{y}_i, \, \sigma^2 \, \mathbf{I}) \left\| D(\boldsymbol{x}; \sigma) - \boldsymbol{y}_i \right\|_2^2 \, \mathrm{d}\boldsymbol{x} \tag{49} \\
&= \int_{\mathbb{R}^d} \underbrace{\frac{1}{Y} \sum_{i=1}^{Y} \mathcal{N}(\boldsymbol{x}; \, \boldsymbol{y}_i, \, \sigma^2 \, \mathbf{I}) \left\| D(\boldsymbol{x}; \sigma) - \boldsymbol{y}_i \right\|_2^2}_{=: \, \mathcal{L}(D; \boldsymbol{x}, \sigma)} \, \mathrm{d}\boldsymbol{x}. \tag{50}
\end{align}$$

Eq. 50 means that we can minimize $\mathcal{L}(D; \sigma)$ by minimizing $\mathcal{L}(D; \boldsymbol{x}, \sigma)$ independently for each $\boldsymbol{x}$:

$$D(\boldsymbol{x}; \sigma) = \arg\min_{D(\boldsymbol{x}; \sigma)} \mathcal{L}(D; \boldsymbol{x}, \sigma). \tag{51}$$

This is a convex optimization problem; its solution is uniquely identified by setting the gradient w.r.t. $D(\boldsymbol{x}; \sigma)$ to zero:

$$\begin{align}
\boldsymbol{0} &= \nabla_{D(\boldsymbol{x}; \sigma)} \left[ \mathcal{L}(D; \boldsymbol{x}, \sigma) \right] \tag{52} \\
\boldsymbol{0} &= \nabla_{D(\boldsymbol{x}; \sigma)} \left[ \frac{1}{Y} \sum_{i=1}^{Y} \mathcal{N}(\boldsymbol{x}; \, \boldsymbol{y}_i, \, \sigma^2 \, \mathbf{I}) \left\| D(\boldsymbol{x}; \sigma) - \boldsymbol{y}_i \right\|_2^2 \right] \tag{53} \\
\boldsymbol{0} &= \sum_{i=1}^{Y} \mathcal{N}(\boldsymbol{x}; \, \boldsymbol{y}_i, \, \sigma^2 \, \mathbf{I}) \, \nabla_{D(\boldsymbol{x}; \sigma)} \left[ \left\| D(\boldsymbol{x}; \sigma) - \boldsymbol{y}_i \right\|_2^2 \right] \tag{54} \\
\boldsymbol{0} &= \sum_{i=1}^{Y} \mathcal{N}(\boldsymbol{x}; \, \boldsymbol{y}_i, \, \sigma^2 \, \mathbf{I}) \left[ 2 \, D(\boldsymbol{x}; \sigma) - 2 \, \boldsymbol{y}_i \right] \tag{55} \\
\boldsymbol{0} &= \left[ \sum_{i=1}^{Y} \mathcal{N}(\boldsymbol{x}; \, \boldsymbol{y}_i, \, \sigma^2 \, \mathbf{I}) \right] D(\boldsymbol{x}; \sigma) - \sum_{i=1}^{Y} \mathcal{N}(\boldsymbol{x}; \, \boldsymbol{y}_i, \, \sigma^2 \, \mathbf{I}) \, \boldsymbol{y}_i \tag{56} \\
D(\boldsymbol{x}; \sigma) &= \frac{\sum_i \mathcal{N}(\boldsymbol{x}; \, \boldsymbol{y}_i, \, \sigma^2 \, \mathbf{I}) \, \boldsymbol{y}_i}{\sum_i \mathcal{N}(\boldsymbol{x}; \, \boldsymbol{y}_i, \, \sigma^2 \, \mathbf{I})}, \tag{57}
\end{align}$$

which gives a closed-form solution for the ideal denoiser $D(\boldsymbol{x}; \sigma)$. Note that Eq. 57 is feasible to compute in practice for small datasets — we show the results for CIFAR-10 in Figure 1b.

Next, let us consider the score of the distribution $p(\boldsymbol{x}; \sigma)$ defined in Eq. 45:

$$\nabla_{\boldsymbol{x}} \log p(\boldsymbol{x}; \sigma) \quad = \quad \frac{\nabla_{\boldsymbol{x}} p(\boldsymbol{x}; \sigma)}{p(\boldsymbol{x}; \sigma)} \tag{58}$$

$$= \quad \frac{\nabla_{\boldsymbol{x}} \left[ \frac{1}{Y} \sum_i \mathcal{N}\big(\boldsymbol{x}; \, \boldsymbol{y}_i, \, \sigma^2 \, \mathbf{I}\big) \right]}{\left[ \frac{1}{Y} \sum_i \mathcal{N}\big(\boldsymbol{x}; \, \boldsymbol{y}_i, \, \sigma^2 \, \mathbf{I}\big) \right]} \tag{59}$$

$$= \quad \frac{\sum_i \nabla_{\boldsymbol{x}} \mathcal{N}\big(\boldsymbol{x}; \, \boldsymbol{y}_i, \, \sigma^2 \, \mathbf{I}\big)}{\sum_i \mathcal{N}\big(\boldsymbol{x}; \, \boldsymbol{y}_i, \, \sigma^2 \, \mathbf{I}\big)}. \tag{60}$$

We can simplify the numerator of Eq. 60 further:

$$\nabla_{\boldsymbol{x}} \mathcal{N}\big(\boldsymbol{x}; \, \boldsymbol{y}_i, \, \sigma^2 \, \mathbf{I}\big) \quad = \quad \nabla_{\boldsymbol{x}} \left[ \big(2\pi\sigma^2\big)^{-\frac{d}{2}} \, \exp \frac{\|\boldsymbol{x} - \boldsymbol{y}_i\|_2^2}{-2\,\sigma^2} \right] \tag{61}$$

$$= \quad \big(2\pi\sigma^2\big)^{-\frac{d}{2}} \, \nabla_{\boldsymbol{x}} \left[ \exp \frac{\|\boldsymbol{x} - \boldsymbol{y}_i\|_2^2}{-2\,\sigma^2} \right] \tag{62}$$

$$= \quad \left[ \big(2\pi\sigma^2\big)^{-\frac{d}{2}} \exp \frac{\|\boldsymbol{x} - \boldsymbol{y}_i\|_2^2}{-2\,\sigma^2} \right] \nabla_{\boldsymbol{x}} \left[ \frac{\|\boldsymbol{x} - \boldsymbol{y}_i\|_2^2}{-2\,\sigma^2} \right] \tag{63}$$

$$= \quad \mathcal{N}\big(\boldsymbol{x}; \, \boldsymbol{y}_i, \, \sigma^2 \, \mathbf{I}\big) \, \nabla_{\boldsymbol{x}} \left[ \frac{\|\boldsymbol{x} - \boldsymbol{y}_i\|_2^2}{-2\,\sigma^2} \right] \tag{64}$$

$$= \quad \mathcal{N}\big(\boldsymbol{x}; \, \boldsymbol{y}_i, \, \sigma^2 \, \mathbf{I}\big) \left[ \frac{\boldsymbol{y}_i - \boldsymbol{x}}{\sigma^2} \right]. \tag{65}$$

Let us substitute the result back to Eq. 60:

$$\nabla_{\boldsymbol{x}} \log p(\boldsymbol{x}; \sigma) \quad = \quad \frac{\sum_i \nabla_{\boldsymbol{x}} \mathcal{N}\big(\boldsymbol{x}; \, \boldsymbol{y}_i, \, \sigma^2 \, \mathbf{I}\big)}{\sum_i \mathcal{N}\big(\boldsymbol{x}; \, \boldsymbol{y}_i, \, \sigma^2 \, \mathbf{I}\big)} \tag{66}$$

$$= \quad \frac{\sum_i \mathcal{N}\big(\boldsymbol{x}; \, \boldsymbol{y}_i, \, \sigma^2 \, \mathbf{I}\big) \left[ \frac{\boldsymbol{y}_i - \boldsymbol{x}}{\sigma^2} \right]}{\sum_i \mathcal{N}\big(\boldsymbol{x}; \, \boldsymbol{y}_i, \, \sigma^2 \, \mathbf{I}\big)} \tag{67}$$

$$= \quad \left( \frac{\sum_i \mathcal{N}\big(\boldsymbol{x}; \, \boldsymbol{y}_i, \, \sigma^2 \, \mathbf{I}\big) \boldsymbol{y}_i}{\sum_i \mathcal{N}\big(\boldsymbol{x}; \, \boldsymbol{y}_i, \, \sigma^2 \, \mathbf{I}\big)} - \boldsymbol{x} \right) / \sigma^2. \tag{68}$$

Notice that the fraction in Eq. 68 is identical to Eq. 57. We can thus equivalently write Eq. 68 as

$$\nabla_{\boldsymbol{x}} \log p(\boldsymbol{x}; \sigma) = \big(D(\boldsymbol{x}; \, \sigma) - \boldsymbol{x}\big) / \sigma^2, \tag{69}$$

which matches Eq. 3 in the main paper.

### B.4 Evaluating our ODE in practice (Algorithm 1)

Let us consider $\boldsymbol{x}$ to be a scaled version of an original, non-scaled variable $\hat{\boldsymbol{x}}$ and substitute $\boldsymbol{x} = s(t)\,\hat{\boldsymbol{x}}$ into the score term that appears in our scaled ODE (Eq. 4):

$$\nabla_{\boldsymbol{x}} \log p\big(\boldsymbol{x}/s(t); \sigma(t)\big) \tag{70}$$

$$= \quad \nabla_{[s(t)\hat{\boldsymbol{x}}]} \log p\big([s(t)\,\hat{\boldsymbol{x}}]/s(t); \sigma(t)\big) \tag{71}$$

$$= \quad \nabla_{s(t)\hat{\boldsymbol{x}}} \log p\big(\hat{\boldsymbol{x}}; \sigma(t)\big) \tag{72}$$

$$= \quad \tfrac{1}{s(t)} \nabla_{\hat{\boldsymbol{x}}} \log p\big(\hat{\boldsymbol{x}}; \sigma(t)\big). \tag{73}$$

We can further rewrite this with respect to $D(\cdot)$ using Eq. 3:

$$\nabla_{\boldsymbol{x}} \log p\big(\boldsymbol{x}/s(t); \sigma(t)\big) \;=\; \tfrac{1}{s(t)\sigma(t)^2}\Big(D\big(\hat{\boldsymbol{x}}; \sigma(t)\big) - \hat{\boldsymbol{x}}\Big). \tag{74}$$

Let us now substitute Eq. 74 into Eq. 4, approximating the ideal denoiser $D(\cdot)$ with our trained model $D_\theta(\cdot)$:

$$\mathrm{d}\boldsymbol{x} \;=\; \Big[\dot{s}(t)\,\boldsymbol{x}/s(t) - s(t)^2\,\dot{\sigma}(t)\,\sigma(t)\Big[\tfrac{1}{s(t)\sigma(t)^2}\big(D_\theta\big(\hat{\boldsymbol{x}}; \sigma(t)\big) - \hat{\boldsymbol{x}}\big)\Big]\Big]\;\mathrm{d}t \tag{75}$$

$$=\; \Big[\tfrac{\dot{s}(t)}{s(t)}\,\boldsymbol{x} - \tfrac{\dot{\sigma}(t)s(t)}{\sigma(t)}\big(D_\theta\big(\hat{\boldsymbol{x}}; \sigma(t)\big) - \hat{\boldsymbol{x}}\big)\Big]\;\mathrm{d}t. \tag{76}$$

Finally, backsubstitute $\hat{\boldsymbol{x}} = \boldsymbol{x}/s(t)$:

$$\mathrm{d}\boldsymbol{x} \;=\; \Big[\tfrac{\dot{s}(t)}{s(t)}\,\boldsymbol{x} - \tfrac{\dot{\sigma}(t)s(t)}{\sigma(t)}\big(D_\theta\big([\hat{\boldsymbol{x}}]; \sigma(t)\big) - [\hat{\boldsymbol{x}}]\big)\Big]\;\mathrm{d}t \tag{77}$$

$$=\; \Big[\tfrac{\dot{s}(t)}{s(t)}\,\boldsymbol{x} - \tfrac{\dot{\sigma}(t)s(t)}{\sigma(t)}\big(D_\theta\big([\boldsymbol{x}/s(t)]; \sigma(t)\big) - [\boldsymbol{x}/s(t)]\big)\Big]\;\mathrm{d}t \tag{78}$$

$$=\; \Big[\tfrac{\dot{s}(t)}{s(t)}\,\boldsymbol{x} - \tfrac{\dot{\sigma}(t)s(t)}{\sigma(t)}D_\theta\big(\boldsymbol{x}/s(t); \sigma(t)\big) + \tfrac{\dot{\sigma}(t)}{\sigma(t)}\,\boldsymbol{x}\Big]\;\mathrm{d}t \tag{79}$$

$$=\; \Big[\Big(\tfrac{\dot{\sigma}(t)}{\sigma(t)} + \tfrac{\dot{s}(t)}{s(t)}\Big)\boldsymbol{x} - \tfrac{\dot{\sigma}(t)s(t)}{\sigma(t)}D_\theta\big(\boldsymbol{x}/s(t); \sigma(t)\big)\Big]\;\mathrm{d}t. \tag{80}$$

We can equivalenty write Eq. 80 as

$$\mathrm{d}\boldsymbol{x}/\mathrm{d}t = \Big(\frac{\dot{\sigma}(t)}{\sigma(t)} + \frac{\dot{s}(t)}{s(t)}\Big)\boldsymbol{x} - \frac{\dot{\sigma}(t)s(t)}{\sigma(t)}D_\theta\Big(\frac{\boldsymbol{x}}{s(t)}; \sigma(t)\Big), \tag{81}$$

matching lines 4 and 7 of Algorithm 1.

### B.5 Our SDE formulation (Eq. 6)

We derive the SDE of Eq. 6 by the following strategy:

- The desired marginal densities $p\big(\boldsymbol{x}; \sigma(t)\big)$ are convolutions of the data density $p_{\mathrm{data}}$ and an isotropic Gaussian density with standard deviation $\sigma(t)$ (see Eq. 20). Hence, considered as a function of the time $t$, the density evolves according to a heat diffusion PDE with time-varying diffusivity. As a first step, we find this PDE.

- We then use the Fokker–Planck equation to recover a family of SDEs for which the density evolves according to this PDE. Eq. 6 is obtained from a suitable parametrization of this family.

#### B.5.1 Generating the marginals by heat diffusion

We consider the time evolution of a probability density $q(\boldsymbol{x}, t)$. Our goal is to find a PDE whose solution with the initial value $q(\boldsymbol{x}, 0) := p_{\mathrm{data}}(\boldsymbol{x})$ is $q(\boldsymbol{x}, t) = p\big(\boldsymbol{x}, \sigma(t)\big)$. That is, the PDE should reproduce the marginals we postulate in Eq. 20.

The desired marginals are convolutions of $p_{\mathrm{data}}$ with isotropic normal distributions of time-varying standard deviation $\sigma(t)$, and as such, can be generated by the heat equation with time-varying diffusivity $\kappa(t)$. The situation is most conveniently analyzed in the Fourier domain, where the marginal densities are simply pointwise products of a Gaussian function and the transformed data density. To find the diffusivity that induces the correct standard deviations, we first write down the heat equation PDE:

$$\frac{\partial q(\boldsymbol{x}, t)}{\partial t} = \kappa(t)\Delta_{\boldsymbol{x}} q(\boldsymbol{x}, t). \tag{82}$$

The Fourier transformed counterpart of Eq. 82, where the transform is taken along the $\boldsymbol{x}$-dimension, is given by

$$\frac{\partial \hat{q}(\boldsymbol{\nu}, t)}{\partial t} = -\kappa(t)|\boldsymbol{\nu}|^2 \hat{q}(\boldsymbol{\nu}, t). \tag{83}$$

The target solution $q(\boldsymbol{x}, t)$ and its Fourier transform $\hat{q}(\boldsymbol{\nu}, t)$ are given by Eq. 20:

$$q(\boldsymbol{x}, t) \quad = \quad p\big(\boldsymbol{x}; \sigma(t)\big) = p_{\text{data}}(\boldsymbol{x}) * \mathcal{N}\big(\boldsymbol{0}, \ \sigma(t)^2 \, \mathbf{I}\big) \tag{84}$$

$$\hat{q}(\boldsymbol{\nu}, t) \quad = \quad \hat{p}_{\text{data}}(\boldsymbol{\nu}) \ \exp\left(-\tfrac{1}{2} \, |\boldsymbol{\nu}|^2 \, \sigma(t)^2\right). \tag{85}$$

Differentiating the target solution along the time axis, we have

$$\frac{\partial \hat{q}(\boldsymbol{\nu}, t)}{\partial t} \quad = \quad -\dot{\sigma}(t)\sigma(t) \, |\boldsymbol{\nu}|^2 \, \hat{p}_{\text{data}}(\boldsymbol{\nu}) \ \exp\left(-\tfrac{1}{2} \, |\boldsymbol{\nu}|^2 \, \sigma(t)^2\right) \tag{86}$$

$$= \quad -\dot{\sigma}(t)\sigma(t) \, |\boldsymbol{\nu}|^2 \, \hat{q}(\boldsymbol{\nu}, t). \tag{87}$$

Eqs. 83 and 87 share the same left hand side. Equating them allows us to solve for $\kappa(t)$ that generates the desired evolution:

$$-\kappa(t)|\boldsymbol{\nu}|^2 \hat{q}(\boldsymbol{\nu}, t) \quad = \quad -\dot{\sigma}(t)\sigma(t) \, |\boldsymbol{\nu}|^2 \, \hat{q}(\boldsymbol{\nu}, t) \tag{88}$$

$$\kappa(t) \quad = \quad \dot{\sigma}(t)\sigma(t). \tag{89}$$

To summarize, the desired marginal densities corresponding to noise levels $\sigma(t)$ are generated by the PDE

$$\frac{\partial q(\boldsymbol{x}, t)}{\partial t} = \dot{\sigma}(t)\sigma(t)\Delta_{\boldsymbol{x}} q(\boldsymbol{x}, t) \tag{90}$$

from the initial density $q(\boldsymbol{x}, 0) = p_{\text{data}}(\boldsymbol{x})$.

### B.5.2    Derivation of our SDE

Given an SDE

$$\mathrm{d}\boldsymbol{x} = \boldsymbol{f}(\boldsymbol{x}, t) \, \mathrm{d}t \ + \ \boldsymbol{g}(\boldsymbol{x}, t) \, \mathrm{d}\omega_t, \tag{91}$$

the Fokker–Planck PDE describes the time evolution of its solution probability density $r(\boldsymbol{x}, t)$ as

$$\frac{\partial r(\boldsymbol{x}, t)}{\partial t} = -\nabla_{\boldsymbol{x}} \cdot \big(\boldsymbol{f}(\boldsymbol{x}, t) \, r(\boldsymbol{x}, t)\big) + \tfrac{1}{2}\nabla_{\boldsymbol{x}}\nabla_{\boldsymbol{x}} : \big(\mathbf{D}(\boldsymbol{x}, t) \, r(\boldsymbol{x}, t)\big), \tag{92}$$

where $\mathbf{D}_{ij} = \sum_k \boldsymbol{g}_{ik}\boldsymbol{g}_{jk}$ is the *diffusion tensor*. We consider the special case $\boldsymbol{g}(\boldsymbol{x}, t) = g(t) \, \mathbf{I}$ of $\boldsymbol{x}$-independent white noise addition, whereby the equation simplifies to

$$\frac{\partial r(\boldsymbol{x}, t)}{\partial t} = -\nabla_{\boldsymbol{x}} \cdot \big(\boldsymbol{f}(\boldsymbol{x}, t) \, r(\boldsymbol{x}, t)\big) + \tfrac{1}{2} \, g(t)^2 \, \Delta_{\boldsymbol{x}} r(\boldsymbol{x}, t). \tag{93}$$

We are seeking an SDE whose solution density is described by the PDE in Eq. 90. Setting $r(\boldsymbol{x}, t) = q(\boldsymbol{x}, t)$ and equating Eqs. 93 and 90, we find the sufficient condition that the SDE must satisfy

$$-\nabla_{\boldsymbol{x}} \cdot \big(\boldsymbol{f}(\boldsymbol{x}, t) \, q(\boldsymbol{x}, t)\big) + \tfrac{1}{2} \, g(t)^2 \, \Delta_{\boldsymbol{x}} q(\boldsymbol{x}, t) \quad = \quad \dot{\sigma}(t) \, \sigma(t) \, \Delta_{\boldsymbol{x}} q(\boldsymbol{x}, t) \tag{94}$$

$$\nabla_{\boldsymbol{x}} \cdot \big(\boldsymbol{f}(\boldsymbol{x}, t) \, q(\boldsymbol{x}, t)\big) \quad = \quad \left(\tfrac{1}{2} \, g(t)^2 - \dot{\sigma}(t) \, \sigma(t)\right) \Delta_{\boldsymbol{x}} q(\boldsymbol{x}, t). \tag{95}$$

Any choice of functions $\boldsymbol{f}(\boldsymbol{x}, t)$ and $g(t)$ satisfying this equation constitute a sought after SDE. Let us now find a specific family of such solutions. The key idea is given by the identity $\nabla_{\boldsymbol{x}} \cdot \nabla_{\boldsymbol{x}} = \Delta_{\boldsymbol{x}}$. Indeed, if we set $\boldsymbol{f}(\boldsymbol{x}, t) \, q(\boldsymbol{x}, t) = \upsilon(t) \, \nabla_{\boldsymbol{x}} q(\boldsymbol{x}, t)$ for any choice of $\upsilon(t)$, the term $\Delta_{\boldsymbol{x}} q(\boldsymbol{x}, t)$ appears on both sides and cancels out:

$$\nabla_{\boldsymbol{x}} \cdot \big(\upsilon(t) \, \nabla_{\boldsymbol{x}} q(\boldsymbol{x}, t)\big) \quad = \quad \left(\tfrac{1}{2} \, g(t)^2 - \dot{\sigma}(t) \, \sigma(t)\right) \Delta_{\boldsymbol{x}} q(\boldsymbol{x}, t) \tag{96}$$

$$\upsilon(t) \, \Delta_{\boldsymbol{x}} q(\boldsymbol{x}, t) \quad = \quad \left(\tfrac{1}{2} \, g(t)^2 - \dot{\sigma}(t) \, \sigma(t)\right) \Delta_{\boldsymbol{x}} q(\boldsymbol{x}, t) \tag{97}$$

$$\upsilon(t) \quad = \quad \tfrac{1}{2} \, g(t)^2 - \dot{\sigma}(t) \, \sigma(t). \tag{98}$$

The stated $\boldsymbol{f}(\boldsymbol{x}, t)$ is in fact proportional to the score function, as the formula matches the gradient of the logarithm of the density:

$$\boldsymbol{f}(\boldsymbol{x}, t) \quad = \quad \upsilon(t) \, \frac{\nabla_{\boldsymbol{x}} q(\boldsymbol{x}, t)}{q(\boldsymbol{x}, t)} \tag{99}$$

$$= \quad \upsilon(t) \, \nabla_{\boldsymbol{x}} \log q(\boldsymbol{x}, t) \tag{100}$$

$$= \quad \left(\tfrac{1}{2} \, g(t)^2 - \dot{\sigma}(t) \, \sigma(t)\right) \nabla_{\boldsymbol{x}} \log q(\boldsymbol{x}, t). \tag{101}$$

Substituting this back into Eq. 91 and writing $p(\boldsymbol{x}; \sigma(t))$ in place of $q(\boldsymbol{x}, t)$, we recover a family of SDEs whose solution densities have the desired marginals with noise levels $\sigma(t)$ for any choice of $g(t)$:

$$\mathrm{d}\boldsymbol{x} = \left( \tfrac{1}{2} g(t)^2 - \dot{\sigma}(t)\, \sigma(t) \right) \nabla_{\boldsymbol{x}} \log p\big(\boldsymbol{x}; \sigma(t)\big) \, \mathrm{d}t \; + \; g(t)\, \mathrm{d}\omega_t. \tag{102}$$

The free parameter $g(t)$ effectively specifies the rate of noise replacement at any given time instance. The special case choice of $g(t) = 0$ corresponds to the probability flow ODE. The parametrization by $g(t)$ is not particularly intuitive, however. To obtain a more interpretable parametrization, we set $g(t) = \sqrt{2\,\beta(t)}\, \sigma(t)$, which yields the (forward) SDE of Eq. 6 in the main paper:

$$\mathrm{d}\boldsymbol{x}_+ = -\dot{\sigma}(t)\sigma(t)\nabla_{\boldsymbol{x}} \log p\big(\boldsymbol{x}; \sigma(t)\big) \, \mathrm{d}t \; + \; \beta(t)\sigma(t)^2 \nabla_{\boldsymbol{x}} \log p\big(\boldsymbol{x}; \sigma(t)\big) \, \mathrm{d}t + \sqrt{2\beta(t)}\sigma(t)\, \mathrm{d}\omega_t. \tag{103}$$

The noise replacement is now proportional to the standard deviation $\sigma(t)$ of the noise, with the proportionality factor $\beta(t)$. Indeed, expanding the score function in the middle term according to Eq. 3 yields $\beta(t) \big[ D\big(\boldsymbol{x}; \sigma(t)\big) - \boldsymbol{x} \big] \, \mathrm{d}t$, which changes $\boldsymbol{x}$ proportionally to the negative noise component; the stochastic term injects new noise at the same rate. Intuitively, scaling the magnitude of Langevin exploration according to the current noise standard deviation is a reasonable baseline, as the data manifold is effectively "spread out" by this amount due to the blurring of the density.

The *reverse* SDE used in denoising diffusion is simply obtained by applying the time reversal formula of Anderson [1] (as stated in Eq. 6 of Song et al. [15]) on Eq. 103; the entire effect of the reversal is a change of sign in the middle term.

The scaled generalization of the SDE can be derived using a similar approach as with the ODE previously. As such, the derivation is omitted here.

### B.6 Our preconditioning and training (Eq. 8)

Following Eq. 2, the denoising score matching loss for a given denoiser $D_\theta$ on a given noise level $\sigma$ is given by

$$\mathcal{L}(D_\theta; \sigma) = \mathbb{E}_{\boldsymbol{y} \sim p_{\text{data}}} \mathbb{E}_{\boldsymbol{n} \sim \mathcal{N}(\boldsymbol{0}, \sigma^2 \mathbf{I})} \left\| D_\theta(\boldsymbol{y} + \boldsymbol{n}; \sigma) - \boldsymbol{y} \right\|_2^2. \tag{104}$$

We obtain overall training loss by taking a weighted expectation of $\mathcal{L}(D_\theta; \sigma)$ over the noise levels:

$$\mathcal{L}(D_\theta) \;=\; \mathbb{E}_{\sigma \sim p_{\text{train}}} \Big[ \lambda(\sigma)\, \mathcal{L}(D_\theta; \sigma) \Big] \tag{105}$$

$$=\; \mathbb{E}_{\sigma \sim p_{\text{train}}} \left[ \lambda(\sigma)\, \mathbb{E}_{\boldsymbol{y} \sim p_{\text{data}}} \mathbb{E}_{\boldsymbol{n} \sim \mathcal{N}(\boldsymbol{0}, \sigma^2 \mathbf{I})} \left\| D_\theta(\boldsymbol{y} + \boldsymbol{n}; \sigma) - \boldsymbol{y} \right\|_2^2 \right] \tag{106}$$

$$=\; \mathbb{E}_{\sigma \sim p_{\text{train}}} \mathbb{E}_{\boldsymbol{y} \sim p_{\text{data}}} \mathbb{E}_{\boldsymbol{n} \sim \mathcal{N}(\boldsymbol{0}, \sigma^2 \mathbf{I})} \left[ \lambda(\sigma) \left\| D_\theta(\boldsymbol{y} + \boldsymbol{n}; \sigma) - \boldsymbol{y} \right\|_2^2 \right] \tag{107}$$

$$=\; \mathbb{E}_{\sigma, \boldsymbol{y}, \boldsymbol{n}} \left[ \lambda(\sigma) \left\| D_\theta(\boldsymbol{y} + \boldsymbol{n}; \sigma) - \boldsymbol{y} \right\|_2^2 \right], \tag{108}$$

where the noise levels are distributed according to $\sigma \sim p_{\text{train}}$ and weighted by $\lambda(\sigma)$.

Using our definition of $D_\theta(\cdot)$ from Eq. 7, we can further rewrite $\mathcal{L}(D_\theta)$ as

$$\mathbb{E}_{\sigma, \boldsymbol{y}, \boldsymbol{n}} \left[ \lambda(\sigma) \big\| c_{\text{skip}}(\sigma)(\boldsymbol{y} + \boldsymbol{n}) + c_{\text{out}}(\sigma) F_\theta\big(c_{\text{in}}(\sigma)(\boldsymbol{y} + \boldsymbol{n}); c_{\text{noise}}(\sigma)\big) - \boldsymbol{y} \big\|_2^2 \right] \tag{109}$$

$$=\; \mathbb{E}_{\sigma, \boldsymbol{y}, \boldsymbol{n}} \left[ \lambda(\sigma) \big\| c_{\text{out}}(\sigma) F_\theta\big(c_{\text{in}}(\sigma)(\boldsymbol{y} + \boldsymbol{n}); c_{\text{noise}}(\sigma)\big) - \big(\boldsymbol{y} - c_{\text{skip}}(\sigma)(\boldsymbol{y} + \boldsymbol{n})\big) \big\|_2^2 \right] \tag{110}$$

$$=\; \mathbb{E}_{\sigma, \boldsymbol{y}, \boldsymbol{n}} \left[ \lambda(\sigma) c_{\text{out}}(\sigma)^2 \big\| F_\theta\big(c_{\text{in}}(\sigma)(\boldsymbol{y} + \boldsymbol{n}); c_{\text{noise}}(\sigma)\big) - \tfrac{1}{c_{\text{out}}(\sigma)} \big(\boldsymbol{y} - c_{\text{skip}}(\sigma)(\boldsymbol{y} + \boldsymbol{n})\big) \big\|_2^2 \right] \tag{111}$$

$$=\; \mathbb{E}_{\sigma, \boldsymbol{y}, \boldsymbol{n}} \left[ w(\sigma) \big\| F_\theta\big(c_{\text{in}}(\sigma)(\boldsymbol{y} + \boldsymbol{n}); c_{\text{noise}}(\sigma)\big) - F_{\text{target}}(\boldsymbol{y}, \boldsymbol{n}; \sigma) \big\|_2^2 \right], \tag{112}$$

which matches Eq. 8 and corresponds to traditional supervised training of $F_\theta$ using standard $L_2$ loss with effective weight $w(\cdot)$ and target $F_{\text{target}}(\cdot)$ given by

$$w(\sigma) = \lambda(\sigma)\, c_{\text{out}}(\sigma)^2 \quad \text{and} \quad F_{\text{target}}(\boldsymbol{y}, \boldsymbol{n}; \sigma) = \tfrac{1}{c_{\text{out}}(\sigma)} \big(\boldsymbol{y} - c_{\text{skip}}(\sigma)(\boldsymbol{y} + \boldsymbol{n})\big), \tag{113}$$

We can now derive formulas for $c_{\text{in}}(\sigma)$, $c_{\text{out}}(\sigma)$, $c_{\text{skip}}(\sigma)$, and $\lambda(\sigma)$ from first principles, shown in the "Ours" column of Table 1.

First, we require the training inputs of $F_\theta(\cdot)$ to have unit variance:

$$\text{Var}_{\boldsymbol{y},\boldsymbol{n}}\left[c_{\text{in}}(\sigma)(\boldsymbol{y}+\boldsymbol{n})\right] = 1 \tag{114}$$

$$c_{\text{in}}(\sigma)^2\ \text{Var}_{\boldsymbol{y},\boldsymbol{n}}\left[\boldsymbol{y}+\boldsymbol{n}\right] = 1 \tag{115}$$

$$c_{\text{in}}(\sigma)^2\left(\sigma_{\text{data}}^2+\sigma^2\right) = 1 \tag{116}$$

$$c_{\text{in}}(\sigma) = 1/\sqrt{\sigma^2+\sigma_{\text{data}}^2}. \tag{117}$$

Second, we require the effective training target $F_{\text{target}}$ to have unit variance:

$$\text{Var}_{\boldsymbol{y},\boldsymbol{n}}\left[F_{\text{target}}(\boldsymbol{y},\boldsymbol{n};\sigma)\right] = 1 \tag{118}$$

$$\text{Var}_{\boldsymbol{y},\boldsymbol{n}}\left[\tfrac{1}{c_{\text{out}}(\sigma)}\left(\boldsymbol{y}-c_{\text{skip}}(\sigma)(\boldsymbol{y}+\boldsymbol{n})\right)\right] = 1 \tag{119}$$

$$\tfrac{1}{c_{\text{out}}(\sigma)^2}\ \text{Var}_{\boldsymbol{y},\boldsymbol{n}}\left[\boldsymbol{y}-c_{\text{skip}}(\sigma)(\boldsymbol{y}+\boldsymbol{n})\right] = 1 \tag{120}$$

$$c_{\text{out}}(\sigma)^2 = \text{Var}_{\boldsymbol{y},\boldsymbol{n}}\left[\boldsymbol{y}-c_{\text{skip}}(\sigma)(\boldsymbol{y}+\boldsymbol{n})\right] \tag{121}$$

$$c_{\text{out}}(\sigma)^2 = \text{Var}_{\boldsymbol{y},\boldsymbol{n}}\left[\left(1-c_{\text{skip}}(\sigma)\right)\boldsymbol{y}-c_{\text{skip}}(\sigma)\,\boldsymbol{n}\right] \tag{122}$$

$$c_{\text{out}}(\sigma)^2 = \left(1-c_{\text{skip}}(\sigma)\right)^2\sigma_{\text{data}}^2+c_{\text{skip}}(\sigma)^2\,\sigma^2. \tag{123}$$

Third, we select $c_{\text{skip}}(\sigma)$ to minimize $c_{\text{out}}(\sigma)$, so that the errors of $F_\theta$ are amplified as little as possible:

$$c_{\text{skip}}(\sigma) = \arg\min_{c_{\text{skip}}(\sigma)} c_{\text{out}}(\sigma). \tag{124}$$

Since $c_{\text{out}}(\sigma) \geq 0$, we can equivalently write

$$c_{\text{skip}}(\sigma) = \arg\min_{c_{\text{skip}}(\sigma)} c_{\text{out}}(\sigma)^2. \tag{125}$$

This is a convex optimization problem; its solution is uniquely identified by setting the derivative w.r.t. $c_{\text{skip}}(\sigma)$ to zero:

$$0 = \text{d}\left[c_{\text{out}}(\sigma)^2\right]/\text{d}c_{\text{skip}}(\sigma) \tag{126}$$

$$0 = \text{d}\left[\left(1-c_{\text{skip}}(\sigma)\right)^2\sigma_{\text{data}}^2+c_{\text{skip}}(\sigma)^2\,\sigma^2\right]/\text{d}c_{\text{skip}}(\sigma) \tag{127}$$

$$0 = \sigma_{\text{data}}^2\,\text{d}\left[\left(1-c_{\text{skip}}(\sigma)\right)^2\right]/\text{d}c_{\text{skip}}(\sigma)+\sigma^2\,\text{d}\left[c_{\text{skip}}(\sigma)^2\right]/\text{d}c_{\text{skip}}(\sigma) \tag{128}$$

$$0 = \sigma_{\text{data}}^2\left[2\,c_{\text{skip}}(\sigma)-2\right]+\sigma^2\left[2\,c_{\text{skip}}(\sigma)\right] \tag{129}$$

$$0 = \left(\sigma^2+\sigma_{\text{data}}^2\right)c_{\text{skip}}(\sigma)-\sigma_{\text{data}}^2 \tag{130}$$

$$c_{\text{skip}}(\sigma) = \sigma_{\text{data}}^2/\left(\sigma^2+\sigma_{\text{data}}^2\right). \tag{131}$$

We can now substitute Eq. 131 into Eq. 123 to complete the formula for $c_{\text{out}}(\sigma)$:

$$c_{\text{out}}(\sigma)^2 = \left(1-\left[c_{\text{skip}}(\sigma)\right]\right)^2\sigma_{\text{data}}^2+\left[c_{\text{skip}}(\sigma)\right]^2\sigma^2 \tag{132}$$

$$c_{\text{out}}(\sigma)^2 = \left(1-\left[\frac{\sigma_{\text{data}}^2}{\sigma^2+\sigma_{\text{data}}^2}\right]\right)^2\sigma_{\text{data}}^2+\left[\frac{\sigma_{\text{data}}^2}{\sigma^2+\sigma_{\text{data}}^2}\right]^2\sigma^2 \tag{133}$$

$$c_{\text{out}}(\sigma)^2 = \left[\frac{\sigma^2\,\sigma_{\text{data}}}{\sigma^2+\sigma_{\text{data}}^2}\right]^2+\left[\frac{\sigma_{\text{data}}^2\,\sigma}{\sigma^2+\sigma_{\text{data}}^2}\right]^2 \tag{134}$$

$$c_{\text{out}}(\sigma)^2 = \frac{\left(\sigma^2\,\sigma_{\text{data}}\right)^2+\left(\sigma_{\text{data}}^2\,\sigma\right)^2}{\left(\sigma^2+\sigma_{\text{data}}^2\right)^2} \tag{135}$$

$$c_{\text{out}}(\sigma)^2 = \frac{\left(\sigma\cdot\sigma_{\text{data}}\right)^2\left(\sigma^2+\sigma_{\text{data}}^2\right)}{\left(\sigma^2+\sigma_{\text{data}}^2\right)^2} \tag{136}$$

$$c_{\text{out}}(\sigma)^2 = \frac{\left(\sigma\cdot\sigma_{\text{data}}\right)^2}{\sigma^2+\sigma_{\text{data}}^2} \tag{137}$$

$$c_{\text{out}}(\sigma) = \sigma\cdot\sigma_{\text{data}}/\sqrt{\sigma^2+\sigma_{\text{data}}^2}. \tag{138}$$

Fourth, we require the effective weight $w(\sigma)$ to be uniform across noise levels:

$$w(\sigma) = 1 \tag{139}$$

$$\lambda(\sigma)\, c_{\text{out}}(\sigma)^2 = 1 \tag{140}$$

$$\lambda(\sigma) = 1/c_{\text{out}}(\sigma)^2 \tag{141}$$

$$\lambda(\sigma) = 1/\left[\frac{\sigma \cdot \sigma_{\text{data}}}{\sqrt{\sigma^2 + \sigma_{\text{data}}^2}}\right]^2 \tag{142}$$

$$\lambda(\sigma) = 1/\left[\frac{(\sigma \cdot \sigma_{\text{data}})^2}{\sigma^2 + \sigma_{\text{data}}^2}\right] \tag{143}$$

$$\lambda(\sigma) = \left(\sigma^2 + \sigma_{\text{data}}^2\right)/(\sigma \cdot \sigma_{\text{data}})^2. \tag{144}$$

We follow previous work and initialize the output layer weights to zero. Consequently, upon initialization $F_\theta(\cdot) = 0$ and the expected value of the loss at each noise level is 1. This can be seen by substituting the choices of $\lambda(\sigma)$ and $c_{\text{skip}}(\sigma)$ into Eq. 109, considered at a fixed $\sigma$:

$$\mathbb{E}_{\boldsymbol{y},\boldsymbol{n}}\left[\lambda(\sigma)\big\|c_{\text{skip}}(\sigma)(\boldsymbol{y}+\boldsymbol{n}) + c_{\text{out}}(\sigma)F_\theta\big(c_{\text{in}}(\sigma)(\boldsymbol{y}+\boldsymbol{n}); c_{\text{noise}}(\sigma)\big) - \boldsymbol{y}\big\|_2^2\right] \tag{145}$$

$$= \mathbb{E}_{\boldsymbol{y},\boldsymbol{n}}\left[\frac{\sigma^2 + \sigma_{\text{data}}^2}{(\sigma \cdot \sigma_{\text{data}})^2}\left\|\frac{\sigma_{\text{data}}^2}{\sigma^2 + \sigma_{\text{data}}^2}(\boldsymbol{y}+\boldsymbol{n}) - \boldsymbol{y}\right\|_2^2\right] \tag{146}$$

$$= \mathbb{E}_{\boldsymbol{y},\boldsymbol{n}}\left[\frac{\sigma^2 + \sigma_{\text{data}}^2}{(\sigma \cdot \sigma_{\text{data}})^2}\left\|\frac{\sigma_{\text{data}}^2 \boldsymbol{n} - \sigma^2 \boldsymbol{y}}{\sigma^2 + \sigma_{\text{data}}^2}\right\|_2^2\right] \tag{147}$$

$$= \mathbb{E}_{\boldsymbol{y},\boldsymbol{n}}\left[\frac{1}{\sigma^2 + \sigma_{\text{data}}^2}\left\|\frac{\sigma_{\text{data}}}{\sigma}\boldsymbol{n} - \frac{\sigma}{\sigma_{\text{data}}}\boldsymbol{y}\right\|_2^2\right] \tag{148}$$

$$= \frac{1}{\sigma^2 + \sigma_{\text{data}}^2}\mathbb{E}_{\boldsymbol{y},\boldsymbol{n}}\left[\frac{\sigma_{\text{data}}^2}{\sigma^2}\langle\boldsymbol{n},\boldsymbol{n}\rangle + \frac{\sigma^2}{\sigma_{\text{data}}^2}\langle\boldsymbol{y},\boldsymbol{y}\rangle - 2\langle\boldsymbol{y},\boldsymbol{n}\rangle\right] \tag{149}$$

$$= \frac{1}{\sigma^2 + \sigma_{\text{data}}^2}\left[\frac{\sigma_{\text{data}}^2}{\sigma^2}\underbrace{\text{Var}(\boldsymbol{n})}_{=\sigma^2} + \frac{\sigma^2}{\sigma_{\text{data}}^2}\underbrace{\text{Var}(\boldsymbol{y})}_{=\sigma_{\text{data}}^2} - 2\underbrace{\text{Cov}(\boldsymbol{y},\boldsymbol{n})}_{=0}\right] \tag{150}$$

$$= 1 \tag{151}$$

## C  Reframing previous methods in our framework

In this section, we derive the formulas shown in Table 1 for previous methods, discuss the corresponding original samplers and pre-trained models, and detail the practical considerations associated with using them in our framework.

In practice, the original implementations of these methods differ considerably in terms of the definitions of model inputs and outputs, dynamic range of image data, scaling of $\boldsymbol{x}$, and interpretation of $\sigma$. We eliminate this variation by standardizing on a unified setup where the model always matches our definition of $F_\theta$, image data is always represented in the continuous range $[-1, 1]$, and the details of $\boldsymbol{x}$ and $\sigma$ are always in agreement with Eq. 4.

We minimize the accumulation of floating point round-off errors by always executing Algorithms 1 and 2 at double precision (`float64`). However, we still execute the network $F_\theta(\cdot)$ at single precision (`float32`) to minimize runtime and remain faithful to previous work in terms of network architecture.

### C.1  Variance preserving formulation

### C.1.1  VP sampling

Song et al. [15] define the VP SDE (Eq. 32 in [15]) as

$$\mathrm{d}\boldsymbol{x} = -\tfrac{1}{2}\left(\beta_{\min} + t\left(\beta_{\max} - \beta_{\min}\right)\right)\boldsymbol{x}\,\mathrm{d}t + \sqrt{\beta_{\min} + t\left(\beta_{\max} - \beta_{\min}\right)}\,\mathrm{d}\omega_t, \tag{152}$$

which matches Eq. 10 with the following choices for $f$ and $g$:

$$f(t) = -\tfrac{1}{2}\,\beta(t), \quad g(t) = \sqrt{\beta(t)}, \quad \text{and} \quad \beta(t) = \left(\beta_{\max} - \beta_{\min}\right) t + \beta_{\min}. \tag{153}$$

Let $\alpha(t)$ denote the integral of $\beta(t)$:

$$\begin{align}
\alpha(t) &= \int_0^t \beta(\xi)\,\mathrm{d}\xi \tag{154} \\
&= \int_0^t \left[\left(\beta_{\max} - \beta_{\min}\right) \xi + \beta_{\min}\right] \mathrm{d}\xi \tag{155} \\
&= \tfrac{1}{2}\left(\beta_{\max} - \beta_{\min}\right) t^2 + \beta_{\min}\, t \tag{156} \\
&= \tfrac{1}{2}\,\beta_{\mathrm{d}}\, t^2 + \beta_{\min}\, t, \tag{157}
\end{align}$$

where $\beta_{\mathrm{d}} = \beta_{\max} - \beta_{\min}$. We can now obtain the formula for $\sigma(t)$ by substituting Eq. 153 into Eq. 12:

$$\begin{align}
\sigma(t) &= \sqrt{\int_0^t \frac{\left[g(\xi)\right]^2}{\left[s(\xi)\right]^2}\,\mathrm{d}\xi} \tag{158} \\
&= \sqrt{\int_0^t \frac{\left[\sqrt{\beta(\xi)}\right]^2}{\left[1/\sqrt{e^{\alpha(\xi)}}\right]^2}\,\mathrm{d}\xi} \tag{159} \\
&= \sqrt{\int_0^t \frac{\beta(\xi)}{1/e^{\alpha(\xi)}}\,\mathrm{d}\xi} \tag{160} \\
&= \sqrt{\int_0^t \dot\alpha(\xi)\, e^{\alpha(\xi)}\,\mathrm{d}\xi} \tag{161} \\
&= \sqrt{e^{\alpha(t)} - e^{\alpha(0)}} \tag{162} \\
&= \sqrt{e^{\frac{1}{2}\beta_{\mathrm{d}}t^2 + \beta_{\min}t} - 1}, \tag{163}
\end{align}$$

which matches the "Schedule" row of Table 1. Similarly for $s(t)$:

$$\begin{align}
s(t) &= \exp\left(\int_0^t \left[f(\xi)\right]\,\mathrm{d}\xi\right) \tag{164} \\
&= \exp\left(\int_0^t \left[-\tfrac{1}{2}\,\beta(\xi)\right]\,\mathrm{d}\xi\right) \tag{165} \\
&= \exp\left(-\tfrac{1}{2}\left[\int_0^t \beta(\xi)\,\mathrm{d}\xi\right]\right) \tag{166} \\
&= \exp\left(-\tfrac{1}{2}\,\alpha(t)\right) \tag{167} \\
&= 1/\sqrt{e^{\alpha(t)}} \tag{168} \\
&= 1/\sqrt{e^{\frac{1}{2}\beta_{\mathrm{d}}t^2 + \beta_{\min}t}}, \tag{169}
\end{align}$$

which matches the "Scaling" row of Table 1. We can equivalently write Eq. 169 in a slightly simpler form by utilizing Eq. 163:

$$s(t) = 1/\sqrt{\sigma(t)^2 + 1}. \tag{170}$$

Song et al. [15] choose to distribute the sampling time steps $\{t_0, \ldots, t_{N-1}\}$ at uniform intervals within $[\epsilon_{\mathrm{s}}, 1]$. This corresponds to setting

$$t_{i<N} = 1 + \tfrac{i}{N-1}(\epsilon_{\mathrm{s}} - 1), \tag{171}$$

which matches the "Time steps" row of Table 1.

Finally, Song et al. [15] set $\beta_{\min} = 0.1$, $\beta_{\max} = 20$, and $\epsilon_{\mathrm{s}} = 10^{-3}$ (Appendix C in [15]), and choose to represent images in the range $[-1, 1]$. These choices are readily compatible with our formulation and are reflected by the "Parameters" section of Table 1.

### C.1.2 VP preconditioning

In the VP case, Song et al. [15] approximate the score of $p_t(\boldsymbol{x})$ of Eq. 13 as[1]

$$\nabla_{\boldsymbol{x}} \log p_t(\boldsymbol{x}) \approx \underbrace{-\tfrac{1}{\bar{\sigma}(t)} F_\theta\big(\boldsymbol{x};\ (M-1)t\big)}_{\text{score}(\boldsymbol{x};F_\theta,t)}, \tag{172}$$

where $M = 1000$, $F_\theta$ denotes the network, and $\bar{\sigma}(t)$ corresponds to the standard deviation of the perturbation kernel of Eq. 11.

Let us expand the definitions of $p_t(\boldsymbol{x})$ and $\bar{\sigma}(t)$ from Eqs. 20 and 11, respectively, and substitute $\boldsymbol{x} = s(t)\hat{\boldsymbol{x}}$ to obtain the corresponding formula with respect to the non-scaled variable $\hat{\boldsymbol{x}}$:

$$\nabla_{\boldsymbol{x}} \log \big[ p(\boldsymbol{x}/s(t); \sigma(t)) \big] \approx -\tfrac{1}{[s(t)\sigma(t)]} F_\theta\big(\boldsymbol{x};\ (M-1)t\big) \tag{173}$$

$$\nabla_{[s(t)\hat{\boldsymbol{x}}]} \log p\big([s(t)\ \hat{\boldsymbol{x}}]/s(t); \sigma(t)\big) \approx -\tfrac{1}{s(t)\sigma(t)} F_\theta\big([s(t)\ \hat{\boldsymbol{x}}];\ (M-1)t\big) \tag{174}$$

$$\tfrac{1}{s(t)} \nabla_{\hat{\boldsymbol{x}}} \log p\big(\hat{\boldsymbol{x}}; \sigma(t)\big) \approx -\tfrac{1}{s(t)\sigma(t)} F_\theta\big(s(t)\ \hat{\boldsymbol{x}};\ (M-1)t\big) \tag{175}$$

$$\nabla_{\hat{\boldsymbol{x}}} \log p\big(\hat{\boldsymbol{x}}; \sigma(t)\big) \approx -\tfrac{1}{\sigma(t)} F_\theta\big(s(t)\ \hat{\boldsymbol{x}};\ (M-1)t\big). \tag{176}$$

We can now replace the left-hand side with Eq. 3 and expand the definition of $s(t)$ from Eq. 170:

$$\Big[\big(D(\hat{\boldsymbol{x}}; \sigma(t)) - \hat{\boldsymbol{x}}\big)/\sigma(t)^2\Big] \approx -\tfrac{1}{\sigma(t)} F_\theta\big(s(t)\ \hat{\boldsymbol{x}};\ (M-1)t\big) \tag{177}$$

$$D\big(\hat{\boldsymbol{x}}; \sigma(t)\big) \approx \hat{\boldsymbol{x}} - \sigma(t)\ F_\theta\big(s(t)\ \hat{\boldsymbol{x}};\ (M-1)t\big) \tag{178}$$

$$D\big(\hat{\boldsymbol{x}}; \sigma(t)\big) \approx \hat{\boldsymbol{x}} - \sigma(t)\ F_\theta\Big(\Big[\tfrac{1}{\sqrt{\sigma(t)^2+1}}\Big]\ \hat{\boldsymbol{x}};\ (M-1)t\Big), \tag{179}$$

which can be further expressed in terms of $\sigma$ by replacing $\sigma(t) \to \sigma$ and $t \to \sigma^{-1}(\sigma)$:

$$D(\hat{\boldsymbol{x}}; \sigma) \approx \hat{\boldsymbol{x}} - \sigma\ F_\theta\Big(\tfrac{1}{\sqrt{\sigma^2+1}}\ \hat{\boldsymbol{x}};\ (M-1)\ \sigma^{-1}(\sigma)\Big). \tag{180}$$

We adopt the right-hand side of Eq. 180 as the definition of $D_\theta$, obtaining

$$D_\theta(\hat{\boldsymbol{x}}; \sigma) = \underbrace{1}_{c_{\text{skip}}} \cdot\ \hat{\boldsymbol{x}}\ \underbrace{-\ \sigma}_{c_{\text{out}}} \cdot\ F_\theta\Big(\underbrace{\tfrac{1}{\sqrt{\sigma^2+1}}}_{c_{\text{in}}} \cdot\ \hat{\boldsymbol{x}};\ \underbrace{(M-1)\ \sigma^{-1}(\sigma)}_{c_{\text{noise}}}\Big), \tag{181}$$

where $c_{\text{skip}}$, $c_{\text{out}}$, $c_{\text{in}}$, and $c_{\text{noise}}$ match the "Network and preconditioning" section of Table 1.

### C.1.3 VP training

Song et al. [15] define their training loss as[2]

$$\mathbb{E}_{t\sim\mathcal{U}(\epsilon_t,1),\boldsymbol{y}\sim p_{\text{data}},\bar{\boldsymbol{n}}\sim\mathcal{N}(\mathbf{0},\mathbf{I})}\Big[\big\|\bar{\sigma}(t)\ \text{score}\big(s(t)\ \boldsymbol{y} + \bar{\sigma}(t)\ \bar{\boldsymbol{n}};\ F_\theta,t\big) + \bar{\boldsymbol{n}}\big\|_2^2\Big], \tag{182}$$

where the definition of $\text{score}(\cdot)$ is the same as in Eq. 172. Let us simplify the formula by substituting $\bar{\sigma}(t) = s(t)\sigma(t)$ and $\bar{\boldsymbol{n}} = \boldsymbol{n}/\sigma(t)$, where $\boldsymbol{n} \sim \mathcal{N}(\mathbf{0}, \sigma(t)^2\mathbf{I})$:

$$\mathbb{E}_{t,\boldsymbol{y},\bar{\boldsymbol{n}}}\Big[\big\|s(t)\sigma(t)\ \text{score}\big(s(t)\ \boldsymbol{y} + [s(t)\sigma(t)]\ \bar{\boldsymbol{n}};\ F_\theta,t\big) + \bar{\boldsymbol{n}}\big\|_2^2\Big] \tag{183}$$

$$= \mathbb{E}_{t,\boldsymbol{y},\boldsymbol{n}}\Big[\big\|s(t)\sigma(t)\ \text{score}\big(s(t)\ \boldsymbol{y} + s(t)\sigma(t)\ [\boldsymbol{n}/\sigma(t)];\ F_\theta,t\big) + [\boldsymbol{n}/\sigma(t)]\big\|_2^2\Big] \tag{184}$$

$$= \mathbb{E}_{t,\boldsymbol{y},\boldsymbol{n}}\Big[\big\|s(t)\sigma(t)\ \text{score}\big(s(t)\ (\boldsymbol{y} + \boldsymbol{n});\ F_\theta,t\big) + \boldsymbol{n}/\sigma(t)\big\|_2^2\Big]. \tag{185}$$

We can express $\text{score}(\cdot)$ in terms of $D_\theta(\cdot)$ by combining Eqs. 172, 170, and 74:

$$\text{score}\big(s(t)\ \boldsymbol{x}; F_\theta, t\big) = \tfrac{1}{s(t)\sigma(t)^2}\Big(D_\theta\big(\boldsymbol{x}; \sigma(t)\big) - \boldsymbol{x}\Big). \tag{186}$$

---

[1] https://github.com/yang-song/score_sde_pytorch/blob/1618ddea340f3e4a2ed7852a0694a809775cf8d0/models/utils.py#L144

[2] https://github.com/yang-song/score_sde_pytorch/blob/1618ddea340f3e4a2ed7852a0694a809775cf8d0/losses.py#L73

Substituting this back into Eq. 185 gives

$$\mathbb{E}_{t,\boldsymbol{y},\boldsymbol{n}}\left[\left\|s(t)\sigma(t)\left[\tfrac{1}{s(t)\sigma(t)^2}\Big(D_\theta\big(\boldsymbol{y}+\boldsymbol{n};\sigma(t)\big)-(\boldsymbol{y}+\boldsymbol{n})\Big)\right]+\tfrac{1}{\sigma(t)}\,\boldsymbol{n}\right\|_2^2\right] \tag{187}$$

$$=\quad\mathbb{E}_{t,\boldsymbol{y},\boldsymbol{n}}\left[\left\|\tfrac{1}{\sigma(t)}\Big(D_\theta\big(\boldsymbol{y}+\boldsymbol{n};\sigma(t)\big)-(\boldsymbol{y}+\boldsymbol{n})\Big)+\tfrac{1}{\sigma(t)}\,\boldsymbol{n}\right\|_2^2\right] \tag{188}$$

$$=\quad\mathbb{E}_{t,\boldsymbol{y},\boldsymbol{n}}\left[\tfrac{1}{\sigma(t)^2}\left\|D_\theta\big(\boldsymbol{y}+\boldsymbol{n};\sigma(t)\big)-\boldsymbol{y}\right\|_2^2\right]. \tag{189}$$

We can further express this in terms of $\sigma$ by replacing $\sigma(t)\to\sigma$ and $t\to\sigma^{-1}(\sigma)$:

$$\underbrace{\mathbb{E}_{\sigma^{-1}(\sigma)\sim\mathcal{U}(\epsilon_{\mathrm{t}},1)}}_{p_{\text{train}}}\mathbb{E}_{\boldsymbol{y},\boldsymbol{n}}\left[\underbrace{\tfrac{1}{\sigma^2}}_{\lambda}\left\|D_\theta\big(\boldsymbol{y}+\boldsymbol{n};\sigma\big)-\boldsymbol{y}\right\|_2^2\right], \tag{190}$$

which matches Eq. 108 with the choices for $p_{\text{train}}$ and $\lambda$ shown in the "Training" section of Table 1.

### C.1.4 VP practical considerations

The pre-trained VP model that we use on CIFAR-10 corresponds to the "DDPM++ cont. (VP)" checkpoint[3] provided by Song et al. [15]. It contains a total of 62 million trainable parameters and supports a continuous range of noise levels $\sigma\in\big[\sigma(\epsilon_{\mathrm{t}}),\sigma(1)\big]\approx[0.001,152]$, i.e., wider than our preferred sampling range $[0.002,80]$. We import the model directly as $F_\theta(\cdot)$ and run Algorithms 1 and 2 using the definitions in Table 1.

In Figure 2a, the differences between the original sampler (blue) and our reimplementation (orange) are explained by oversights in the implementation of Song et al. [15], also noted by Jolicoeur-Martineau et al. [8] (Appendix D in [8]). First, the original sampler employs an incorrect multiplier[4] in the Euler step: it multiplies $\mathrm{d}\boldsymbol{x}/\mathrm{d}t$ by $-1/N$ instead of $(\epsilon_{\mathrm{s}}-1)/(N-1)$. Second, it either overshoots or undershoots on the last step by going from $t_{N-1}=\epsilon_{\mathrm{s}}$ to $t_N=\epsilon_{\mathrm{s}}-1/N$, where $t_N<0$ when $N<1000$. In practice, this means that the generated images contain noticeable noise that becomes quite severe with, e.g., $N=128$. Our formulation avoids these issues, because the step sizes in Algorithm 1 are computed consistently from $\{t_i\}$ and $t_N=0$.

### C.2 Variance exploding formulation

### C.2.1 VE sampling in theory

Song et al. [15] define the VE SDE (Eq. 30 in [15]) as

$$\mathrm{d}\boldsymbol{x}=\sigma_{\min}\left(\frac{\sigma_{\max}}{\sigma_{\min}}\right)^t\sqrt{2\log\frac{\sigma_{\max}}{\sigma_{\min}}}\,\mathrm{d}\omega_t, \tag{191}$$

which matches Eq. 10 with

$$f(t)=0,\quad g(t)=\sigma_{\min}\sqrt{2\log\sigma_{\mathrm{d}}}\,\sigma_{\mathrm{d}}^t,\quad\text{and}\quad\sigma_{\mathrm{d}}=\sigma_{\max}/\sigma_{\min}. \tag{192}$$

The VE formulation does not employ scaling, which can be easily seen from Eq. 12:

$$s(t)=\exp\left(\int_0^t\big[f(\xi)\big]\,\mathrm{d}\xi\right)=\exp\left(\int_0^t\big[0\big]\,\mathrm{d}\xi\right)=\exp(0)=1. \tag{193}$$

---

[3] `vp/cifar10_ddpmpp_continuous/checkpoint_8.pth`, https://drive.google.com/drive/folders/1xYjVMx10N9ivQQBIsEoXEeu9nvSGTBrC

[4] https://github.com/yang-song/score_sde_pytorch/blob/1618ddea340f3e4a2ed7852a0694a809775cf8d0/sampling.py#L182

Substituting Eq. 192 into Eq. 12 suggests the following form for $\sigma(t)$:

$$\sigma(t) \;=\; \sqrt{\int_0^t \frac{\big[g(\xi)\big]^2}{\big[s(\xi)\big]^2}\,\mathrm{d}\xi} \tag{194}$$

$$=\; \sqrt{\int_0^t \frac{\big[\sigma_{\min}\sqrt{2\log\sigma_{\mathrm{d}}}\;\sigma_{\mathrm{d}}^{\xi}\big]^2}{\big[1\big]^2}\,\mathrm{d}\xi} \tag{195}$$

$$=\; \sqrt{\int_0^t \sigma_{\min}^2\,\big[2\log\sigma_{\mathrm{d}}\big]\,\big[\sigma_{\mathrm{d}}^{2\xi}\big]\,\mathrm{d}\xi} \tag{196}$$

$$=\; \sigma_{\min}\sqrt{\int_0^t \Big[\log\big(\sigma_{\mathrm{d}}^2\big)\Big]\,\Big[\big(\sigma_{\mathrm{d}}^2\big)^{\xi}\Big]\,\mathrm{d}\xi} \tag{197}$$

$$=\; \sigma_{\min}\sqrt{\big(\sigma_{\mathrm{d}}^2\big)^t - \big(\sigma_{\mathrm{d}}^2\big)^0} \tag{198}$$

$$=\; \sigma_{\min}\sqrt{\sigma_{\mathrm{d}}^{2t} - 1}. \tag{199}$$

Eq. 199 is consistent with the perturbation kernel reported by Song et al. (Eq. 29 in [15]). However, we note that this does not fulfill their intended definition of $\sigma(t) = \sigma_{\min}\left(\frac{\sigma_{\max}}{\sigma_{\min}}\right)^t$ (Appendix C in [15]).

### C.2.2 VE sampling in practice

The original implementation[5] of Song et al. [15] uses reverse diffusion predictor[6] to integrate discretized reverse probability flow[7] of discretized VE SDE[8]. Put together, these yield the following update rule for $\boldsymbol{x}_{i+1}$:

$$\boldsymbol{x}_{i+1} = \boldsymbol{x}_i + \tfrac{1}{2}\left(\bar{\sigma}_i^2 - \bar{\sigma}_{i+1}^2\right)\nabla_{\boldsymbol{x}}\log\bar{p}_i(\boldsymbol{x}), \tag{200}$$

where

$$\bar{\sigma}_{i<N} = \sigma_{\min}\left(\frac{\sigma_{\max}}{\sigma_{\min}}\right)^{1-i/(N-1)} \quad\text{and}\quad \bar{\sigma}_N = 0. \tag{201}$$

Interestingly, Eq. 200 is identical to the Euler iteration of our ODE with the following choices:

$$s(t) = 1, \quad \sigma(t) = \sqrt{t}, \quad\text{and}\quad t_i = \bar{\sigma}_i^2. \tag{202}$$

These formulas match the "Sampling" section of Table 1, and their correctness can be verified by substituting them into line 5 of Algorithm 1:

$$\boldsymbol{x}_{i+1} \;=\; \boldsymbol{x}_i + (t_{i+1} - t_i)\,\boldsymbol{d}_i \tag{203}$$

$$=\; \boldsymbol{x}_i + (t_{i+1} - t_i)\left[\left(\frac{\dot{\sigma}(t)}{\sigma(t)} + \frac{\dot{s}(t)}{s(t)}\right)\boldsymbol{x} - \frac{\dot{\sigma}(t)s(t)}{\sigma(t)}D\left(\frac{\boldsymbol{x}}{s(t)};\sigma(t)\right)\right] \tag{204}$$

$$=\; \boldsymbol{x}_i + (t_{i+1} - t_i)\left[\frac{\dot{\sigma}(t)}{\sigma(t)}\,\boldsymbol{x} - \frac{\dot{\sigma}(t)}{\sigma(t)}\,D\big(\boldsymbol{x};\sigma(t)\big)\right] \tag{205}$$

$$=\; \boldsymbol{x}_i - (t_{i+1} - t_i)\,\dot{\sigma}(t)\,\sigma(t)\left[\big(D\big(\boldsymbol{x};\sigma(t)\big) - \boldsymbol{x}\big)/\sigma(t)^2\right] \tag{206}$$

$$=\; \boldsymbol{x}_i - (t_{i+1} - t_i)\,\dot{\sigma}(t)\,\sigma(t)\,\nabla_{\boldsymbol{x}}\log p\big(\boldsymbol{x};\sigma(t)\big) \tag{207}$$

$$=\; \boldsymbol{x}_i - (t_{i+1} - t_i)\left[\tfrac{1}{2\sqrt{t}}\right]\left[\sqrt{t}\right]\nabla_{\boldsymbol{x}}\log p\big(\boldsymbol{x};\sigma(t)\big) \tag{208}$$

$$=\; \boldsymbol{x}_i + \tfrac{1}{2}(t_i - t_{i+1})\,\nabla_{\boldsymbol{x}}\log p\big(\boldsymbol{x};\sigma(t)\big) \tag{209}$$

$$=\; \boldsymbol{x}_i + \tfrac{1}{2}\left(\bar{\sigma}_i^2 - \bar{\sigma}_{i+1}^2\right)\nabla_{\boldsymbol{x}}\log p\big(\boldsymbol{x};\sigma(t)\big), \tag{210}$$

---

[5] `https://github.com/yang-song/score_sde_pytorch`

[6] `https://github.com/yang-song/score_sde_pytorch/blob/1618ddea340f3e4a2ed7852a0694a809775cf8d0/sampling.py#L191`

[7] `https://github.com/yang-song/score_sde_pytorch/blob/1618ddea340f3e4a2ed7852a0694a809775cf8d0/sde_lib.py#L102`

[8] `https://github.com/yang-song/score_sde_pytorch/blob/1618ddea340f3e4a2ed7852a0694a809775cf8d0/sde_lib.py#L246`

which is made identical to Eq. 200 by the choice $\bar{p}_i(\boldsymbol{x}) = p\big(\boldsymbol{x}; \sigma(t_i)\big)$.

Finally, Song et al. [15] set $\sigma_{\min} = 0.01$ and $\sigma_{\max} = 50$ for CIFAR-10 (Appendix C in [15]), and choose to represent their images in the range $[0, 1]$ to match previous SMLD models. Since our standardized range $[-1, 1]$ is twice as large, we must multiply $\sigma_{\min}$ and $\sigma_{\max}$ by $2\times$ to compensate. The "Parameters" section of Table 1 reflects these adjusted values.

### C.2.3   VE preconditioning

In the VE case, Song et al. [15] approximate the score of $p_t(\boldsymbol{x})$ of Eq. 13 directly as[9]

$$\nabla_{\boldsymbol{x}} \log p_t(\boldsymbol{x}) \;\approx\; \bar{F}_\theta\big(\boldsymbol{x}; \sigma(t)\big), \tag{211}$$

where the network $\bar{F}_\theta$ is designed to include additional pre-[10] and[11] postprocessing[12] steps:

$$\bar{F}_\theta(\boldsymbol{x}; \sigma) \;=\; \tfrac{1}{\sigma} \, F_\theta\big(2\boldsymbol{x}-1; \log(\sigma)\big). \tag{212}$$

For consistency, we handle the pre- and postprocessing using $\{c_{\text{skip}}, c_{\text{out}}, c_{\text{in}}, c_{\text{noise}}\}$ as opposed to baking them into the network itself.

We cannot use Eqs. 211 and 212 directly in our framework, however, because they assume that the images are represented in range $[0, 1]$. In order to use $[-1, 1]$ instead, we replace $p_t(\boldsymbol{x}) \to p_t(2\boldsymbol{x}-1)$, $\boldsymbol{x} \to \tfrac{1}{2}\boldsymbol{x} + \tfrac{1}{2}$ and $\sigma \to \tfrac{1}{2}\sigma$:

$$\nabla_{[\frac{1}{2}\boldsymbol{x}+\frac{1}{2}]} \log p_t\big(2\big[\tfrac{1}{2}\boldsymbol{x} + \tfrac{1}{2}\big]-1\big) \;\approx\; \tfrac{1}{[\frac{1}{2}\sigma]} \, F_\theta\big(2\big[\tfrac{1}{2}\boldsymbol{x} + \tfrac{1}{2}\big]-1; \log\big[\tfrac{1}{2}\sigma\big]\big) \tag{213}$$

$$2 \, \nabla_{\boldsymbol{x}} \log p_t(\boldsymbol{x}) \;\approx\; \tfrac{2}{\sigma} \, F_\theta\Big(\boldsymbol{x}; \log\big(\tfrac{1}{2}\sigma\big)\Big) \tag{214}$$

$$\nabla_{\boldsymbol{x}} \log p(\boldsymbol{x}; \sigma) \;\approx\; \tfrac{1}{\sigma} \, F_\theta\Big(\boldsymbol{x}; \log\big(\tfrac{1}{2}\sigma\big)\Big). \tag{215}$$

We can now express the model in terms of $D_\theta(\cdot)$ by replacing the left-hand side of Eq. 215 with Eq. 3:

$$\Big(D_\theta(\boldsymbol{x}; \sigma) - \boldsymbol{x}\Big)/\sigma^2 \;=\; \tfrac{1}{\sigma} \, F_\theta\Big(\boldsymbol{x}; \log\big(\tfrac{1}{2}\sigma\big)\Big) \tag{216}$$

$$D_\theta(\boldsymbol{x}; \sigma) \;=\; \underbrace{1 \cdot}_{c_{\text{skip}}} \boldsymbol{x} + \underbrace{\sigma \cdot}_{c_{\text{out}}} F_\theta\Big(\underbrace{1 \cdot}_{c_{\text{in}}} \boldsymbol{x}; \underbrace{\log\big(\tfrac{1}{2}\sigma\big)}_{c_{\text{noise}}}\Big), \tag{217}$$

where $c_{\text{skip}}$, $c_{\text{out}}$, $c_{\text{in}}$, and $c_{\text{noise}}$ match the "Network and preconditioning" section of Table 1.

### C.2.4   VE training

Song et al. [15] define their training loss similarly for VP and VE, so we can reuse Eq. 185 by borrowing the definition of score($\cdot$) from Eq. 216:

$$\mathbb{E}_{t,\boldsymbol{y},\boldsymbol{n}}\Big[\big\|s(t)\sigma(t) \; \text{score}\big(s(t)\,(\boldsymbol{y}+\boldsymbol{n}); \, F_\theta, t\big) + \boldsymbol{n}/\sigma(t)\big\|_2^2\Big] \tag{218}$$

$$= \;\; \mathbb{E}_{t,\boldsymbol{y},\boldsymbol{n}}\Big[\big\|\sigma(t) \; \text{score}\big(\boldsymbol{y}+\boldsymbol{n}; \, F_\theta, t\big) + \boldsymbol{n}/\sigma(t)\big\|_2^2\Big] \tag{219}$$

$$= \;\; \mathbb{E}_{t,\boldsymbol{y},\boldsymbol{n}}\Big[\big\|\sigma(t) \Big[\big(D_\theta(\boldsymbol{y}+\boldsymbol{n}; \sigma(t)) - (\boldsymbol{y}+\boldsymbol{n})\big)/\sigma(t)^2\Big] + \boldsymbol{n}/\sigma(t)\big\|_2^2\Big] \tag{220}$$

$$= \;\; \mathbb{E}_{t,\boldsymbol{y},\boldsymbol{n}}\Big[\tfrac{1}{\sigma(t)^2} \big\|D_\theta(\boldsymbol{y}+\boldsymbol{n}; \sigma(t)) - \boldsymbol{y}\big\|_2^2\Big]. \tag{221}$$

For VE training, the original implementation[13] defines $\sigma(t) = \sigma_{\min} \left(\frac{\sigma_{\max}}{\sigma_{\min}}\right)^t$. We can thus rewrite Eq. 221 as

$$\underbrace{\mathbb{E}_{\ln(\sigma)\sim\mathcal{U}(\ln(\sigma_{\min}),\ln(\sigma_{\max}))}}_{p_{\text{train}}} \mathbb{E}_{\boldsymbol{y},\boldsymbol{n}}\Big[\underbrace{\tfrac{1}{\sigma^2}}_{\lambda} \big\|D_\theta(\boldsymbol{y}+\boldsymbol{n}; \sigma) - \boldsymbol{y}\big\|_2^2\Big], \tag{222}$$

which matches Eq. 108 with the choices for $p_{\text{train}}$ and $\lambda$ shown in the "Training" section of Table 1.

---

[9] https://github.com/yang-song/score_sde_pytorch/blob/1618ddea340f3e4a2ed7852a0694a809775cf8d0/models/utils.py#L163

[10] https://github.com/yang-song/score_sde_pytorch/blob/1618ddea340f3e4a2ed7852a0694a809775cf8d0/models/ncsnpp.py#L239

[11] https://github.com/yang-song/score_sde_pytorch/blob/1618ddea340f3e4a2ed7852a0694a809775cf8d0/models/ncsnpp.py#L261

[12] https://github.com/yang-song/score_sde_pytorch/blob/1618ddea340f3e4a2ed7852a0694a809775cf8d0/models/ncsnpp.py#L379

[13] https://github.com/yang-song/score_sde_pytorch/blob/1618ddea340f3e4a2ed7852a0694a809775cf8d0/sde_lib.py#L234

### C.2.5 VE practical considerations

The pre-trained VE model that we use on CIFAR-10 corresponds to the "NCSN++ cont. (VE)" checkpoint[14] provided by Song et al. [15]. It contains a total of 63 million trainable parameters and supports a continuous range of noise levels $\sigma \in \big[\sigma(\epsilon_{\mathrm{t}}), \sigma(1)\big] \approx [0.02, 100]$. This is narrower than our preferred sampling range $[0.002, 80]$, so we set $\sigma_{\min} = 0.02$ in all related experiments. Note that this limitation is lifted by our training improvements in config E, so we revert back to using $\sigma_{\min} = 0.002$ with configs E and F in Table 2. When importing the model, we remove the pre- and postprocessing steps shown in Eq. 212 to stay consistent with the definition of $F_\theta(\cdot)$ in Eq. 217. With these changes, we can run Algorithms 1 and 2 using the definitions in Table 1.

In Figure 2b, the differences between the original sampler (blue) and our reimplementation (orange) are explained by floating point round-off errors that the original implementation suffers from at high step counts. Our results are more accurate in these cases because we represent $\boldsymbol{x}_i$ at double precision in Algorithm 1.

### C.3 Improved DDPM and DDIM

#### C.3.1 DDIM ODE formulation

Song et al. [14] make the observation that their deterministic DDIM sampler can be expressed as Euler integration of the following ODE (Eq. 14 in [14]):

$$\mathrm{d}\boldsymbol{x}(t) = \epsilon_\theta^{(t)}\left(\frac{\boldsymbol{x}(t)}{\sqrt{\sigma(t)^2 + 1}}\right)\,\mathrm{d}\sigma(t), \tag{223}$$

where $\boldsymbol{x}(t)$ is a scaled version of the iterate that appears in their discrete update formula (Eq. 10 in [14]) and $\epsilon_\theta$ is a model trained to predict the normalized noise vector, i.e., $\epsilon_\theta^{(t)}\big(\boldsymbol{x}(t)/\sqrt{\sigma(t)^2 + 1}\big) \approx \boldsymbol{n}(t)/\sigma(t)$ for $\boldsymbol{x}(t) = \boldsymbol{y}(t) + \boldsymbol{n}(t)$. In our formulation, $D_\theta$ is trained to approximate the clean signal, i.e., $D_\theta\big(\boldsymbol{x}(t); \sigma(t)\big) \approx \boldsymbol{y}$, so we can reinterpret $\epsilon_\theta$ in terms of $D_\theta$ as follows:

$$\boldsymbol{n}(t) = \boldsymbol{x}(t) - \boldsymbol{y}(t) \tag{224}$$

$$\big[\boldsymbol{n}(t)/\sigma(t)\big] = \big(\boldsymbol{x}(t) - \big[\boldsymbol{y}(t)\big]\big)/\sigma(t) \tag{225}$$

$$\epsilon_\theta^{(t)}\big(\boldsymbol{x}(t)/\sqrt{\sigma(t)^2 + 1}\big) = \big(\boldsymbol{x}(t) - D_\theta\big(\boldsymbol{x}(t); \sigma(t)\big)\big)/\sigma(t). \tag{226}$$

Assuming ideal $\epsilon(\cdot)$ and $D(\cdot)$ in $L_2$ sense, we can further simplify the above formula using Eq. 3:

$$\epsilon^{(t)}\big(\boldsymbol{x}(t)/\sqrt{\sigma(t)^2 + 1}\big) = \big(\boldsymbol{x}(t) - D\big(\boldsymbol{x}(t); \sigma(t)\big)\big)/\sigma(t) \tag{227}$$

$$= -\sigma(t)\left[\big(D\big(\boldsymbol{x}(t); \sigma(t)\big) - \boldsymbol{x}(t)\big)/\sigma(t)^2\right] \tag{228}$$

$$= -\sigma(t)\,\nabla_{\boldsymbol{x}(t)} \log p\big(\boldsymbol{x}(t); \sigma(t)\big). \tag{229}$$

Substituting Eq. 229 back into Eq. 223 gives

$$\mathrm{d}\boldsymbol{x}(t) = -\sigma(t)\,\nabla_{\boldsymbol{x}(t)} \log p\big(\boldsymbol{x}(t); \sigma(t)\big)\,\mathrm{d}\sigma(t), \tag{230}$$

which we can further simplify by setting $\sigma(t) = t$:

$$\mathrm{d}\boldsymbol{x} = -t\,\nabla_{\boldsymbol{x}} \log p\big(\boldsymbol{x}; \sigma(t)\big)\,\mathrm{d}t. \tag{231}$$

This matches our Eq. 4 with $s(t) = 1$ and $\sigma(t) = t$, reflected by the "Sampling" section of Table 1.

#### C.3.2 iDDPM time step discretization

The original DDPM formulation of Ho et al. [6] defines the forward process (Eq. 2 in [6]) as a Markov chain that gradually adds Gaussian noise to $\bar{\boldsymbol{x}}_0 \sim p_{\mathrm{data}}$ according to a discrete variance schedule $\{\beta_1, \ldots, \beta_T\}$:

$$q(\bar{\boldsymbol{x}}_t \mid \bar{\boldsymbol{x}}_{t-1}) = \mathcal{N}\big(\bar{\boldsymbol{x}}_t;\ \sqrt{1 - \beta_t}\,\bar{\boldsymbol{x}}_{t-1},\ \beta_t\,\mathbf{I}\big). \tag{232}$$

---

[14] `ve/cifar10_ncsnpp_continuous/checkpoint_24.pth`, https://drive.google.com/drive/folders/1b0gy_LLgO_DaQBgoWXwlVnL_rcAUgREh

The corresponding transition probability from $\bar{x}_0$ to $\bar{x}_t$ (Eq. 4 in [6]) is given by

$$q(\bar{x}_t \mid \bar{x}_0) = \mathcal{N}\big(\bar{x}_t;\ \sqrt{\bar{\alpha}_t}\,\bar{x}_0,\ (1 - \bar{\alpha}_t)\,\mathbf{I}\big), \quad \text{where} \quad \bar{\alpha}_t = \prod_{s=1}^{t} (1 - \beta_s). \tag{233}$$

Ho et al. [6] define $\{\beta_t\}$ based on a linear schedule and then calculate the corresponding $\{\bar{\alpha}_t\}$ from Eq. 233. Alternatively, one can also define $\{\bar{\alpha}_t\}$ first and then solve for $\{\beta_t\}$:

$$\bar{\alpha}_t = \prod_{s=1}^{t} (1 - \beta_s) \tag{234}$$

$$\bar{\alpha}_t = \bar{\alpha}_{t-1} (1 - \beta_t) \tag{235}$$

$$\beta_t = 1 - \frac{\bar{\alpha}_t}{\bar{\alpha}_{t-1}}. \tag{236}$$

The improved DDPM formulation of Nichol and Dhariwal [13] employs a cosine schedule for $\bar{\alpha}_t$ (Eq. 17 in [13]), defined as

$$\bar{\alpha}_t = \frac{f(t)}{f(0)}, \quad \text{where} \quad f(t) = \cos^2\left(\frac{t/T + s}{1 + s} \cdot \frac{\pi}{2}\right), \tag{237}$$

where $s = 0.008$. In their implementation[15], however, Nichol et al. leave out the division by $f(0)$ and simply define[16]

$$\bar{\alpha}_t = \cos^2\left(\frac{t/T + s}{1 + s} \cdot \frac{\pi}{2}\right). \tag{238}$$

To prevent singularities near $t = T$, they also clamp $\beta_t$ to 0.999. We can express the clamping in terms of $\bar{\alpha}_t$ by utilizing Eq. 233 and Eq. 234:

$$\bar{\alpha}'_t = \prod_{s=1}^{t} \big(1 - [\beta'_s]\big) \tag{239}$$

$$= \prod_{s=1}^{t} \Big(1 - \min\big([\beta_s],\ 0.999\big)\Big) \tag{240}$$

$$= \prod_{s=1}^{t} \left(1 - \min\left(1 - \frac{\bar{\alpha}_s}{\bar{\alpha}_{s-1}},\ 0.999\right)\right) \tag{241}$$

$$= \prod_{s=1}^{t} \max\left(\frac{\bar{\alpha}_s}{\bar{\alpha}_{s-1}},\ 0.001\right). \tag{242}$$

Let us now reinterpret the above formulas in our unified framework. Recall from Table 1 that we denote the original iDDPM sampling steps by $\{u_j\}$ in the order of descending noise level $\sigma(u_j)$, where $j \in \{0, \dots, M\}$. To harmonize the notation of Eq. 233, Eq. 238, and Eq. 239, we thus have to replace $T \longrightarrow M$ and $t \longrightarrow M - j$:

$$q(\bar{x}_j \mid \bar{x}_M) = \mathcal{N}\big(\bar{x}_j;\ \sqrt{\bar{\alpha}'_j}\,\bar{x}_M,\ (1 - \bar{\alpha}'_j)\,\mathbf{I}\big), \tag{243}$$

$$\bar{\alpha}_j = \cos^2\left(\frac{(M - j)/M + C_2}{1 + C_2} \cdot \frac{\pi}{2}\right), \quad \text{and} \tag{244}$$

$$\bar{\alpha}'_j = \prod_{s=M-1}^{j} \max\left(\frac{\bar{\alpha}_j}{\bar{\alpha}_{j+1}},\ C_1\right) = \bar{\alpha}'_{j+1} \max\left(\frac{\bar{\alpha}_j}{\bar{\alpha}_{j+1}},\ C_1\right), \tag{245}$$

where the constants are $C_1 = 0.001$ and $C_2 = 0.008$.

---

[15] https://github.com/openai/improved-diffusion

[16] https://github.com/openai/improved-diffusion/blob/783b6740edb79fdb7d063250db2c51cc9545dcd1/improved_diffusion/gaussian_diffusion.py#L39

We can further simplify Eq. 244:

$$\bar{\alpha}_j \;=\; \cos^2\left(\frac{(M-j)/M+C_2}{1+C_2}\cdot\frac{\pi}{2}\right) \tag{246}$$

$$=\; \cos^2\left(\frac{\pi}{2}\,\frac{(1+C_2)-j/M}{1+C_2}\right) \tag{247}$$

$$=\; \cos^2\left(\frac{\pi}{2}-\frac{\pi}{2}\,\frac{j}{M(1+C_2)}\right) \tag{248}$$

$$=\; \sin^2\left(\frac{\pi}{2}\,\frac{j}{M(1+C_2)}\right), \tag{249}$$

giving the formula shown in the "Parameters" section of Table 1.

To harmonize the definitions of $\boldsymbol{x}$ and $\bar{\boldsymbol{x}}$, we must match the perturbation kernel of Eq. 11 with the transition probability of Eq. 243 for each time step $t = u_j$:

$$p_{0t}\big(\boldsymbol{x}(u_j)\mid\boldsymbol{x}(0)\big) \;=\; q(\bar{\boldsymbol{x}}_j\mid\bar{\boldsymbol{x}}_M) \tag{250}$$

$$\mathcal{N}\big(\boldsymbol{x}(u_j);\; s(t)\,\boldsymbol{x}(0),\; s(u_j)^2\,\sigma(u_j)^2\,\mathbf{I}\big) \;=\; \mathcal{N}\left(\bar{\boldsymbol{x}}_j;\; \sqrt{\bar{\alpha}'_j}\,\bar{\boldsymbol{x}}_M,\; \big(1-\bar{\alpha}'_j\big)\,\mathbf{I}\right). \tag{251}$$

Substituting $s(t) = 1$ and $\sigma(t) = t$ from Appendix C.3.1, as well as $\bar{\boldsymbol{x}}_M = \boldsymbol{x}(0)$:

$$\mathcal{N}\big(\boldsymbol{x}(u_j);\; \boldsymbol{x}(0),\; u_j^2\,\mathbf{I}\big) = \mathcal{N}\left(\bar{\boldsymbol{x}}_j;\; \sqrt{\bar{\alpha}'_j}\,\boldsymbol{x}(0),\; \big(1-\bar{\alpha}'_j\big)\,\mathbf{I}\right). \tag{252}$$

We can match the means of these two distributions by defining $\bar{\boldsymbol{x}}_j = \sqrt{\bar{\alpha}'_j}\,\boldsymbol{x}(u_j)$:

$$\mathcal{N}\big(\boldsymbol{x}(u_j);\; \boldsymbol{x}(0),\; u_j^2\,\mathbf{I}\big) \;=\; \mathcal{N}\left(\sqrt{\bar{\alpha}'_j}\,\boldsymbol{x}(u_j);\; \sqrt{\bar{\alpha}'_j}\,\boldsymbol{x}(0),\; \big(1-\bar{\alpha}'_j\big)\,\mathbf{I}\right) \tag{253}$$

$$=\; \mathcal{N}\left(\boldsymbol{x}(u_j);\; \boldsymbol{x}(0),\; \frac{1-\bar{\alpha}'_j}{\bar{\alpha}'_j}\,\mathbf{I}\right). \tag{254}$$

Matching the variances and solving for $\bar{\alpha}'_j$ gives

$$u_j^2 \;=\; (1-\bar{\alpha}'_j)\,/\,\bar{\alpha}'_j \tag{255}$$

$$u_j^2\,\bar{\alpha}'_j \;=\; 1-\bar{\alpha}'_j \tag{256}$$

$$u_j^2\,\bar{\alpha}'_j + \bar{\alpha}'_j \;=\; 1 \tag{257}$$

$$(u_j^2+1)\,\bar{\alpha}'_j \;=\; 1 \tag{258}$$

$$\bar{\alpha}'_j \;=\; 1\,/\,(u_j^2+1). \tag{259}$$

Finally, we can expand the left-hand side using Eq. 245 and solve for $u_{j-1}$:

$$\bar{\alpha}'_{j+1}\,\max(\bar{\alpha}_j/\bar{\alpha}_{j+1},\, C_1) \;=\; 1\,/\,(u_j^2+1) \tag{260}$$

$$\bar{\alpha}'_j\,\max(\bar{\alpha}_{j-1}/\bar{\alpha}_j,\, C_1) \;=\; 1\,/\,(u_{j-1}^2+1) \tag{261}$$

$$\big[1\,/\,(u_j^2+1)\big]\,\max(\bar{\alpha}_{j-1}/\bar{\alpha}_j,\, C_1) \;=\; 1\,/\,(u_{j-1}^2+1) \tag{262}$$

$$\max(\bar{\alpha}_{j-1}/\bar{\alpha}_j,\, C_1)\,(u_{j-1}^2+1) \;=\; u_j^2+1 \tag{263}$$

$$u_{j-1}^2+1 \;=\; (u_j^2+1)\,/\,\max(\bar{\alpha}_{j-1}/\bar{\alpha}_j,\, C_1) \tag{264}$$

$$u_{j-1} \;=\; \sqrt{\frac{u_j^2+1}{\max(\bar{\alpha}_{j-1}/\bar{\alpha}_j,\, C_1)} - 1}, \tag{265}$$

giving a recurrence formula for $\{u_j\}$, bootstrapped by $u_M = 0$, that matches the "Time steps" row of Table 1.

### C.3.3 iDDPM preconditioning and training

We can solve $D_\theta(\cdot)$ from Eq. 227 by substituting $\sigma(t) = t$ from Appendix C.3.1:

$$\epsilon_\theta^{(j)}\left(\boldsymbol{x}/\sqrt{\sigma^2+1}\right) = \left(\boldsymbol{x} - D_\theta(\boldsymbol{x};\sigma)\right)/\sigma \tag{266}$$

$$D_\theta(\boldsymbol{x};\sigma) = \boldsymbol{x} - \sigma\,\epsilon_\theta^{(j)}\left(\boldsymbol{x}/\sqrt{\sigma^2+1}\right). \tag{267}$$

We choose to define $F_\theta(\cdot; j) = \epsilon_\theta^{(j)}(\cdot)$ and solve $j$ from $\sigma$ by finding the nearest $u_j$:

$$D_\theta(\boldsymbol{x};\sigma) = \underbrace{1\cdot}_{c_{\text{skip}}}\boldsymbol{x} \underbrace{-\sigma}_{c_{\text{out}}} \cdot F_\theta\Big(\underbrace{\tfrac{1}{\sqrt{\sigma^2+1}}\cdot\boldsymbol{x}}_{c_{\text{in}}};\ \underbrace{\arg\min_j |u_j - \sigma|}_{c_{\text{noise}}}\Big), \tag{268}$$

where $c_{\text{skip}}$, $c_{\text{out}}$, $c_{\text{in}}$, and $c_{\text{noise}}$ match the "Network and preconditioning" section of Table 1.

Note that Eq. 268 is identical to the VP preconditioning formula in Eq. 181. Furthermore, Nichol and Dhariwal [13] define their main training loss $L_{\text{simple}}$ (Eq. 14 in [13]) the same way as Song et al. [15], with $\sigma$ drawn uniformly from $\{u_j\}$. Thus, we can reuse Eq. 190 with $\sigma = u_j$, $j \sim \mathcal{U}(0, M-1)$, and $\lambda(\sigma) = 1/\sigma^2$, matching the "Training" section of Table 1. In addition to $L_{\text{simple}}$, Nichol and Dhariwal [13] also employ a secondary loss term $L_{\text{vlb}}$; we refer the reader to Section 3.1 in [13] for details.

### C.3.4 iDDPM practical considerations

The pre-trained iDDPM model that we use on ImageNet-64 corresponds to the "ADM (dropout)" checkpoint[17] provided by Dhariwal and Nichol [4]. It contains 296 million trainable parameters and supports a discrete set of $M = 1000$ noise levels $\sigma \in \{u_j\} \approx \{20291, 642, 321, 214, 160, 128, 106, 92, 80, 71, \ldots, 0.0064\}$. The fact that we can only evaluate $F_\theta$ these specific choices of $\sigma$ presents three practical challenges:

1. In the context of DDIM, we must choose how to resample $\{u_j\}$ to yield $\{t_i\}$ for $N \neq M$. Song et al. [14] employ a simple resampling scheme where $t_i = u_{k\cdot i}$ for resampling factor $k \in \mathbb{Z}^+$. This scheme, however, requires that $1000 \equiv 0 \pmod{N}$, which limits the possible choices for $N$ considerably. Nichol and Dhariwal [13], on the other hand, employ a more flexible scheme where $t_i = u_j$ with $j = \lfloor (M-1)/(N-1)\cdot i \rfloor$. We note, however, that in practice the values of $u_{j<8}$ are considerably larger than our preferred $\sigma_{\max} = 80$. We choose to skip these values by defining $j = \lfloor j_0 + (M-1-j_0)/(N-1)\cdot i \rfloor$ with $j_0 = 8$, matching the "Time steps" row in Table 1. In Figure 2c, the differences between the original sampler (blue) and our reimplementation (orange) are explained by this choice.

2. In the context of our time step discretization (Eq. 5), we must ensure that $\sigma_i \in \{u_j\}$. We accomplish this by rounding each $\sigma_i$ to its nearest supported counterpart, i.e., $\sigma_i \leftarrow u_{\arg\min_j |u_j - \sigma_i|}$, and setting $\sigma_{\min} = 0.0064 \approx u_{N-1}$. This is sufficient, because Algorithm 1 only evaluates $D_\theta(\cdot; \sigma)$ with $\sigma \in \{\sigma_{i<N}\}$.

3. In the context of our stochastic sampler, we must ensure that $\hat{t}_i \in \{u_j\}$. We accomplish this by replacing line 5 of Algorithm 2 with $\hat{t}_i \leftarrow u_{\arg\min_j |u_j - (t_i+\gamma_i t_i)|}$.

With these changes, we are able to import the pre-trained model directly as $F_\theta(\cdot)$ and run Algorithms 1 and 2 using the definitions in Table 1. Note that the model outputs both $\epsilon_\theta(\cdot)$ and $\Sigma_\theta(\cdot)$, as described in Section 3.1 of [13]; we use only the former and ignore the latter.

## D  Further analysis of deterministic sampling

### D.1  Truncation error analysis and choice of discretization parameters

As discussed in Section 3, the fundamental reason why diffusion models tend to require a large number of sampling steps is that any numerical ODE solver is necessarily an approximation; the

---

[17]https://openaipublic.blob.core.windows.net/diffusion/jul-2021/64x64_diffusion.pt

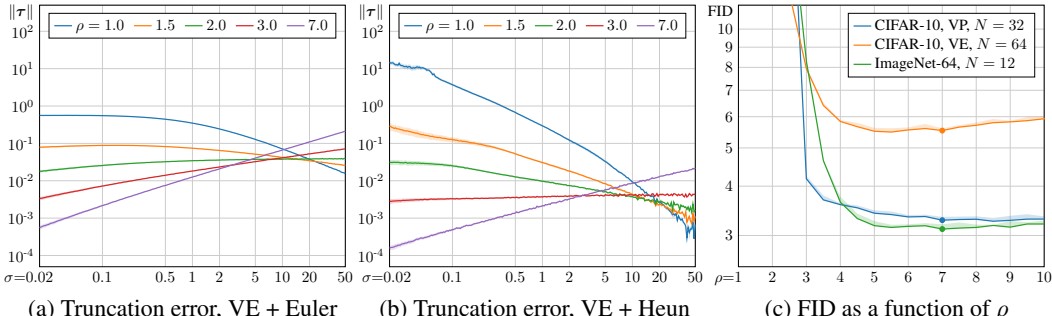

|           |                           |
| :-------: | :-----------------------: |
| (a) Truncation error, VE + Euler | (b) Truncation error, VE + Heun | (c) FID as a function of $\rho$ |

Figure 13: **(a)** Local truncation error ($y$-axis) at different noise levels ($x$-axis) using Euler's method with the VE-based CIFAR-10 model. Each curve corresponds to a different time step discretization, defined for $N = 64$ and a specific choice for the polynomial exponent $\rho$. The values represent the root mean square error (RMSE) between one Euler iteration and a sequence of multiple smaller Euler iterations, representing the ground truth. The shaded regions, barely visible at low $\sigma$, represent standard deviation over different latents $\boldsymbol{x}_0$. **(b)** Corresponding error curves for Heun's 2$^{\text{nd}}$ order method (Algorithm 1). **(c)** FID ($y$-axis) as a function of the polynomial exponent ($x$-axis) for different models, measured using Heun's 2$^{\text{nd}}$ order method. The shaded regions indicate the range of variation between the lowest and highest observed FID, and the dots indicate the value of $\rho$ that we use in all other experiments.

larger the steps, the farther away we drift from the true solution at each step. Specifically, given the value of $\boldsymbol{x}_{i-1}$ at time step $i - 1$, the solver approximates the true $\boldsymbol{x}_i^*$ as $\boldsymbol{x}_i$, resulting in local truncation error $\boldsymbol{\tau}_i = \boldsymbol{x}_i^* - \boldsymbol{x}_i$. The local errors get accumulated over the $N$ steps, ultimately leading to global truncation error $\boldsymbol{e}_N$.

Euler's method is a first order ODE solver, meaning that $\boldsymbol{\tau}_i = \mathcal{O}\left(h_i^2\right)$ for any sufficiently smooth $\boldsymbol{x}(t)$, where $h_i = |t_i - t_{i-1}|$ is the local step size [16]. In other words, there exist some $C$ and $H$ such that $||\boldsymbol{\tau}_i|| < Ch_i^2$ for every $h_i < H$, i.e., halving $h_i$ reduces $\boldsymbol{\tau}_i$ by $4\times$. Furthermore, if we assume that $D_\theta$ is Lipschitz continuous — which is true for all network architectures considered in this paper — the global truncation error is bounded by $||\boldsymbol{e}_N|| \leq E \max_i ||\boldsymbol{\tau}_i||$, where the value of $E$ depends on $N$, $t_0$, $t_N$, and the Lipschitz constant [16]. Thus, reducing the global error for given $N$, which in turn enables reducing $N$ itself, boils down to choosing the solver and $\{t_i\}$ so that $\max_i ||\boldsymbol{\tau}_i||$ is minimized.

To gain insight on how the local truncation error behaves in practice, we measure the values of $\boldsymbol{\tau}_i$ over different noise levels using the VE-based CIFAR-10 model. For a given noise level, we set $t_i = \sigma^{-1}(\sigma_i)$ and choose some $t_{i-1} > t_i$ depending on the case. We then sample $\boldsymbol{x}_{i-1}$ from $p(\boldsymbol{x}; \sigma_{i-1})$ and estimate the true $\boldsymbol{x}_i^*$ by performing 200 Euler steps over uniformly selected subintervals between $t_{i-1}$ and $t$. Finally, we plot the mean and standard deviation of the root mean square error (RMSE), i.e., $||\boldsymbol{\tau}_i||/\sqrt{\dim \boldsymbol{\tau}}$, as a function of $\sigma_i$, averaged over 200 random samples of $\boldsymbol{x}_{i-1}$. Results for Euler's method are shown in Figure 13a, where the blue curve corresponds to uniform step size $h_\sigma = 1.25$ with respect to $\sigma$, i.e., $\sigma_{i-1} = \sigma_i + h_\sigma$ and $t_{i-1} = \sigma^{-1}(\sigma_{i-1})$. We see that the error is very large (RMSE $\approx 0.56$) for low noise levels ($\sigma_i \leq 0.5$) and considerably smaller for high noise levels. This is in line with the common intuition that, in order to reduce $\boldsymbol{e}_N$, the step size should be decreased monotonically with decreasing $\sigma$. Each curve is surrounded by a shaded region that indicates standard deviation, barely visible at low values of $\sigma$. This indicates that $\boldsymbol{\tau}_i$ is nearly constant with respect to $\boldsymbol{x}_{i-1}$, and thus there would be no benefit in varying $\{t_i\}$ schedule on a per-sample basis.

A convenient way to vary the local step size depending on the noise level is to define $\{\sigma_i\}$ as a linear resampling of some monotonically increasing, unbounded warp function $w(z)$. In other words, $\sigma_{i<N} = w(Ai + B)$ and $\sigma_N = 0$, where constants $A$ and $B$ are selected so that $\sigma_0 = \sigma_{\max}$ and $\sigma_{N-1} = \sigma_{\min}$. In practice, we set $\sigma_{\min} = \max(\sigma_{\text{lo}}, 0.002)$ and $\sigma_{\max} = \min(\sigma_{\text{hi}}, 80)$, where $\sigma_{\text{lo}}$ and $\sigma_{\text{hi}}$ are the lowest and highest noise levels supported by a given model, respectively; we have found these choices to perform reasonably well in practice. Now, to balance $\boldsymbol{\tau}_i$ between low and high noise levels, we can, for example, use a polynomial warp function $w(z) = z^\rho$ parameterized by the

exponent $\rho$. This choice leads to the following formula for $\{\sigma_i\}$:

$$\sigma_{i<N} = \left(\sigma_{\max}^{\frac{1}{\rho}} + \frac{i}{N-1}\left(\sigma_{\min}^{\frac{1}{\rho}} - \sigma_{\max}^{\frac{1}{\rho}}\right)\right)^{\rho}, \sigma_N = 0, \tag{269}$$

which reduces to uniform discretization when $\rho = 1$ and gives more and more emphasis to low noise levels as $\rho$ increases.[18]

Based on the value of $\sigma_i$, we can now compute $\sigma_{i-1} = \left(\sigma_i^{1/\rho} - A\right)^{\rho}$, which enables us to visualize $\boldsymbol{\tau}_i$ for different choices of $\rho$ in Figure 13a. We see that increasing $\rho$ reduces the error for low noise levels ($\sigma < 10$) while increasing it for high noise levels ($\sigma > 10$). Approximate balance is achieved at $\rho = 2$, but RMSE remains relatively high ($\sim 0.03$), meaning that Euler's method drifts away from the correct result by several ULPs at each step. While the error could be reduced by increasing $N$, we would ideally like the RMSE to be well below 0.01 even with low step counts.

Heun's method introduces an additional correction step for $\boldsymbol{x}_{i+1}$ to account for the fact that $\mathrm{d}\boldsymbol{x}/\mathrm{d}t$ may change between $t_i$ and $t_{i+1}$; Euler's method assumes it to be constant. The correction leads to cubic convergence of the local truncation error, i.e., $\boldsymbol{\tau}_i = \mathcal{O}\left(h_i^3\right)$, at the cost of one additional evaluation of $D_\theta$ per step. We discuss the general family of Heun-like schemes later in Appendix D.2. Figure 13b shows local truncation error for Heun's method using the same setup as Figure 13a. We see that the differences in $||\boldsymbol{\tau}_i||$ are generally more pronounced, which is to be expected given the quadratic vs. cubic convergence of the two methods. Cases where Euler's method has low RMSE tend to have even lower RMSE with Heun's method, and vice versa for cases with high RMSE. Most remarkably, the red curve shows almost constant RMSE $\in [0.0030, 0.0045]$. This means that the combination of Eq. 269 and Heun's method is, in fact, very close to optimal with $\rho = 3$.

Thus far, we have only considered the raw numerical error, i.e., component-wise deviation from the true result in RGB space. The raw numerical error is relevant for certain use cases, e.g., image manipulation where the ODE is first evaluated in the direction of increasing $t$ and then back to $t = 0$ again — in this case, $||\boldsymbol{e}_N||$ directly tells us how much the original image degrades in the process and we can use $\rho = 3$ to minimize it. Considering the generation of novel images from scratch, however, it is reasonable to expect different noise levels to introduce different kinds of errors that may not necessarily be on equal footing considering their perceptual importance. We investigate this in Figure 13c, where we plot FID as a function of $\rho$ for different models and different choices of $N$. Note that the ImageNet-64 model was only trained for a discrete set of noise levels; in order to use it with Eq. 269, we round each $t_i$ to its nearest supported counterpart, i.e., $t_i' = u_{\arg\min_j |u_j - t_i|}$.

From the plot, we can see that even though $\rho = 3$ leads to relatively good FID, it can be reduced further by choosing $\rho > 3$. This corresponds to intentionally introducing error at high noise levels to reduce it at low noise levels, which makes intuitive sense because the value of $\sigma_{\max}$ is somewhat arbitrary to begin with — increasing $\sigma_{\max}$ can have a large impact on $||\boldsymbol{e}_N||$, but it does not affect the resulting image distribution nearly as much. In general, we have found $\rho = 7$ to perform reasonably well in all cases, and use this value in all other experiments.

## D.2 General family of 2$^{\text{nd}}$ order Runge–Kutta variants

Heun's method illustrated in Algorithm 1 belongs to a family of explicit two-stage 2$^{\text{nd}}$ order Runge–Kutta methods, each having the same computational cost. A common parameterization [16] of this family is,

$$\boldsymbol{d}_i = f(\boldsymbol{x}_i; t_i) \quad ; \quad \boldsymbol{x}_{i+1} = \boldsymbol{x}_i + h\left[\left(1 - \frac{1}{2\alpha}\right)\boldsymbol{d}_i + \frac{1}{2\alpha}f(\boldsymbol{x}_i + \alpha h\boldsymbol{d}_i; t_i + \alpha h)\right], \tag{270}$$

where $h = t_{i+1} - t_i$ and $\alpha$ is a parameter that controls where the additional gradient is evaluated and how much it influences the step taken. Setting $\alpha = 1$ corresponds to Heun's method, and $\alpha = \frac{1}{2}$ and $\alpha = \frac{2}{3}$ yield so-called midpoint and Ralston methods, respectively. All these variants differ in the kind of approximation error they incur due to the geometry of the underlying function $f$.

To establish the optimal $\alpha$ in our use case, we ran a separate series of experiments. According to the results, it appears that $\alpha = 1$ is very close to being optimal. Nonetheless, the experimentally

---

[18]In the limit, Eq. 269 reduces to the same geometric sequence employed by original VE ODE when $\rho \to \infty$. Thus, our discretization can be seen as a parametric generalization of the one proposed by Song et al. [15].

**Algorithm 3** Deterministic sampling using general 2$^{\text{nd}}$ order Runge–Kutta, $\sigma(t) = t$ and $s(t) = 1$.

---

1: **procedure** ALPHASAMPLER($D_\theta(\boldsymbol{x}; \sigma)$, $t_{i \in \{0, \ldots, N\}}$, $\alpha$)
2:      **sample** $\boldsymbol{x}_0 \sim \mathcal{N}(\boldsymbol{0},\, t_0^2\, \mathbf{I})$
3:      **for** $i \in \{0, \ldots, N-1\}$ **do**
4:          $h_i \leftarrow t_{i+1} - t_i$                                         ▷ Step length
5:          $\boldsymbol{d}_i \leftarrow \big(\boldsymbol{x}_i - D_\theta(\boldsymbol{x}_i; t_i)\big)/t_i$              ▷ Evaluate $d\boldsymbol{x}/dt$ at $(\boldsymbol{x}, t_i)$
6:          $(\boldsymbol{x}_i', t_i') \leftarrow (\boldsymbol{x}_i + \alpha h \boldsymbol{d}_i, t_i + \alpha h)$            ▷ Additional evaluation point
7:          **if** $t_i' \neq 0$ **then**
8:              $\boldsymbol{d}_i' \leftarrow \big(\boldsymbol{x}_i' - D_\theta(\boldsymbol{x}_i'; t_i')\big)/t_i'$           ▷ Evaluate $d\boldsymbol{x}/dt$ at $(\boldsymbol{x}_i', t_i')$
9:              $\boldsymbol{x}_{i+1} \leftarrow \boldsymbol{x}_i + h\Big[\Big(1 - \frac{1}{2\alpha}\Big)\boldsymbol{d}_i + \frac{1}{2\alpha}\boldsymbol{d}_i'\Big]$     ▷ Second order step from $t_i$ to $t_{i+1}$
10:         **else**
11:             $\boldsymbol{x}_{i+1} \leftarrow \boldsymbol{x}_i + h\boldsymbol{d}_i$                   ▷ Euler step from $t_i$ to $t_{i+1}$
12:      **return** $\boldsymbol{x}_N$

---

best choice was $\alpha = 1.1$ that performed slightly better, even though values greater than one are theoretically hard to justify as they overshoot the target $t_{i+1}$. As we have no good explanation for this observation and cannot tell if it holds in general, we chose not to make $\alpha$ a new hyperparameter and instead fixed it to 1, corresponding exactly to Heun's method. Further analysis is left as future work, including the possibility of having $\alpha$ vary during sampling.

An additional benefit of setting $\alpha = 1$ is that it makes it possible to use pre-trained neural networks $D_\theta(\boldsymbol{x}; \sigma)$ that have been trained only for specific values of $\sigma$. This is because a Heun step evaluates the additional gradient at exactly $t_{i+1}$ unlike the other 2$^{\text{nd}}$ order variants. Hence it is sufficient to ensure that each $t_i$ corresponds to a value of $\sigma$ that the network was trained for.

Algorithm 3 shows the pseudocode for a general 2$^{\text{nd}}$ order solver parameterized by $\alpha$. For clarity, the pseudocode assumes the specific choices of $\sigma(t) = t$ and $s(t) = 1$ that we advocate in Section 3. Note that the fallback to Euler step (line 11) can occur only when $\alpha \geq 1$.

# E    Further results with stochastic sampling

## E.1    Image degradation due to excessive stochastic iteration

Figure 14 illustrates the image degradation caused by excessive Langevin iteration (Section 4, "Practical considerations"). These images are generated by doing a specified number of iterations at a fixed noise level $\sigma$ so that at each iteration an equal amount of noise is added and removed. In theory, Langevin dynamics should bring the distribution towards the ideal distribution $p(\boldsymbol{x}; \sigma)$ but as noted in Section 4, this holds only if the denoiser $D_\theta(\boldsymbol{x}; \sigma)$ induces a conservative vector field in Eq. 3.

As seen in the figure, it is clear that the image distribution suffers from repeated iteration in all cases, although the exact failure mode depends on dataset and noise level. For low noise levels (below 0.2 or so), the images tend to oversaturate starting at 2k iterations and become fully corrupted after that. Our heuristic of setting $S_{\text{tmin}} > 0$ is designed to prevent stochastic sampling altogether at very low noise levels to avoid this effect.

For high noise levels, we can see that iterating without the standard deviation correction, i.e., when $S_{\text{noise}} = 1.000$, the images tend to become more abstract and devoid of color at high iteration counts; this is especially visible in the 10k column of CIFAR-10 where the images become mostly black and white with no discernible backgrounds. Our heuristic inflation of standard deviation by setting $S_{\text{noise}} > 1$ counteracts this tendency efficiently, as seen in the corresponding images on the right hand side of the figure. Notably, this still does not fix the oversaturation and corruption at low noise levels, suggesting multiple sources for the detrimental effects of excessive iteration. Further research will be required to better understand the root causes of these observed effects.

Figure 15 presents the output quality of our stochastic sampler in terms of FID as a function of $S_{\text{churn}}$ at fixed NFE, using pre-trained networks of Song et al. [15] and Dhariwal and Nichol [4]. Generally, for each case and combination of our heuristic corrections, there is an optimal amount of stochasticity after which the results start to degrade. It can also be seen that regardless of the value of $S_{\text{churn}}$, the

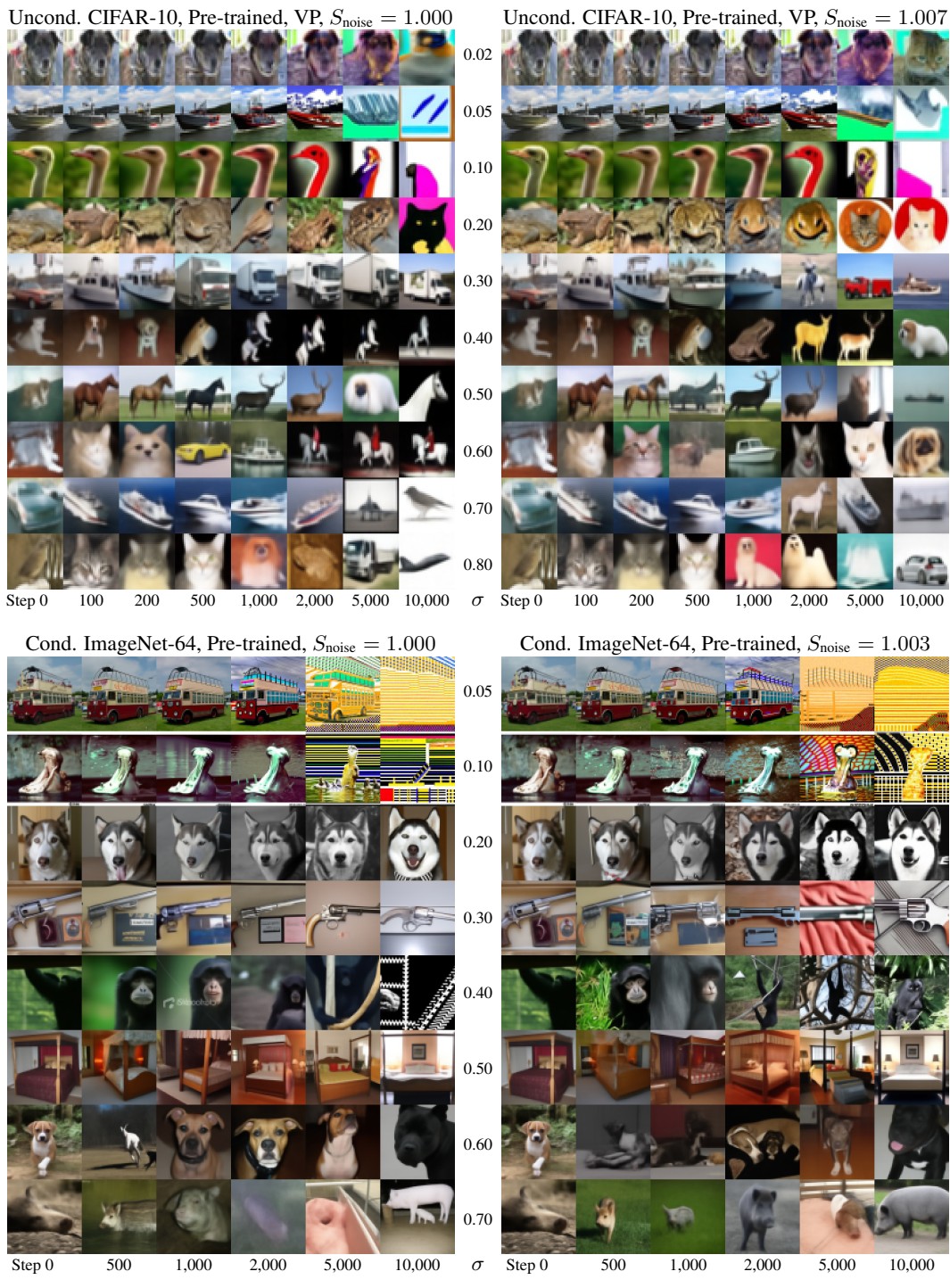

Figure 14: Gradual image degradation with repeated addition and removal of noise. We start with a random image drawn from $p(\boldsymbol{x}; \sigma)$ (first column) and run Algorithm 2 for a certain number of steps (remaining columns) with fixed $\gamma_i = \sqrt{2} - 1$. Each row corresponds to a specific choice of $\sigma$ (indicated in the middle) that we keep fixed throughout the entire process. We visualize the results after running them through the denoiser, i.e., $D_\theta(\boldsymbol{x}_i; \sigma)$.

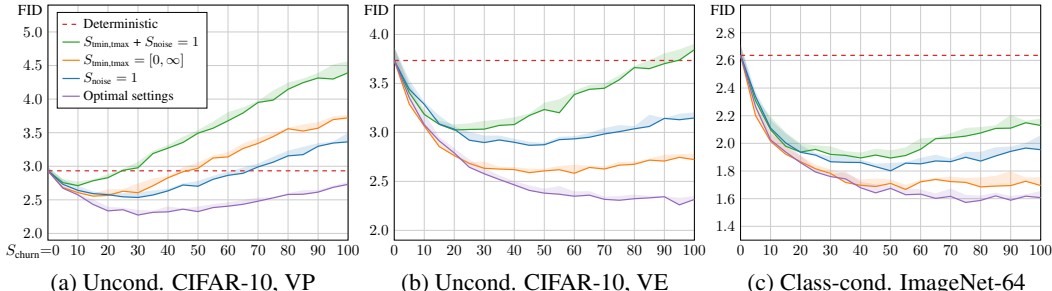

| (a) Uncond. CIFAR-10, VP | (b) Uncond. CIFAR-10, VE | (c) Class-cond. ImageNet-64 |

Figure 15: Ablations of our stochastic sampler (Algorithm 2) parameters using pre-trained networks of Song et al. [15] and Dhariwal and Nichol [4]. Each curve shows FID ($y$-axis) as a function of $S_{\text{churn}}$ ($x$-axis) for $N = 256$ steps (NFE = 511). The dashed red lines correspond to our deterministic sampler (Algorithm 1), equivalent to setting $S_{\text{churn}} = 0$. The purple curves correspond to optimal choices for $\{S_{\text{tmin}}, S_{\text{tmax}}, S_{\text{noise}}\}$, found separately for each case using grid search. Orange, blue, and green correspond to disabling the effects of $S_{\text{tmin,tmax}}$ and/or $S_{\text{noise}}$. The shaded regions indicate the range of variation between the lowest and highest observed FID.

Table 5: Parameters used for the stochastic sampling experiments in Section 4.

| Parameter | CIFAR-10 | | ImageNet | | Grid search |
|---|---|---|---|---|---|
| | VP | VE | Pre-trained | Our model | |
| $S_{\text{churn}}$ | 30 | 80 | 80 | 40 | 0, 10, 20, 30, ..., 70, 80, 90, 100 |
| $S_{\text{tmin}}$ | 0.01 | 0.05 | 0.05 | 0.05 | 0, 0.005, 0.01, 0.02, ..., 1, 2, 5, 10 |
| $S_{\text{tmax}}$ | 1 | 1 | 50 | 50 | 0.2, 0.5, 1, 2, ..., 10, 20, 50, 80 |
| $S_{\text{noise}}$ | 1.007 | 1.007 | 1.003 | 1.003 | 1.000, 1.001, ..., 1.009, 1.010 |

best results are obtained by enabling all corrections, although whether $S_{\text{noise}}$ or $S_{\text{tmin,tmax}}$ is more important depends on the case.

### E.2 Stochastic sampling parameters

Table 5 lists the values for $S_{\text{churn}}$, $S_{\text{tmin}}$, $S_{\text{tmax}}$, and $S_{\text{noise}}$ that we used in our stochastic sampling experiments. These were determined with a grid search over the combinations listed in the rightmost column. It can be seen that the optimal parameters depend on the case; better understanding of the degradation phenomena will hopefully give rise to more direct ways of handling the problem in the future.

## F  Implementation details

We implemented our techniques in a newly written codebase, based loosely on the original implementations by Song et al.[19] [15], Dhariwal and Nichol[20] [4], and Karras et al.[21] [10]. We performed extensive testing to verify that our implementation produced exactly the same results as previous work, including samplers, pre-trained models, network architectures, training configurations, and evaluation. We ran all experiments using PyTorch 1.10.0, CUDA 11.4, and CuDNN 8.2.0 on NVIDIA DGX-1's with 8 Tesla V100 GPUs each.

Our implementation and pre-trained models are available at `https://github.com/NVlabs/edm`

---

[19]`https://github.com/yang-song/score_sde_pytorch`
[20]`https://github.com/openai/guided-diffusion`
[21]`https://github.com/NVlabs/stylegan3`

Table 6: Our augmentation pipeline. Each training image undergoes a combined geometric transformation based on 8 random parameters that receive non-zero values with a certain probability. The model is conditioned with an additional 9-dimensional input vector derived from these parameters.

| Augmentation | Transformation | Parameters | Prob. | Conditioning | Constants |
|---|---|---|---|---|---|
| $x$-flip | SCALE2D$(1 - 2a_0,\ 1)$ | $a_0 \sim \mathcal{U}\{0,1\}$ | 100% | $a_0$ | $A_{\text{prob}} = 12\%$ |
| $y$-flip | SCALE2D$(1,\ 1 - 2a_1)$ | $a_1 \sim \mathcal{U}\{0,1\}$ | $A_{\text{prob}}$ | $a_1$ | or 15% |
| Scaling | SCALE2D$\big((A_{\text{scale}})^{a_2},$ $(A_{\text{scale}})^{a_2}\big)$ | $a_2 \sim \mathcal{N}(0,1)$ | $A_{\text{prob}}$ | $a_2$ | $A_{\text{scale}} = 2^{0.2}$ |
| Rotation | ROTATE2D$(-a_3)$ | $a_3 \sim \mathcal{U}(-\pi, \pi)$ | $A_{\text{prob}}$ | $\cos a_3 - 1$ $\sin a_3$ | |
| Anisotropy | ROTATE2D$(a_4)$ SCALE2D$\big((A_{\text{aniso}})^{a_5},$ $1/(A_{\text{aniso}})^{a_5}\big)$ ROTATE2D$(-a_4)$ | $a_4 \sim \mathcal{U}(-\pi, \pi)$ $a_5 \sim \mathcal{N}(0,1)$ | $A_{\text{prob}}$ | $a_5\ \cos a_4$ $a_5\ \sin a_4$ | $A_{\text{aniso}} = 2^{0.2}$ |
| Translation | TRANSLATE2D$\big((A_{\text{trans}})a_6,$ $(A_{\text{trans}})a_7\big)$ | $a_6 \sim \mathcal{N}(0,1)$ $a_7 \sim \mathcal{N}(0,1)$ | $A_{\text{prob}}$ | $a_6$ $a_7$ | $A_{\text{trans}} = 1/8$ |

## F.1 FID calculation

We calculate FID [5] between 50,000 generated images and all available real images, without any augmentation such as $x$-flips. We use the pre-trained Inception-v3 model provided with StyleGAN3[22] [10] that is, in turn, a direct PyTorch translation of the original TensorFlow-based model[23]. We have verified that our FID implementation produces identical results compared to Dhariwal and Nichol [4] and Karras et al. [10]. To reduce the impact of random variation, typically in the order of $\pm 2\%$, we compute FID three times in each experiment and report the minimum. We also highlight the difference between the highest and lowest achieved FID in Figures 4, 5b, 13c, and 15.

## F.2 Augmentation regularization

In Section 5, we propose to combat overfitting of $D_\theta$ using conditional augmentation. We build our augmentation pipeline around the same concepts that were originally proposed by Karras et al. [9] in the context of GANs. In practice, we employ a set of 6 geometric transformations; we have found other types of augmentations, such as color corruption and image-space filtering, to be consistently harmful for diffusion-based models.

The details of our augmentation pipeline are shown in Table 6. We apply the augmentations independently to each training image $\boldsymbol{y} \sim p_{\text{data}}$ prior to adding the noise $\boldsymbol{n} \sim \mathcal{N}(\mathbf{0}, \sigma^2 \mathbf{I})$. First, we determine whether to enable or disable each augmentation based on a weighted coin toss. The probability of enabling a given augmentation ("Prob." column) is fixed to 12% for CIFAR-10 and 15% for FFHQ and AFHQv2, except for $x$-flips that are always enabled. We then draw 8 random parameters from their corresponding distributions ("Parameters" column); if a given augmentation is disabled, we override the associated parameters with zero. Based on these, we construct a homogeneous 2D transformation matrix based on the parameters ("Transformation" column). This transformation is applied to the image using the implementation of [9] that employs $2\times$ supersampled high-quality Wavelet filters. Finally, we construct a 9-dimensional conditioning input vector ("Conditioning" column) and feed it to the denoiser network, in addition to the image and noise level inputs.

The role of the conditioning input is to present the network with a set of auxiliary tasks; in addition to the main task of modeling $p(\boldsymbol{x}; \sigma)$, we effectively ask the network to also model an infinite set of distributions $p(\boldsymbol{x}; \sigma, \boldsymbol{a})$ for each possible choice of the augmentation parameters $\boldsymbol{a}$. These auxiliary tasks provide the network with a large variety of unique training samples, preventing it from

---

[22] https://api.ngc.nvidia.com/v2/models/nvidia/research/stylegan3/versions/1/files/metrics/inception-2015-12-05.pkl

[23] http://download.tensorflow.org/models/image/imagenet/inception-2015-12-05.tgz

Table 7: Hyperparameters used for the training runs in Section 5.

| Hyperparameter | CIFAR-10 | | FFHQ & AFHQv2 | | ImagetNet |
| --- | --- | --- | --- | --- | --- |
| | Baseline | Ours | Baseline | Ours | Ours |
| Number of GPUs | 4 | 8 | 4 | 8 | 32 |
| Duration (Mimg) | 200 | 200 | 200 | 200 | 2500 |
| Minibatch size | 128 | 512 | 128 | 256 | 4096 |
| Gradient clipping | ✓ | – | ✓ | – | – |
| Mixed-precision (FP16) | – | – | – | – | ✓ |
| Learning rate $\times 10^4$ | 2 | 10 | 2 | 2 | 1 |
| LR ramp-up (Mimg) | 0.64 | 10 | 0.64 | 10 | 10 |
| EMA half-life (Mimg) | 0.89 / 0.9 (VP / VE) | 0.5 | 0.89 / 0.9 (VP / VE) | 0.5 | 50 |
| Dropout probability | 10% | 13% | 10% | 5% / 25% (FFHQ / AFHQ) | 10% |
| Channel multiplier | 128 | 128 | 128 | 128 | 192 |
| Channels per resolution | 1-2-2-2 | 2-2-2 | 1-1-2-2-2 | 1-2-2-2 | 1-2-3-4 |
| Dataset $x$-flips | ✓ | – | ✓ | – | – |
| Augment probability | – | 12% | – | 15% | – |

overfitting to any individual sample. Still, the auxiliary tasks appear to be beneficial for the main task; we speculate that this is because the denoising operation itself is similar for every choice of $\boldsymbol{a}$.

We have designed the conditioning input so that zero corresponds to the case where no augmentations were applied. During sampling, we simply set $\boldsymbol{a} = \boldsymbol{0}$ to obtain results consistent with the main task. We have not observed any leakage between the auxiliary tasks and the main task; the generated images exhibit no traces of out-of-domain geometric transformations even with $A_{\text{prob}} = 100\%$. In practice, this means that we are free to choose the constants $\{A_{\text{prob}}, A_{\text{scale}}, A_{\text{aniso}}, A_{\text{trans}}\}$ any way we like as long as the results improve. Horizontal flips serve as an interesting example. Most of the prior work augments the training set with random $x$-flips, which is beneficial for most datasets but has the downside that any text or logos may appear mirrored in the generated images. With our non-leaky augmentations, we get the same benefits without the downsides by executing the $x$-flip augmentation with 100% probability. Thus, we rely exclusively on our augmentation scheme and disable dataset $x$-flips to ensure that the generated images stay true to the original distribution.

### F.3 Training configurations

Table 7 shows the exact set of hyperparameters that we used in our training experiments reported in Section 5. We will first detail the configurations used with CIFAR-10, FFHQ, and AFHQv2, and then discuss the training of our improved ImageNet model.

Config A of Table 2 ("Baseline") corresponds to the original setup of Song et al. [15] for the two cases (VP and VE), and config F ("Ours") corresponds to our improved setup. We trained each model until a total of 200 million images had been drawn from the training set, abbreviated as "200 Mimg" in Table 7; this corresponds to a total of $\sim$400,000 training iterations using a batch size of 512. We saved a snapshot of the model every 2.5 million images and reported results for the snapshot that achieved the lowest FID according to our deterministic sampler with NFE = 35 or NFE = 79, depending on the resolution.

In config B, we re-adjust the basic hyperparameters to enable faster training and obtain a more meaningful point of comparison. Specifically, we increase the parallelism from 4 to 8 GPUs and batch size from 128 to 512 or 256, depending on the resolution. We also disable gradient clipping, i.e., forcing $\|\mathrm{d}\mathcal{L}(D_\theta)/\mathrm{d}\theta\|_2 \leq 1$, that we found to provide no benefit in practice. Furthermore, we increase the learning rate from 0.0002 to 0.001 for CIFAR-10, ramping it up during the first 10 million images, and standardize the half-life of the exponential moving average of $\theta$ to 0.5 million images. Finally, we adjust the dropout probability for each dataset as shown in Table 7 via a full grid search at 1% increments. Our total training time is approximately 2 days for CIFAR-10 at 32×32 resolution and 4 days for FFHQ and AFHQv2 at 64×64 resolution.

Table 8: Details of the network architectures used in this paper.

| Parameter | DDPM++ (VP) | NCSN++ (VE) | ADM (ImageNet) |
|---|---|---|---|
| Resampling filter | Box | Bilinear | Box |
| Noise embedding | Positional | Fourier | Positional |
| Skip connections in encoder | – | Residual | – |
| Skip connections in decoder | – | – | – |
| Residual blocks per resolution | 4 | 4 | 3 |
| Attention resolutions | {16} | {16} | {32, 16, 8} |
| Attention heads | 1 | 1 | 6-9-12 |
| Attention blocks in encoder | 4 | 4 | 9 |
| Attention blocks in decoder | 2 | 2 | 13 |

In config C, we improve the expressive power of the model by removing the 4×4 layers and doubling the capacity of the 16×16 layers instead; we found the former to mainly contribute to overfitting, whereas the latter were critical for obtaining high-quality results. The original models of Song et al. [15] employ 128 channels at 64×64 (where applicable) and 32×32, and 256 channels at 16×16, 8×8, and 4×4. We change these numbers to 128 channels at 64×64 (where applicable), and 256 channels at 32×32, 16×16, and 8×8. We abbreviate these counts in Table 7 as multiples of 128, listed from the highest resolution to the lowest. In practice, this rebalancing reduces the total number of trainable parameters slightly, resulting in ∼56 million parameters for each model at 32×32 resolution and ∼62 million parameters at 64×64 resolution.

In config D, we replace the original preconditioning with our improved formulas ("Network and preconditioning" section in Table 1). In config E, we do the same for the noise distribution and loss weighting ("Training" section in Table 1). Finally, in config F, we enable augmentation regularization as discussed in Appendix F.2. The other hyperparameters remain the same as in config C.

With ImageNet-64, it is necessary to train considerably longer compared to the other datasets in order to reach state-of-the-art results. To reduce the training time, we employed 32 NVIDIA Ampere GPUs (4 nodes) with a batch size of 4096 (128 per GPU) and utilized the high-performance Tensor Cores via mixed-precision FP16/FP32 training. In practice, we store the trainable parameters as FP32 but cast them to FP16 when evaluating $F_\theta$, except for the embedding and self-attention layers, where we found the limited exponent range of FP16 to occasionally lead to stability issues. We trained the model for two weeks, corresponding to ∼2500 million images drawn from the training set and ∼600,000 training iterations, using learning rate 0.0001, exponential moving average of 50 million images, and the same model architecture and dropout probability as Dhariwal and Nichol [4]. We did not find overfitting to be a concern, and thus chose to not employ augmentation regularization.

## F.4 Network architectures

As a result of our training improvements, the VP and VE cases become otherwise identical in config F except for the network architecture; VP employs the DDPM++ architecture while VE employs NCSN++, both of which were originally proposed by Song et al. [15]. These architectures correspond to relatively straightforward variations of the same U-net backbone with three differences, as illustrated in Table 8. First, DDPM++ employs box filter $[1, 1]$ for the upsampling and downsampling layers whereas NCSN++ employs bilinear filter $[1, 3, 3, 1]$. Second, DDPM++ inherits its positional encoding scheme for the noise level directly from DDPM [6] whereas NCSN++ replaces it with random Fourier features [18]. Third, NCSN++ incorporates additional residual skip connections from the input image to each block in the encoder, as explained in Appendix H of [15] ("progressive growing architectures").

For class conditioning and augmentation regularization, we extend the original DDPM++ and NCSN++ arhictectures by introducing two optional conditioning inputs alongside the noise level input. We represent class labels as one-hot encoded vectors that we first scale by $\sqrt{C}$, where $C$ is the total number of classes, and then feed through a fully-connected layer. For the augmentation parameters, we feed the conditioning inputs of Appendix F.2 through a fully-connected layer as-is.

We then combine the resulting feature vectors with the original noise level conditioning vector through elementwise addition.

For class-conditional ImageNet-64, we use the ADM architecture of Dhariwal and Nichol [4] with no changes. The model has a total of ∼296 million trainable parameters. As detailed in Tables 7 and 8, the most notable differences to DDPM++ include the use of a slightly shallower model (3 residual blocks per resolution instead of 4) with considerably more channels (e.g., 768 in the lowest resolution instead of 256), more self-attention layers interspersed throughout the network (22 instead of 6), and the use of multi-head attention (e.g., 12 heads in the lowest resolution). We feel that the precise impact of architectural choices remains an interesting question for future work.

### F.5  Licenses

Datasets:

- CIFAR-10 [12]:  MIT license
- FFHQ [11]:  Creative Commons BY-NC-SA 4.0 license
- AFHQv2 [2]:  Creative Commons BY-NC 4.0 license
- ImageNet [3]:  The license status is unclear

Pre-trained models:

- CIFAR-10 models by Song et al. [15]:  Apache V2.0 license
- ImageNet-64 model by Dhariwal and Nichol [4]:  MIT license
- Inception-v3 model by Szegedy et al. [17]:  Apache V2.0 license