# OpenReview forum: "Elucidating the Design Space of Diffusion-Based Generative Models"
_NeurIPS.cc/2022/Conference — NeurIPS 2022 Accept_

### Official Review · Reviewer_2v7C · 2022-06-23

**Rating:** 7
**Confidence:** 5
**Soundness:** 4 excellent
**Presentation:** 3 good
**Contribution:** 3 good

**Summary:**

The paper proposes to consider 1) network parameterization (preconditioning) 2) training and 3) sampling of diffusion models in separation. This allows for putting many existing works in a common framework and for a simplified investigation of the three aspects in isolation. Considering, for example, the sampling process in isolation, the authors show how to improve several pre-trained networks from the literature. Furthermore, making "optimal" choices for each of the three aspects leads to new state-of-the-art models on several image benchmark datasets. Additionally, the paper proposes to train diffusion models on augmented data, importantly conditioning the network on the augmentation process. Lastly, the paper shows that stochastic sampling outperforms deterministic sampling for "suboptimal models", whereas deterministic sampling is preferred for "well-trained models".

**Questions:**

Main questions:
* Figure 5(b): The figure is of very high value as it shows that stochasticity benefits/hurts suboptimal/well-trained models. I am, however, curious if the figure would look similar for more challenging datasets such as ImageNet. My suspicion is that for ImageNet a well-trained model may still be considerably different than the true data model and therefore I suspect that stochasticity would even improve a well-trained model. Could the authors comment on this?

* Network parameterization (preconditioning): The proposed network parameterization seems very similar to the v-diffusion parameterization introduced in PD. The denoiser in v-diffusion is $\alpha_t x_t - \sigma_t F_\theta(x_t, t)$, where $F_\theta$ is the neural network. Could the authors comment on connections/advantages of v-diffusion in the proposed network parameterization as well as loss function?

Extra (research) questions (for which I don't necessarily expect an answer):
* Do the authors believe that variance-learning diffusion models (as, for example, introduced in iDDPM) could also be unified with their approach?


**Strengths And Weaknesses:**

Strengths:
* The paper is extremely well-written and all claims are well-supported by experiments.
* The paper has a great impact on the current field of diffusion models: the independently proposed training recipes, network parameterizations (preconditioning), and sampling schemes can potentially be used in many future works.

Weaknesses:
* The paper lacks related work in certain places; to list only a few:
    * Model parameterization (preconditioning):
        * [PD] introduces v-diffusion parameterization
        * [LSGM, CLD] introduce mixed score parameterizations
    * Sampling:
        * [PNDM, DEIS] apply linear multistep methods to the Diffusion Model ODE (DEIS was just recently proposed, so missing this citation is not a big deal)
        * [LFS] learns efficient samplers
        * [PD, KD] accelerating sampling via distillation
* Otherwise, the paper has very little weaknesses; see Questions below.

[PD] - Progressive Distillation for Fast Sampling of Diffusion Models

[LSGM] - Score-based Generative Modeling in Latent Space

[CLD] - Score-Based Generative Modeling with Critically-Damped Langevin Diffusion

[PNDM] - Pseudo Numerical Methods for Diffusion Models on Manifolds

[DEIS] - Fast Sampling of Diffusion Models with Exponential Integrator

[LFS] - Learning Fast Samplers for Diffusion Models by Differentiating Through Sample Quality

[KD] - Knowledge Distillation in Iterative Generative Models for Improved Sampling Speed

---

> ### Author Response · Authors · 2022-08-02
> **Author Response**
>
> We are thankful for the pointers to additional previous work. We will amend the paper accordingly.
>
> Unfortunately we lacked the compute capacity to determine curves such as in Fig. 5(b) for ImageNet-64, as that would have required a training effort we could not afford prior to submission. It is our expectation too that ImageNet is diverse enough w.r.t. network capacity that stochasticity would be beneficial, and we are extremely interested in knowing if this is the case. We should have results soon, and plan to add those in the paper before the camera-ready deadline.
>
> Thank you for pointing out the connection to v-diffusion parameterization in [PD]. Indeed, they make an observation that also motivates our skip connection: at high noise levels, predicting the noise leads to arbitrarily large amplification of the network output, along with any errors it makes. Their proposed mixture prediction can be interpreted in our framework as a noise level dependent skip connection. We will acknowledge this connection in the camera-ready revision.
>
> We have not considered including variance learning in our model so far, but it could be an avenue for future development.

---

> > ### Comment · Reviewer_2v7C · 2022-08-04
> > **Thanks**
> >
> > Thank you for the response. I am looking forward to see Fig. 5(b) for ImageNet in the camera-ready version.

---

### Official Review · Reviewer_XT6R · 2022-07-11

**Rating:** 8
**Confidence:** 4
**Soundness:** 4 excellent
**Presentation:** 4 excellent
**Contribution:** 4 excellent

**Summary:**

In this paper, the authors explore a wide variety of design choices made when designing diffusion models, both those which are typically given attention explicitly in prior work (e.g. the noise schedule) and ones which have been more implicitly agreed upon (e.g. the choice of of ODE solver). They highlight a set of potential sources of generalization to improve sampling - e.g., the lack of need for the sampling process to correspond to model or training details. They generalize a set of the stochastic diffusion model SDE with a single equation (Eq. 6) corresponding to moving forward and backward in time, and propose an algorithm to reduce stochastic sampling errors. In essence, prior works have explored relatively constrained subspaces of the decisions which define a particular diffusion model - this paper attempts to unify these decisions through a shared framework, allowing for new non-obvious generalizations, and then to quantitatively understand their optima.

**Questions:**

What kinds of diversity/fidelity tradeoffs are implied by the various suggestions of this paper?

**Limitations:**

The limitations are reasonably well addressed.

**Strengths And Weaknesses:**

This paper is remarkably thorough, considering an impressively broad range of parameters and demonstrating repeatedly why such a generalized framework is useful by applying it and making nontrivial algorithmic/formal contributions. They also serve to provide an important baseline, reducing the gap between these models and those leveraging recent advances such as cascaded diffusion models, without sacrificing (as much) theoretical interpretability. The practical consequences of the suggestions made in this paper are significant, allowing for substantially more efficient sampling, and addressing what has historically been a significant limitation of diffusion models. The paper itself is excellent and beyond its practical and theoretical contributions, represents a powerful and remarkably accessible survey of diffusion model literature (excluding a few nuances like approaches consisting of multiple diffusion models).

One noteworthy detail of this paper is the repeated highlighting on decisions which, while they could be and often are made with some theoretical motivation, are in fact best left to be empirically determined (such as the relative rates of noise decay and injection over time). While it is useful to have these empirical results, and empirical results often yield useful theoretical generalizations, it is less practical to need to perform a “case-by-case … grid search” for each new design decision. In addition, FID and other Inception-based metrics have well-known limitations e.g. [1, 2, 3]. While FID is widely used to evaluate generative models, when analyzing many hyperparameters and exploring some relatively small differences in image quality, it is difficult to know whether some empirical decisions are actually improving the generated images or overfitting to FID. It would be helpful to understand/interpret the tradeoffs made by these decisions in terms of other metrics as well, such as precision and recall [4].

[1] An Improved Evaluation Framework for Generative Adversarial Networks, Liu et al 2018
[2] Pros and cons of GAN evaluation measures, Borji 2019
[3] A Note on the Inception Score, Barratt and Sharma 2018
[4] Improved Precision and Recall Metric for Assessing Generative Models, Kynkäänniemi et al 2019.

---

> ### Author Response · Authors · 2022-08-02
> **Author Response**
>
> The case-by-case grid search for stochasticity hyperparameters is indeed quite expensive. We hope that countering the image degradation effects illustrated in Fig. 13 can be achieved in other ways in the future, reducing the number of these hyperparameters or at least making them less sensitive to the dataset.
>
> We did not perform measurements in metrics other than FID, and thus have not analyzed the diversity vs fidelity tradeoff resulting from different design choices. Given that this tradeoff can be adjusted directly via, e.g., classifier-free guidance, a proper study would have to take such methods into account. It may well be that some of our advocated design choices are not optimal for, say, systems that strongly favor fidelity at the expense of diversity (e.g., Dall-E 2, Imagen), and finding out which design choices suit such systems best is an interesting topic for future work.

---

### Official Review · Reviewer_YPzX · 2022-07-12

**Rating:** 9
**Confidence:** 4
**Soundness:** 4 excellent
**Presentation:** 4 excellent
**Contribution:** 4 excellent

**Summary:**

This work proposes a reformulation of continuous-time diffusion/score-based models that is more modular and easier to analyze. With this new formulation, authors carefully analyze different design choices in those models, including discretization for sampling, parameterization for the score model, stochasticity in the sampling process, noise distributions in training, and scaling schedules of inputs. Experiments demonstrate significant improvements in various settings, creating new/near SOTA results on CIFAR-10 and ImageNet-64.

**Questions:**

The stochastic sampler in Algorithm 2 is different from solving equation (6) with numerical SDE solvers. Why not perform stochastic sampling with existing numerical SDE solvers? How do they fare against the sampler in Algorithm 2?

**Limitations:**

Authors discussed limitations implicitly as avenues for future work, such as the precise interaction between stochasticity and the training objective. A comprehensive discussion on negative societal impacts was also included.

**Strengths And Weaknesses:**

This is an excellent work with many strengths:

1. Unlike previous works, authors formulate diffusion/score-based models around the probability flow ODE. This novel perspective leads to a new analysis on the role of stochasticity in sampling, showing that previous SDE-based formulations are only a special case of combining  the probability flow ODE and the Langevin diffusion SDE with different relative weights. Stochasticity only exists in the Langevin component, which effectively corrects for potential errors in solving the probability flow ODE. This interpretation motivates authors' new stochastic samplers that outperform existing ones on image datasets. I'd like to point it out that a similar reasoning was also used to form the predictor-corrector samplers in ref. [41], where the Langevin component is the "corrector", and the "predictor" can be either a probability flow ODE or a reverse-time SDE.

2. The analysis on deterministic sampling is very convincing, backed by both insightful illustrations (Figure 3) and strong empirical improvements. I especially like the analysis on how to choose the noise schedule $\sigma(t)$ to make the probability flow ODE easy to solve. It is also a valuable contribution to show that 2nd order ODE solver (Heun's method) significantly accelerates sampling.

3. The parameterization of $D_\theta$ in equation (7) is a nice contribution, capturing the intuition that we should predict the denoised image for bigger noise levels, and predict the noise itself for smaller noise levels. The analysis on choosing $\lambda(\sigma)$, $p_{\text{train}}(\sigma)$ and others provides an effective set of hyperparameters for improved empirical performance.

There are no major weaknesses. Below are some thoughts if I have to nitpick:

1. I don't fully agree that the proposed reformulation is more modular than previous ones. It seems to me that all design choices in this work can be translated to the variational formulation of diffusion models, or the continuous-time formulation of score-based generative models. The authors' new formulation seems to facilitate the analysis of stochastic samplers the most. Other contributions can be made with existing formulations with appropriate modifications.

2. Derivations of hyperparameters are not fully principled. They are more or less based on intuitions (such as fixing the magnitudes of input and output signals), or discovered from experiments. This raises a question on whether the same design choice can perform as well for other data domains.

---

> ### Author Response · Authors · 2022-08-02
> **Author Response**
>
> There are several motivations behind the use of a tailored stochastic sampler, some of which are only tersely hinted at in the manuscript due to space constraints. Space permitting, we are happy to expand this section with additional arguments upon request:
>
> - General-purpose SDE solvers must be prepared to tackle fully general SDEs correctly. The SDE in Eq. 6 is a simpler special case, where in particular the diffusion term is independent of $\boldsymbol{x}$.
>
> - The noise replacement schedule $\beta(t)$ in the SDE formulation is a somewhat awkward way to control stochasticity, as its effect depends on discretization step lengths. At a given noise level, the distribution $p(\boldsymbol{x};\sigma)$ corresponds to the data manifold mollified by $\sigma$, suggesting that the appropriate scale for discrete exploration jumps would be proportional to $\sigma$. Thus, at each optimization step we should replace the same proportion of noise, but implementing this in the SDE would require retrofitting details of the discretization step lengths into $\beta(t)$. Instead of doing this, we view the SDE as an inspiration for fusing an explicit Langevin sampling mechanism with the ODE.
>
> - When discretizing Eq. 6 by Euler-Maruyama, there is a subtle discrepancy between the contributions of the denoising and noise injection terms. Between the sub-steps of our algorithm, $\boldsymbol{x}$ and $\sigma$ (or $t$) supplied to $D_\theta$ correspond to the state after noise injection, whereas standard Euler-Maruyama can be interpreted as first adding noise and then performing an ODE step, not from the intermediate state after noise injection, but assuming $\boldsymbol{x}$ and $\sigma$ remained at the initial state at the beginning of the iteration step. In the limit of $\Delta_t$ approaching zero there is no difference, but the distinction appears to become significant when pursuing low NFE with large steps. Experiments not included in the paper indicated that Euler-Maruyama required larger and potentially discretization-dependent noise level corrections via $S_\text{noise}$, whereas in our formulation the optimal value for $S_\text{noise}$ was closer to 1 and not sensitive to such hyperparameters.
>
>   As we did not fully analyze these findings and have no firm theory to support them, we chose to steer away from speculating on the merits of the two-step method vs Euler-Maruyama or higher-order SDE solvers in general, and instead focused on the practical results. The benefit of our approach is evaluated numerically in Figure 4, including a comparison with standard Euler-Maruyama, a predictor-corrector sampler, and a higher-order SDE solver tailored for diffusion models. However, we did not make an attempt to retrofit the noise replacement schedules into the comparison SDEs.

---

> > ### Comment · Reviewer_YPzX · 2022-08-09
> > **Thanks for the response.**
> >
> > Thank you for clarifying the motivation of your tailored stochastic sampler. My rating about this paper stays unchanged.

---

### Official Review · Reviewer_er1y · 2022-07-12

**Rating:** 8
**Confidence:** 4
**Soundness:** 4 excellent
**Presentation:** 3 good
**Contribution:** 4 excellent

**Summary:**

This paper studies the algorithmic design space of diffusion models. In doing so, the authors make the following contributions:
(a) identify and characterize the degrees of freedom w.r.t. sampling, network parameterization, training
(b) propose to disentangle sampling from training and consequently, design schemes for both deterministic and stochastic sampling based on higher-order Runge Kutta solvers that significantly reduce the number of function evaluations
(c) improve training by better preconditioning of inputs, outputs, training losses and augmentation schemes
Empirically, these improvements translate to sota sample quality on standard image generation benchmarks

**Questions:**

- See last 2 points in the weaknesses para above and please let me know your thoughts
- I wasn't quite clear on why the authors choose to discuss the generalized ODE in Eq. 4 when eventually they propose to set s(t)=1 which reduces to Eq (1)
- Is there any specific reason to exclude Imagenet-64 for evaluating the effect of preconditioning in Table 2? I am particularly curious because I was hoping to contrast it with the numbers in Fig. 4c for stochastic sampling. Since we might intuitively expect that stochastic sampling could relatively be more beneficial for diverse datasets, I wonder if that intuition carries over empirically.
- Finally, while the paper does characterize the design space reasonably well, I am curious if the authors found that their final diffusion model is robust to different choices (not just their prescribed choice) of some of the design parameters in Table 1.

**Limitations:**

The authors discuss negative results in the context of stochastic sampling. Perhaps a broader discussion of negative results and context around their empirical proposals could help --- concretely, one example of such a discussion point is: which design choices are data-dependent and on what properties (eg, size, dimensionality, modality)? Such information could vastly aid practitioners looking to train and deploy these models on their custom data and modalities.

**Strengths And Weaknesses:**

Strengths:

- This paper presents a timely and important contribution to an area of growing significance: deep generative modeling using diffusion models. Arguably, these models, while clearly having a lot of potential given their recent successes, are notorious to train requiring several training tricks.
- Related to the above, the paper also stands out in contrasting existing models and algorithms --- in a way, the community can look at this paper both as an excellent survey of some major works in diffusion models while providing a generalized characterization that enables the development of newer models.
- Finally, the resulting improvements on CIFAR-10 and Imagenet are impressive -- the paper truly achieves the best of both worlds as the number of function evaluations reduce very significantly while also improving the sample quality metrics in both scenarios of deterministic and stochastic sampling. I also liked learning about some negative results and intuitions on their failures --- such as the use of stochastic samplers leading to oversaturation --  I have observed this to be a common problem in the results of some past works in diffusion models, but couldn't come across a satisfying explanation.

Weaknesses:

- While such a paper is not easy to write as it covers a bunch of distinct contributions, I found it a little hard to review constantly shuttling between the main text and the appendix. Some of the very important and interesting discussions such as those concerning the step sizes in Section 3 were delegated to the appendix, making readability slightly hard.
- At certain places, it is hard to distinguish what is prior work vs what is new --- this isn't a question re: novelty of this paper, but more on the exposition lacking discussion and details. For example, Jolicoeur-Martineau et al. (duly cited) also explore the use of higher order solvers for sampling-- what exactly is different between the current work and the one before? Is it the scheme for setting timesteps to minimize truncation error in Eq. 5 or something else?
- I think some of the claims can be placed better in context or need more empirical justification. For example, in L103-104, the authors highlight there are no dependencies between the components but does this hold broadly for all parameters in Table 1? One extreme interpretation of this could be eg, we can mix and match preconditioning schemes, i.e., skip scaling from model A (say VP), output scaling from model B (say DDIM), etc. and still train a diffusion model reasonably well. Given the brittleness of these models, I am not sure if such arbitrary mix-and-match schemes will translate to a functioning model with reasonable performance.

---

> ### Author Response · Authors · 2022-08-02
> **Author Response**
>
> We agree that the split between main paper and appendix is not entirely satisfying. Our original draft was considerably longer than permitted by the conference, and some interesting findings and analysis unfortunately had to be moved to the appendix.
>
> The differences between Jolicoeur-Martineau et al. and our method (Fig. 4) include the choices for noise and scale schedules and the discretization time steps, all established in Section 3, as well as our SDE solver limiting the stochasticity to noise range determined by $S_\text{tmin}$ and $S_\text{tmax}$ and adjusting the level via $S_\text{noise}$. Our SDE solver is also structured slightly differently. As Fig. 4 illustrates, these choices have a major impact on result quality, and using a 2nd order solver does not in itself guarantee low FIDs.
>
> The statement about modularity refers to the theoretical independence between components listed in Table 1, i.e., that changing one component does not necessitate changes elsewhere in order to, e.g., maintain the property that the model converges to the data in the limit. In this sense, one could indeed mix and match, say, preconditioning schemes from different models, although in practice training might be more difficult, and the results might be worse. We shall clarify this in the text.
>
> We consider arbitrary noise level and scale schedules in Sections 2 and 3 in order to properly analyze previous methods in our theoretical framework. Based on evaluating VE, VP and DDIM in this common framework, we then standardize to $\sigma(t)=t$ and $s(t)=1$ and reflect this in the formulas in Section 4 onwards and Algorithm 2 to reduce the (significant!) notational clutter.
>
> ImageNet-64 was not included in Section 5 evaluation due to our lack of compute capacity to train the model in time for submission. All prior experiments could be performed with pre-trained networks, but those in Section 5 could not, which limited us to smaller datasets. We should have results for ImageNet-64 fairly soon, though, so we can still add those in the camera-ready version.
>
> We did not re-evaluate earlier design choices after arriving at our final models. Like well-optimized models in general, we believe they are not very sensitive to small changes in hyperparameters or minor design details. However, modifying high-level choices such as noise or scale schedule would most likely have a large effect on result quality.
>
> Regarding the data-dependence of design choices, it appears that only the stochasticity parameters are heavily dependent on the dataset (Table 5). The higher-level choices seem to hold across datasets and score network architectures -- with the caveat that our tests have been limited to small resolutions and mostly small datasets.

---

> > ### Comment · Reviewer_er1y · 2022-08-06
> > **Response Ack**
> >
> > Thanks for your response. I'll stick to my original rating recommending a Strong Accept.

---

### Meta-Review · Area_Chair_Zo5n · 2022-08-23

**Recommendation:** Accept
**Confidence:** Certain

**Metareview:**

Ratings: 8/9/8/7.
Confidence: 4/4/4/5.
Discussion among reviewers: No.

Summary: This is an excellent paper analyzing the design space of diffusion models. The paper clarifies the design space by disentangling the effects of (1) parameterization, (2) sampling, and (3) training separately. The researchers uniformily agree that the paper is well written and that the empirical results are impressive. Given the enormous interest in diffusion models in the research community, and the likely high impact of advancements in this subfield, this paper is well timed, and will probably be very well received by the NeurIPS community.

Decision: I highly recommend to accept this paper.

**Award:**

Yes

---

### Decision · Program_Chairs · 2022-09-14

Accept